# Aligned and oblique dynamics in recurrent neural networks

**Friedrich Schuessler[1,2]\*, Francesca Mastrogiuseppe[3], Srdjan Ostojic[4], Omri Barak[5]**

[1]Faculty of Electrical Engineering and Computer Science, Technical University of Berlin, Berlin, Germany; [2]Science of Intelligence, Research Cluster of Excellence, Berlin, Germany; [3]Champalimaud Foundation, Lisbon, Portugal; [4]Laboratoire de Neurosciences Cognitives et Computationnelles, INSERM U960, Ecole Normale Superieure-PSL Research University, Paris, France; [5]Rappaport Faculty of Medicine and Network Biology Research Laboratories, Technion - Israel Institute of Technology, Haifa, Israel

## eLife Assessment

This work provides an **important** and novel framework for interpreting the interactions between recurrent dynamics across stages of neural processing. The authors report that two different kinds of dynamics exist in recurrent networks differing in the extent to which they align with the output weights. The authors also present **convincing** evidence that both types of dynamics exist in the brain.

**\*For correspondence:**
f.schuessler@tu-berlin.de

**Abstract** The relation between neural activity and behaviorally relevant variables is at the heart of neuroscience research. When strong, this relation is termed a neural representation. There is increasing evidence, however, for partial dissociations between activity in an area and relevant external variables. While many explanations have been proposed, a theoretical framework for the relationship between external and internal variables is lacking. Here, we utilize recurrent neural networks (RNNs) to explore the question of when and how neural dynamics and the network's output are related from a geometrical point of view. We find that training RNNs can lead to two dynamical regimes: dynamics can either be aligned with the directions that generate output variables, or oblique to them. We show that the choice of readout weight magnitude before training can serve as a control knob between the regimes, similar to recent findings in feedforward networks. These regimes are functionally distinct. Oblique networks are more heterogeneous and suppress noise in their output directions. They are furthermore more robust to perturbations along the output directions. Crucially, the oblique regime is specific to recurrent (but not feedforward) networks, arising from dynamical stability considerations. Finally, we show that tendencies toward the aligned or the oblique regime can be dissociated in neural recordings. Altogether, our results open a new perspective for interpreting neural activity by relating network dynamics and their output.

## Introduction

The relation between neural activity and behavioral variables is often expressed in terms of neural representations. Sensory input and motor output have been related to the tuning curves of single neurons (*Hubel and Wiesel, 1962*; *O'Keefe and Dostrovsky, 1971*; *Hafting et al., 2005*) and, since the advent of large-scale recordings, to population activity (*Buonomano and Maass, 2009*; *Saxena and Cunningham, 2019*; *Vyas et al., 2020*). Both input and output can be decoded from population activity (*Churchland et al., 2012*; *Mante et al., 2013*), even in real-time, closed-loop settings (*Sadtler*

*et al., 2014*; *Willett et al., 2021*). However, neural activity is often not fully explained by observable behavioral variables. Some components of the unexplained neural activity have been interpreted as random trial-to-trial fluctuations (*Galgali et al., 2023*), potentially linked to unobserved behavior (*Stringer et al., 2019b*; *Musall et al., 2019*; *Wang et al., 2023*). Activity may further be due to other ongoing computations not immediately related to behavior, such as preparatory motor activity in a null space of the motor readout (*Kaufman et al., 2014*; *Hennequin et al., 2014*). Finally, neural activity may partially be due to other constraints, e.g., related to the underlying connectivity (*Atallah and Scanziani, 2009*; *Okun and Lampl, 2008*), the process of learning (*Sadtler et al., 2014*), or stability, i.e., the robustness of the neural dynamics to perturbations (*Russo et al., 2020*).

Here, we aim for a theoretical understanding of neural representations: Which factors might determine how strongly activity and behavioral output variables are related? To this end, we use trained recurrent neural networks (RNNs). In this setting, output variables are determined by the task at hand, and neural activity can be described by its projection onto the principal components (PCs). We show that these networks can operate between two extremes: an 'aligned' regime in which the output weights and the largest PCs are strongly correlated, and a second, 'oblique' regime, where the output weights and the largest PCs are poorly correlated.

What determines the regime in which a network operates? We show that quite general considerations lead to a link between the magnitude of output weights and the regime of the network. As a consequence, we can use output magnitude as a control knob for trained RNNs. Indeed, when we trained RNN models on different neuroscience tasks, large output weights led to oblique dynamics, and small output weights to aligned dynamics. Recent results in feedforward networks identified two regimes – rich and lazy – that can also arise from choices of output weights (*Chizat et al., 2019*; *Jacot et al., 2018*). In an extensive Methods section, we further analyze in detail how the oblique and aligned regimes arise during learning. There we show that the dynamical nature of RNNs, in particular demanding stable dynamics, leads to the replacement of unstable, lazy, solutions by oblique ones.

We then considered the functional consequences of the two regimes. Building on the concept of feedback loops driving the network dynamics (*Sussillo and Abbott, 2009*; *Rivkind and Barak, 2017*), we show that, in the aligned regime, the largest PCs and the output are qualitatively similar. In the oblique regime, in contrast, the two may be qualitatively different. This functional decoupling

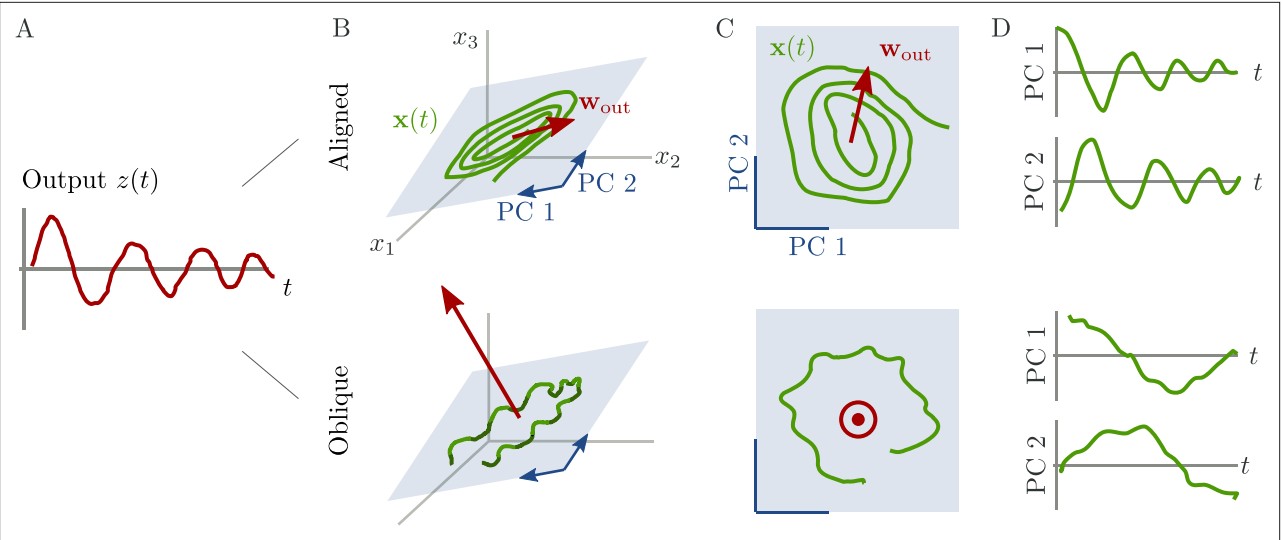

**Figure 1.** Schematic of aligned and oblique dynamics in recurrent neural networks. (**A**) Output generated by both networks. (**B**) Neural activity of aligned (top) and oblique (bottom) dynamics, visualized in the space spanned by three neurons. Here, the activity (green) is three-dimensional, but most of the variance is concentrated along the two largest principal components (PCs) (blue). For aligned dynamics, the output weights (red) are small and lie in the subspace spanned by the largest PCs; they are hence correlated to the activity. For oblique dynamics, the output weights are large and lie outside of the subspace spanned by the largest PCs; they are hence poorly correlated to the activity. (**C**) Projection of activity onto the two largest PCs. For oblique dynamics, the output weights are orthogonal to the leading PCs. (**D**) Evolution of PC projections over time. For aligned dynamics, the projection on the PCs resembles the output $z(t)$, and reconstructing the output from the largest two components is possible. For the oblique dynamics, such reconstruction is not possible, because the projections oscillate much more slowly than the output.

in oblique networks leads to a large freedom for neural dynamics. Different networks with oblique dynamics thus tend to employ different dynamics for the same tasks. Aligned dynamics, in contrast, are much more stereotypical. Furthermore, as a result of how neural dynamics and output are coupled, oblique and aligned networks react differently to perturbations of the neural activity along the output direction. In particular, oblique (but not aligned) networks develop an additional negative feedback loop that suppresses output noise. We finally show that neural recordings from different experiments can have different degrees of alignment, which indicates that our theoretical results may be useful in identifying different regimes for different experiments, tasks, or brain regions.

Altogether, our work opens a new perspective relating network dynamics and their output, yielding important insights for modeling brain dynamics as well as experimentally accessible questions about learning and dynamics in the brain.

## Results

### Aligned and oblique population dynamics

We consider an animal performing a task while both behavior and neural activity are recorded. For example, the task might be to produce a periodic motion, described by the output $z(t)$ of *Figure 1A*. For simplicity, we assume that the behavioral output can be decoded linearly from the neural activity (*Mante et al., 2013*; *Sadtler et al., 2014*; *Gallego et al., 2017*; *Russo et al., 2018*; *Willett et al., 2021*). We can thus write

$$z(t) = \sum_{i=1}^{N} w_{\text{out},i}\, x_i(t) = \mathbf{w}_{\text{out}}^T \mathbf{x}(t)\,, \tag{1}$$

with readout weights $\mathbf{w}_{\text{out}}$. The activity of neuron $i \in \{1, \ldots, N\}$ is given by $x_i(t)$, and we refer to the vector $\mathbf{x}$ as the state of the network.

Neural activity has to generate the output in some subspace of the state space, where each axis represents the activity of one neuron. In the simplest case (*Figure 1B*, top), the output is produced along the largest PCs of activity, as shown by the fact that projecting the neural activity $\mathbf{x}(t)$ onto the largest PCs returns the target oscillation (*Figure 1D*, top). We call such dynamics 'aligned' because of the alignment between the subspace spanned by the largest PCs and the output vector (red).

There is, however, another possibility. Neural activity may have many other components not directly related to the output and these other components may even dominate the overall activity. In this case (*Figure 1B and D*, bottom), the two largest PCs are not enough to read out the output, and smaller PCs are needed. We call such dynamics 'oblique' because the subspace spanned by the largest PCs and the output vector are poorly aligned.

We consider these two possibilities as distinct dynamical regimes, noting that intermediate situations are also possible. The actual regime of neural dynamics has important consequences for how one interprets neural recordings. For aligned dynamics, analyzing the dynamics within the largest PCs may lead to insights about the computations generating the output (*Vyas et al., 2020*). For oblique dynamics, such an analysis is hampered by the dissociation between the large PCs and the components generating the output (*Russo et al., 2018*).

### Magnitude of output weights controls regime in trained RNNs

What determines which regime a neural network operates in? Given that behavior is the same, but representations differ, we study this question using trained RNNs. This framework constrains what networks do, but not how they do it (*Sussillo, 2014*; *Barak, 2017*). The specific property of representation we are interested in is the alignment, or correlation, between output weights and states:

$$\rho(t) = \mathbf{w}_{\text{out}}^T \mathbf{x}(t) / \left( \|\mathbf{w}_{\text{out}}\|\, \|\mathbf{x}(t)\| \right)\,, \tag{2}$$

where the vector norms $\|\mathbf{w}_{\text{out}}\|$ and $\|\mathbf{x}\|$ quantify the magnitude of each vector.

For aligned dynamics, the correlation is large, corresponding to the alignment between the leading PCs of the neural activity and the output weights (*Figure 1B*, top). In contrast, for oblique dynamics,

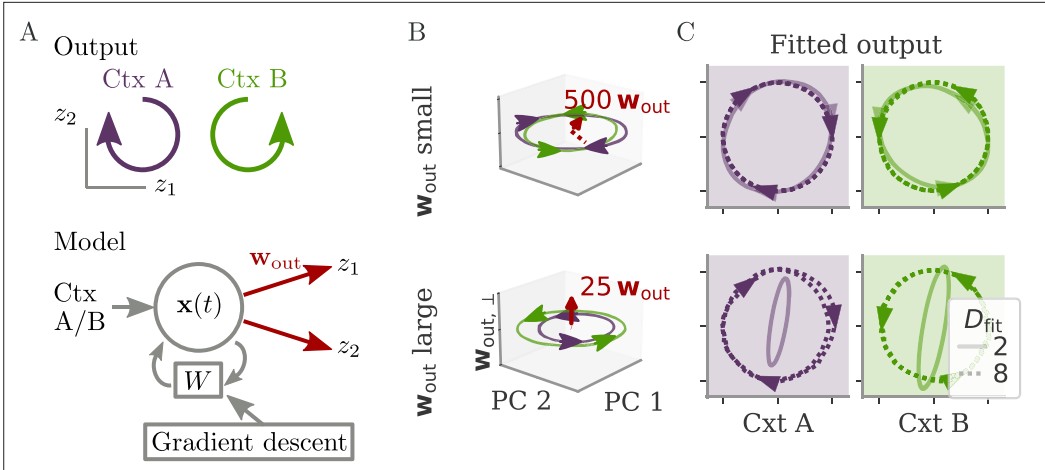

**Figure 2.** Aligned and oblique dynamics for a cycling task (*Russo et al., 2018*). (**A**) A network with two outputs was trained to generate either clockwise or anticlockwise rotations, depending on the context (top). Our model recurrent neural network (RNN) (bottom) received a context input pulse, generated dynamics $\mathbf{x}(t)$ via recurrent weights $W$, and yielded the output as linear projections of the states. We trained the recurrent weights $W$ with gradient descent. (**B, C**) Resulting internal dynamics for two networks with small (top) and large (bottom) output weights, corresponding to aligned and oblique dynamics, respectively. (**B**) Dynamics projected on the first two principal components (PCs) and the remaining direction $\mathbf{w}_{\mathrm{out},\perp}$ of the first output vector (for $z_1$). The output weights are amplified to be visible. Arrowheads indicate the direction of the dynamics. Note that for the large output weights, the dynamics in the first two PCs co-rotated, despite the counter-rotating output. (**C**) Output reconstructed from the largest PCs, with dimension $D = 2$ (full lines) or 8 (dotted). Two dimensions already yield a fit with $R^2 = 0.99$ for aligned dynamics (top), but almost no output for oblique (bottom, $R^2 = 0.005$, no arrows shown). For the latter, a good fit with $R^2 > 90\%$ is only reached with $D = 8$.

this correlation is small (*Figure 1B*, bottom). Note that the concept of correlation can be generalized to accommodate multiple time points and multidimensional output (see section Generalized correlation).

Studying the same task means that the output $z$ is the same, so it is instructive to express it in terms of the correlation $\rho$:

$$z(t) = \rho(t)\,\|\mathbf{w}_{\mathrm{out}}\|\,\|\mathbf{x}(t)\|\,. \tag{3}$$

Recent work on feedforward networks showed that $\|\mathbf{w}_{\mathrm{out}}\|$ can have a large effect on the resulting representations (*Jacot et al., 2018*; *Chizat et al., 2019*). *Equation 3* shows that $\rho$ is indeed linked to $\|\mathbf{w}_{\mathrm{out}}\|$, but $\|\mathbf{x}(t)\|$ can also vary. In a detailed analysis in the Methods (sections Analysis of solutions under noiseless conditions to Oblique solutions arise for noisy, nonlinear systems), we show that for recurrent networks, stability considerations preclude $\|\mathbf{x}(t)\|$ from being small. This implies that if we choose a readout norm $\|\mathbf{w}_{\mathrm{out}}\|$ and then train the RNN on a given task, the correlation must compensate.

If we choose small output weights, we expect aligned dynamics, because a large correlation is necessary to generate sufficiently large output (*Figure 1B*, top). If instead we choose large output weights, we expect oblique dynamics, because only a small correlation keeps the output magnitude from growing too large.

We tested whether output weights can serve as a control knob to select dynamical regimes using an RNN model trained on an abstract version of the cycling task introduced in *Russo et al., 2018*. The networks were trained to generate a 2D signal that rotated in the plane spanned by two outputs $z_1(t)$ and $z_2(t)$ (*Figure 2A*). An input pulse at the beginning of each trial indicated the desired direction of rotation. We set up two models with either small or large output weights and trained the recurrent weights of each with gradient descent (section Details on RNN models and training).

After both models learned the task, we projected the network activity into a three-dimensional (3D) space spanned by the two largest PCs of the dynamics $\mathbf{x}(t)$. A third direction, $\mathbf{w}_{\mathrm{out},\perp}$, spanned the remaining part of the first output vector $\mathbf{w}_{\mathrm{out},1}$. The resulting plots, *Figure 2B*, corroborate our

hypothesis: Small output weights led to aligned dynamics with a large correlation between the largest PCs and the output weights. In contrast, the large output weights of the second network were almost orthogonal, or oblique, to the two leading PCs. Further qualitative differences between the two solutions in terms of the direction of trajectories will be discussed below.

Another way to quantify these regimes is by the ability to reconstruct the output from the large PCs of neural activity, as quantified by the coefficient of determination $R^2$. For the aligned network, the projection on the two largest PCs (*Figure 2C*, solid) already led to a good reconstruction. For the oblique networks, the two largest PCs were not sufficient. We needed the first eight dimensions (*Figure 2C*, dashed) to obtain a good reconstruction ($R^2 > 0.9$). In contrast to the differences in these fits, the neural dynamics themselves were much more similar between the networks. Specifically, 90% of the variance was explained by four and five dimensions for the aligned and oblique networks, respectively.

Can we use the output weights to induce aligned or oblique dynamics in more general settings? We trained RNN models with small or large initial output weights on five different neuroscience tasks (section Task details). All weights (input, recurrent, and output) were trained using the Adam algorithm (section Details on RNN models and training). After training, we measured the three quantities of

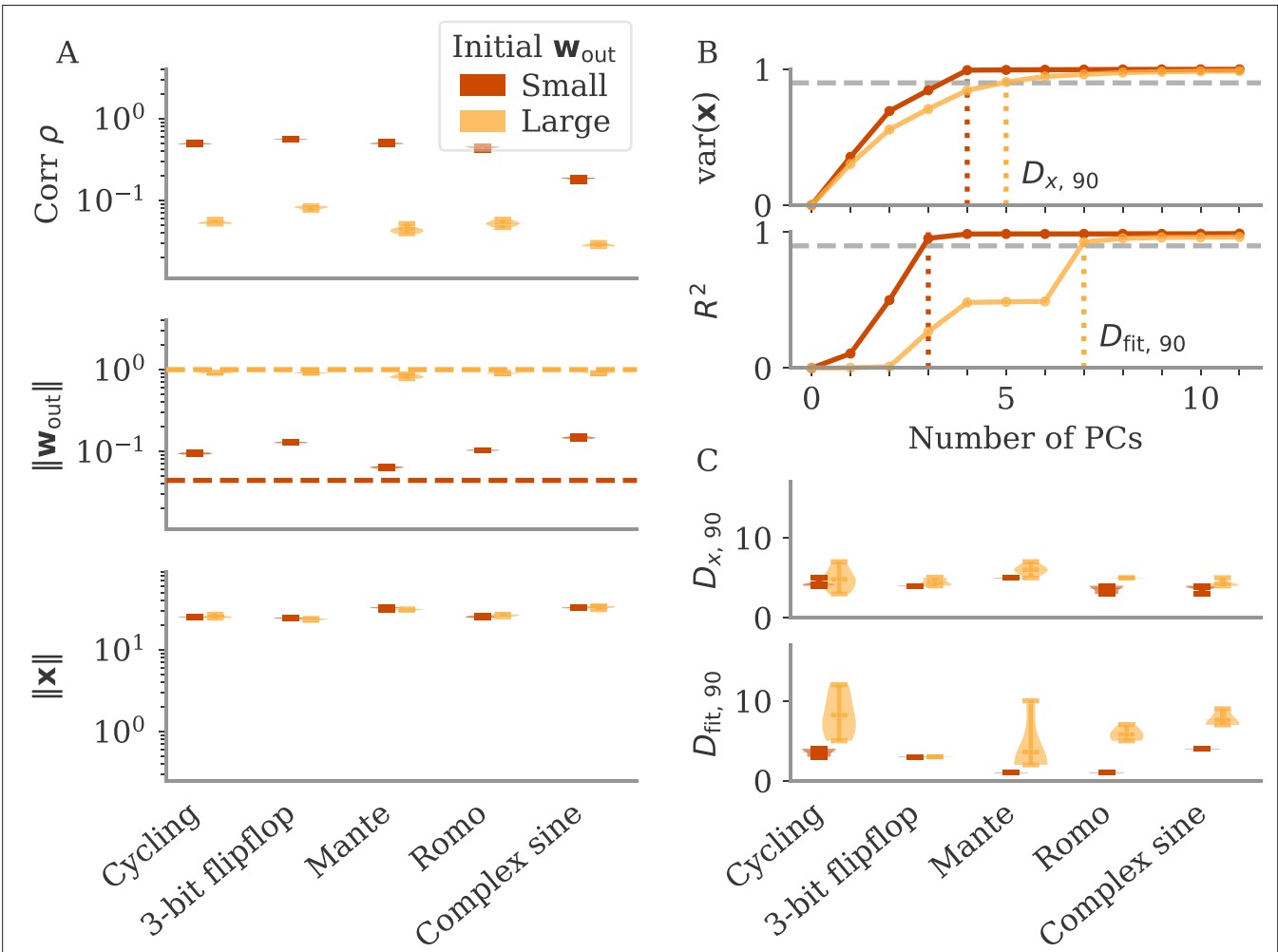

**Figure 3.** The magnitude of output weights determines regimes across multiple neuroscience tasks (*Russo et al., 2018*; *Mante et al., 2013*; *Romo et al., 1999*; *Sussillo and Barak, 2013*). (**A**) Correlation and norms of output weights and neural activity. For each task, we initialized networks with small or large output weights (dark vs light orange). The initial norms $\|\mathbf{w}_{out}\|$ are indicated by the dashed lines. Learning only weakly changes the norm of the output weights. Note that all y-axes are logarithmically scaled. (**B**) Variance of $\mathbf{x}$ explained and $R^2$ of reconstructed output for projections of $\mathbf{x}$ on increasing number of principal components (PCs). Results from one example network trained on the cycling task are shown for each condition. (**C**) Number of PCs necessary to reach 90% of the variance of $\mathbf{x}(t)$ or of the $R^2$ of the output reconstruction (top/bottom; dotted lines in **B**). In (**A, C**) violin plots show the distribution over five sample networks, with vertical bars indicating the mean and the extreme values (where visible).

*Equation 3*: magnitudes of neural activity and output weights, and the correlation between the two. The results in *Figure 3A* show that across tasks, initialization with large output weights led to oblique dynamics (small correlation), and with small output weights to aligned dynamics (large correlation). While training could, in principle, change the initially small output weights to large ones (and vice versa), we noticed that this does not happen. Small output weights did increase with training, but the large gap in norms remained. This shows that setting the output weights at initialization can serve to determine their scale after learning under realistic settings. While explaining this observation is beyond the scope of this work, we note that (1) changing the internal weights suffices to solve the task, and (2) the extent to which the output weights change during learning depends on the algorithm and specific parameterization (*Jacot et al., 2018*; *Geiger et al., 2020*; *Yang and Hu, 2020*).

In *Figure 3B and C*, we adopted the perspective of *Figure 2C* and quantified how well we can reconstruct the output from a projection of **x** onto its largest $D$ PCs (section Regression). As expected, both the variance of **x** explained and the quality of output reconstruction increased, for an increasing number of PCs $D$ (*Figure 3B*). How both quantities increased, however, differs between the two regimes. While the variance explained increased similarly in both cases, the quality of the reconstruction increased much more slowly for the model with large output weights. We quantified this phenomenon by comparing the dimensions at which either the variance of **x** explained or $R^2$ reaches 90%, denoted by $D_{x,90}$ and $D_{\text{fit},90}$, respectively.

In *Figure 3C*, we compare $D_{x,90}$ and $D_{\text{fit},90}$ across multiple networks and tasks. Generally, larger output weights led to larger numbers for both. However, for large output weights, the number of PCs necessary to obtain a good reconstruction increased much more drastically than the dimension of the data. Thus, the output was less well represented by the large PCs of the dynamics for networks with large output weights, in agreement with our notion of oblique dynamics.

Importantly, reaching the aligned and oblique regimes relies on ensuring robust and stable dynamics, which we achieve by adding noise to the dynamics during training. This yields a similar magnitude of neural activity $\|\mathbf{x}\|$ across networks and tasks (*Figure 3A*). We show in Methods, section Analysis of solutions under noiseless conditions, that learning in simple, noise-free conditions with large output weights can lead to solutions not captured by either aligned or oblique dynamics; those solutions, however, are unstable. Furthermore, we observed that some of the qualitative differences between aligned and oblique dynamics are less pronounced if we initialized networks with small recurrent weights and initially decaying dynamics (*Appendix 1—figure 2*).

## Neural dynamics decouple from the output for the oblique regime

What are the functional consequences of the two regimes? A hint might be seen in an intriguing qualitative difference between the aligned and oblique solutions for the cycling task in *Figure 2*. For the

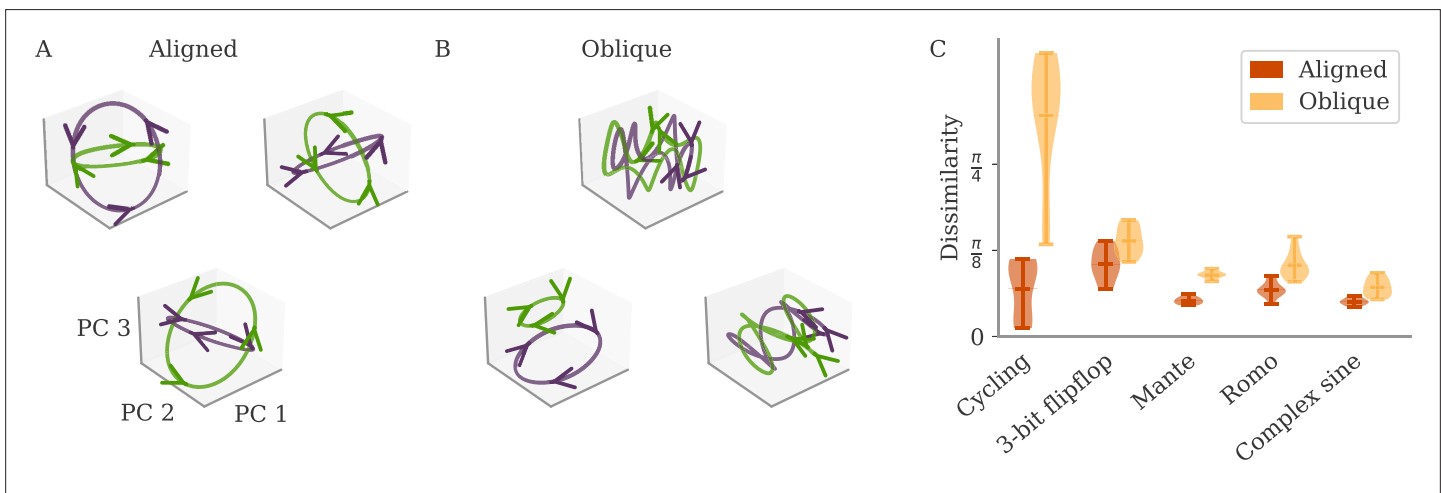

**Figure 4.** Variability between learners for the two regimes. (**A, B**) Examples of networks trained on the cycling task with small (aligned) or large (oblique) output weights. The top left and central networks, respectively, are the same as those plotted in *Figure 2*. (**C**) Dissimilarity between solutions across different tasks. Aligned dynamics (red) were less dissimilar to each other than oblique ones (yellow). The violin plots show the distribution over all possible different pairs for five samples (mean and extrema as bars).

aligned network, the two trajectories for the two different contexts (green and purple) are counter-rotating (*Figure 2B*, top). This agrees with the output, which also counter-rotates as demanded by the task (*Figure 2A*). In contrast, the neural activity of the oblique network *co-rotates* in the leading two PCs (*Figure 2B*, bottom). This is despite the counter-rotating output, since this network also solves the task (not shown). The co-rotation also indicates why reconstructing the output from the leading two PCs is not possible (*Figure 2C*). Naturally, the dynamics also contain counter-rotating trajectories for producing the correct output, but these are only present in low-variance PCs. (Note also that for aligned networks, one can also observe co-rotation in low-variance PCs, see *Appendix 1—figure 3*.) Taken together, aligned and oblique dynamics differ in the coupling between leading neural dynamics and output. For aligned dynamics, the two are strongly coupled. For oblique dynamics, the two decouple qualitatively.

Such a decoupling for oblique, but not aligned, dynamics leads to a prediction regarding the universality of solutions (*Maheswaranathan et al., 2019*; *Turner et al., 2021*; *Pagan et al., 2022*). For aligned dynamics, the coupling implies that the internal dynamics are strongly constrained by the task. We thus expect different learners to converge to similar solutions, even if their initial connectivity is random and unstructured. In *Figure 4A*, we show the dynamics of three randomly initialized aligned networks trained on the cycling task, projected onto the three leading PCs. Apart from global rotations, the dynamics in the three networks are very similar.

For oblique dynamics, the task-defined output exerts weaker constraints on the internal dynamics. Any variability experienced during learning can potentially build up, and eventually create qualitatively different solutions. Three examples of oblique networks solving the cycling tasks indeed show visibly different dynamics (*Figure 4B*). Further analysis shows that the models also differ in the frequency components in the leading dynamics (*Appendix 1—figure 4*).

The degree of variability between learners depends on the task. The observable differences in the PC projections were most striking for the cycling task. For the flip-flop task, for example, solutions were generally noisier in the oblique regime than in the aligned but did not have observable qualitative differences in either regime (*Appendix 1—figure 5*). We quantified the difference between models for the different neuroscience tasks considered before. To compare different neural dynamics, we used a dissimilarity measure invariant under rotation (section Dissimilarity measure) (*Williams et al., 2021*). The results are shown in *Figure 4C*. Two observations stand out: First, across tasks, the dissimilarity was higher for networks in the oblique regime than for those in the aligned. Second, both overall dissimilarity and the discrepancy between regimes differed strongly between tasks. The largest dissimilarity (for oblique dynamics) and the largest discrepancy between regimes was found for the cycling. The smallest discrepancy between regimes was found for the flip-flop task. Such a difference between tasks is consistent with the differences in the range of possible solutions for different tasks, as reported in *Turner et al., 2021*; *Maheswaranathan et al., 2019*.

What are the underlying mechanisms for the qualitative decoupling in oblique, but not aligned networks? For aligned dynamics, we saw that the small output weights demand large activity to generate the output. In other words, the activity along the largest PCs must be coupled to the output. For oblique dynamics, this constraint is not present, which opens the possibility for small components outside the largest PCs to generate the output. If this is the case, we have a decoupling, such as the observed co-rotation in the cycling task, and the possible variability between solutions. We discuss this point in more detail in the Methods, section Mechanisms behind decoupling of neural dynamics and output.

In the following two sections, we will explore how the decoupling between neural dynamics and output for oblique, but not aligned, dynamics influences the response to perturbations and the effects of noise during learning.

## Differences in response to perturbations

Understanding how networks respond to external perturbations and internal noise requires some insight into how dynamics are generated. Dynamics of trained networks are mostly generated internally, through recurrent interactions. In robust networks, these internally generated dynamics are a prominent part of the largest PCs (among input-driven components; section Analysis of solutions under noiseless conditions and Oblique solutions arise for noisy, nonlinear systems). Internally generated dynamics are sustained by positive feedback loops, through which neurons excite each other.

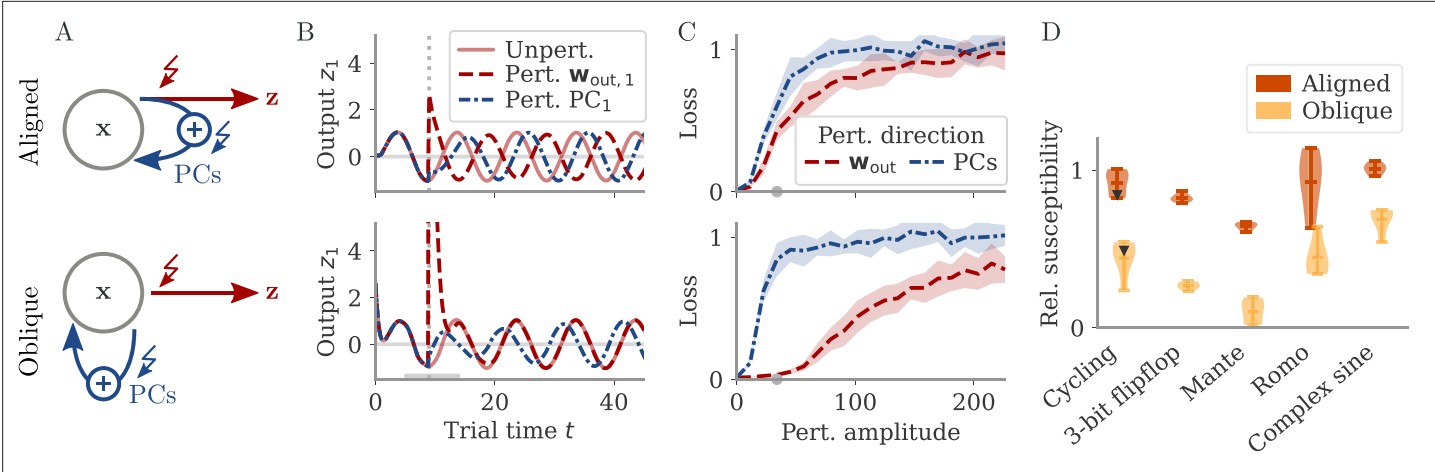

**Figure 5.** Perturbations differentially affect dynamics in the aligned and oblique regimes. (**A**) Cartoon illustrating the relationship between perturbations along output weights or principal components (PCs) and the feedback loops driving autonomous dynamics. (**B**) Output after perturbation for aligned (top) and oblique (bottom) networks trained on the cycling task. The unperturbed network (light red line) yields a sine wave along the first output direction $z_1$. At $t_p = 9$, a perturbation with amplitude $\|\Delta \mathbf{x}\| = 34$ is applied along the output weights (dashed red) or the first PC (dashed-dotted blue). The perturbations only differ in the directions applied. While the immediate response for the oblique network to a perturbation along the output weights is much larger, $z_1(t_p) \approx 80$, the long-term dynamics yield the same output as the unperturbed network. See also **Appendix 1—figure 6** for more details. (**C**) Loss for perturbations of different amplitudes for the two networks in (**B**). Lines and shades are means and standard deviations over different perturbation times $t_p \in [5, 15]$ and random directions spanned by the output weights (red) or the two largest PCs (blue). The loss is the mean squared error between output and target for $t > 20$. The gray dot indicates an example in (**B**). (**D**) Relative susceptibility of networks to perturbation directions for different tasks and dynamical regimes. We measured the area under the curve (AUC) of loss over perturbation amplitude for perturbations along the output weights of the two largest PCs. The relative susceptibility is the ratio between the two AUCs. The example in (**C**) is indicated by gray triangles.

Those loops are low-dimensional, with activity along a few directions of the dynamics being amplified and fed back along the same directions. This results in dynamics being driven by effective feedback loops along the largest PCs (**Figure 5A**). As shown above, the largest PCs can either be aligned or not aligned, with the output weights. This leads to predictions about how aligned and oblique networks differentiate in their responses to perturbations along different directions.

Our intuition about feedback loops suggests that networks respond strongly to a perturbation that is aligned with the directions contributing to the feedback loop, but weakly to a perturbation that is orthogonal to them. In particular, if a perturbation is applied along the output weights, aligned and oblique dynamics should dissociate, with a strong disruption of dynamics for aligned, but not for oblique dynamics (**Figure 5A**).

To test this, we compare the response to perturbations along the output direction and the largest PCs. We apply perturbations to the neural activity at a single point in time: $\mathbf{x}(t)$ evolves undisturbed until time $t_p$. At that point, it is shifted to $\mathbf{x}(t_p) + \Delta \mathbf{x}$. After the perturbation, we let the network evolve freely and compare this evolution to that of an unperturbed copy. Such a perturbation mimics a very short optogenetic perturbation applied to a selected neural population (**O'Shea et al., 2022**; **Finkelstein et al., 2021**). In **Figure 5B**, we show the output after such perturbations for an aligned (top) and an oblique network (bottom) trained on the cycling task. The time point and amplitude are the same for both directions and networks. For each network and type of perturbation, there is an immediate deflection and a long-term response. For both networks, perturbing along the PCs (blue) leads to a long-term phase shift. Only in the aligned network, however, perturbation along the output direction (red) leads to a visible long-term response. In the oblique network, the amplitude of the immediate response is larger, but the long-term response is *smaller*. Our results for the oblique network, but not for the aligned, agree with simulations of networks generating EMG data from the cycling experiments (**Saxena et al., 2022**).

To quantify the relative long-term susceptibility of networks to perturbations along output weights or PCs, we sampled from different times $t_p$ and different directions in the 2D subspaces spanned either by the two output vectors or by the two largest PCs. For each perturbation, we measured the

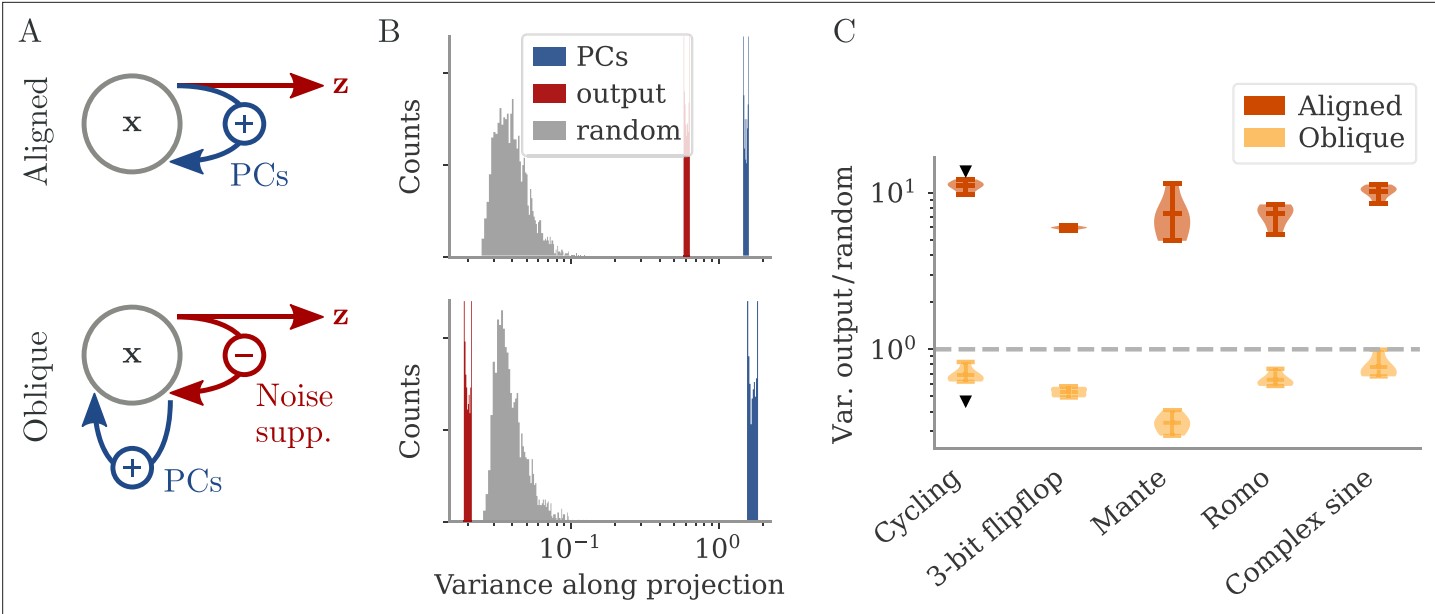

**Figure 6.** Noise suppression along the output direction in the oblique regime. (**A**) A cartoon of the feedback loop structure for aligned (top) and oblique (bottom) dynamics. The latter develops a negative feedback loop which suppresses fluctuations along the output direction. (**B**) Comparing the distribution of variance of mean-subtracted activity along different directions for networks trained on the cycling task (see **Appendix 1—figure 7**): principal components (PCs) of trial-averaged activity (blue), readout (red), and random (gray) directions. For the PCs and output weights, we sampled 100 normalized combinations of either the first two PCs or the two output vectors. For the random directions, we drew 1000 random vectors in the full, $N$-dimensional space. (**C**) Noise compression across tasks as measured by the ratio between variance along output and random directions. The dashed line indicates neither compression nor expansion. Black markers indicate the values for the two examples in (**B, C**). Note the log-scales in (**B, C**).

loss of the perturbed networks on the original task (excluding the immediate deflection after the perturbation by starting to compute the loss at $t_p + 5$). **Figure 5C** shows that the aligned network is almost equally susceptible to perturbations along the PCs and the output weights. In contrast, the oblique network is much more susceptible to perturbations along the PCs.

We repeated this analysis for oblique and aligned networks trained on the five different tasks. We computed the area under the curve (AUC) for both loss profiles in **Figure 5C**. We then defined the 'relative susceptibility' as the ratio $\mathrm{AUC_{w_{out}}}/\mathrm{AUC_{PC}}$, **Figure 5D**. For aligned networks (red), the relative susceptibility was close to 1 indicating similarly strong responses to both types of perturbations. For oblique networks (yellow), it was much smaller than 1, indicating that long-term responses to perturbations along the output direction were weaker than those to perturbations along the PCs.

## Noise suppression for oblique dynamics

In the oblique regime, the output weights are large. To produce the correct output (and not a too large one), the large PCs of the dynamics are almost orthogonal to the output weights. The large output weights, however, pose a robustness problem: Small noise in the direction of the output weights is also amplified at the level of the readout. We show that learning leads to a slow process of sculpting noise statistics to avoid this effect (Figure 11). Specifically, a negative feedback loop is generated that suppresses fluctuations along the output direction (**Figure 6A**, Figure 10; **Kadmon et al., 2020**). Because the positive feedback loop that gives rise to the large PCs is mostly orthogonal to the output direction, it remains unaffected by this additional negative feedback loop. A detailed analysis of how learning is affected by noise shows that, for large output weights, the network first learns a solution that is not robust to noise. This solution is then transformed to increasingly stable and oblique dynamics over longer time scales (section Learning with noise for linear RNNs and Oblique solutions arise for noisy, nonlinear systems).

To illustrate the effect of the negative feedback loop, we consider the fluctuations around trial averages. We take a collection of states $\mathbf{x}(t)$ and then subtract the task-conditioned averages $\bar{\mathbf{x}}(t)$ to compute $\delta\mathbf{x}(t) = \mathbf{x}(t) - \bar{\mathbf{x}}(t)$. We then project $\delta\mathbf{x}(t)$ onto three different direction categories: the largest PCs of the averaged data $\bar{\mathbf{x}}(t)$, the output directions, or randomly drawn directions.

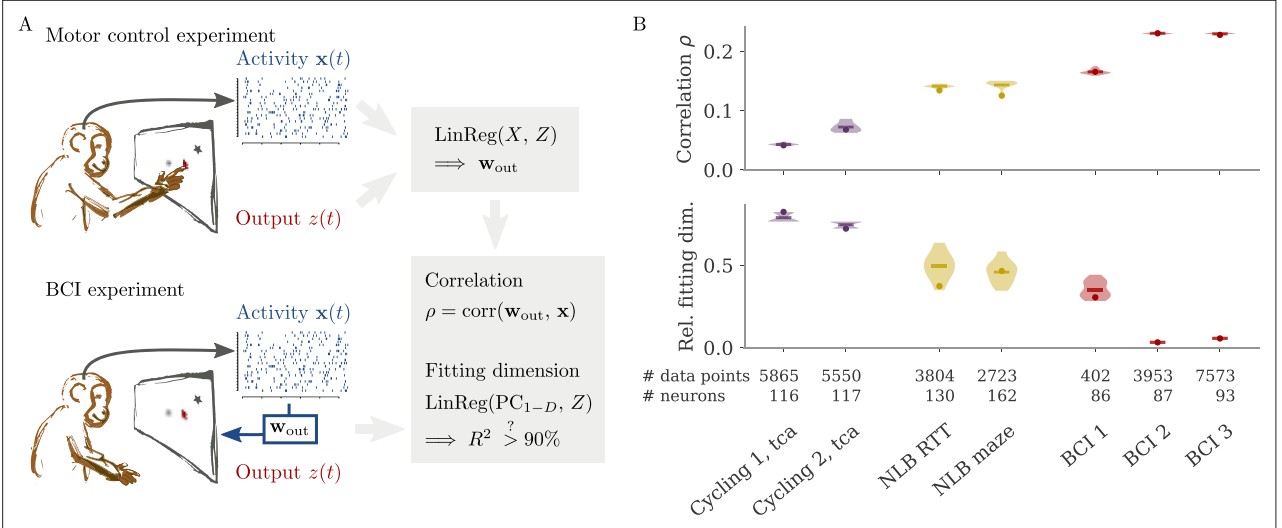

**Figure 7.** Quantifying aligned and oblique dynamics in experimental data (*Russo et al., 2018*; *Pei et al., 2021*; *Golub et al., 2018*; *Hennig et al., 2018*; *Degenhart et al., 2020*). (**A**) Diagram of the two types of experimental data considered. Here, we always took the velocity as the output (hand, finger, or cursor). In motor control experiments (top), we first needed to obtain the output weights $\mathbf{w}_{out}$ via linear regression. We then computed the correlation $\rho$ and the reconstruction dimension $D_{fit,90}$, i.e., the number of principal components (PCs) of $\mathbf{x}$ necessary to obtain a coefficient of determination $R^2 > 90\%$. In brain-computer interface (BCI) experiments (bottom), the output (cursor velocity) is generated from neural activity $\mathbf{x}(t)$ via output weights $\mathbf{w}_{out}$ defined by the experimenter. This allowed us to directly compute correlations and fitting dimensions. (**B**) Correlation $\rho$ (top) and relative fitting dimension $D_{fit,90}/D_{x,90}$ (bottom) for several publicly available data sets. The cycling task data (purple) were trial-conditioned averages, the BCI experiments (red) and Neural Latents Benchmark (NLB) tasks (yellow) single-trial data. Results for the full data sets are shown as dots. Violin plots indicate results for 20 random subsets of 25% of the data points in each data set (bars indicate mean). See small text below x-axis for the number of time points and neurons.

How strongly the activity fluctuates along each direction is quantified by the variance of the projections (*Figure 6B*). For both aligned and oblique dynamics, the variance is much larger along the PCs than along random directions. This is not necessarily expected, because the PCA was performed on the *averaged* activity, without the fluctuations. Instead, it is a dynamical effect: the same positive feedback that generates the autonomous dynamics also amplifies the noise (section Oblique solutions arise for noisy, nonlinear systems).

The two network regimes, however, dissociate when considering the variance along the output direction. For aligned dynamics, there is no negative feedback loop, and $\mathbf{w}_{out}$ is correlated with the PCs. The variance along the output direction is hence similar to that along the PCs, and larger than along random directions. For oblique dynamics, the negative feedback loop suppresses the fluctuations along the output direction, so that they become weaker than along random directions.

In *Figure 6C*, we quantify this dissociation across different tasks. We measured the ratio between variance along output and random directions. Aligned networks have a ratio much larger than one, indicating that the fluctuations along the output direction are increased due to the autonomous dynamics along the PCs. In contrast, oblique networks have a ratio smaller than 1 for all tasks, which indicates noise compression along the output.

## Different degrees of alignment in experimental settings

For the cycling task, we observed that dynamics were qualitatively different for the two regimes, with trajectories either counter- or co-rotating (*Figure 2B*). Interestingly, the experimental results of *Russo et al., 2018*, matched the oblique, but not the aligned dynamics. The authors observed co-rotating dynamics in the leading PCs of motor cortex activity despite counter-rotating activity of simultaneously recorded muscle activity. Here, we test whether our theory can help to more clearly and quantitatively distinguish between the two regimes in experimental settings.

In typical experimental settings, we do not have direct access to output weights. We can, however, approximate these by fitting neural data to simultaneously recorded behavioral output, such as hand velocity in a motor control experiment (*Figure 7A*, top). Following the model above, where the

output is a weighted average of the states, we reconstruct the output from the neural activity with linear regression. To quantify the dynamical regime, we then compute the correlation $\rho$ between the weights from fitting and the neural data. Additionally, we can also quantify the alignment by the 'relative fitting dimension' $D_{\text{fit},90}/D_{x,90}$, where $D_{\text{fit},90}$ is the number of PCs necessary to recover the output and $D_{x,90}$ number to the number of PCs necessary to represent 90% of the variance of the neural data. We computed both the correlation and the relative fitting dimension for different publicly available data sets (*Figure 7B*). For details, see section Experimental data.

We started with data sets from two monkeys performing the cycling task (*Russo et al., 2018*). The data contained motor cortex activity, hand movement, and EMG from the arms, all averaged over multiple trials of the same condition. In *Figure 7B*, we show results for reconstructing the hand velocity. The correlation was small, $\rho \in [0.04, 0.07]$. To obtain a good reconstruction, we needed a substantial fraction of the dimension of the neural data: The relative fitting dimension was $D_{\text{fit},90}/D_{x,90} \in [0.7, 0.8]$. Our results agree with previous studies, showing that the best decoding directions are only weakly correlated with the leading PCs of motor cortex activity (*Schroeder et al., 2022*).

We also analyzed data made available through the Neural Latents Benchmark (NLB) (*Pei et al., 2021*). In two different tasks, monkeys needed to perform movements along a screen. In a random target task, the monkeys had to point at a randomly generated target position on a screen, with a successive target point generated once the previous one was reached (*Makin et al., 2018*). In a maze task, the monkeys were trained to follow a trajectory through a maze with their hand (*Churchland et al., 2010*). In both cases, we reconstructed the finger or hand velocity from neural activity on single trials. The correlation was higher than in the cycling task, $\rho = 0.13$. The relative fitting dimension was lower than in the trial-averaged cycling data, albeit still on the same order: $D_{\text{fit},90}/D_{x,90} \in [0.4, 0.5]$.

Finally, we considered brain-computer interface (BCI) experiments (*Sadtler et al., 2014*). In these experiments, monkeys were trained to control a cursor on a screen via activity read out from their motor cortex (*Figure 7A*, bottom). The output weights generating the cursor velocity were set by the experimenter (so we don't need to fit). Importantly, the output weights were typically chosen to be spanned by the largest PCs (the 'neural manifold'), suggesting aligned dynamics. For three example data sets (*Golub et al., 2018*; *Hennig et al., 2018*; *Degenhart et al., 2020*), we obtained higher correlation values, $\rho \in [0.17, 0.23]$ (*Figure 7B*). The relative fitting dimension was much smaller than for the non-BCI data sets, especially for the two largest data sets, where $D_{\text{fit},90}/D_{x,90} \in [0.03, 0.06]$.

The higher correlation and much smaller relative fitting dimension suggest that, indeed, the neural dynamics arising in BCI experiments are more aligned, and those in non-BCI settings are more oblique. These trends also hold when decoding other behavioral outputs for the cycling task and the NLB tasks (position, acceleration, or EMG), even if the ability to decode and the numerical values for correlation and fitting dimension may fluctuate considerably (*Appendix 1—figure 8*). Thus, while we do not observe strongly different regimes as in the simulations, we do see an ordering between different data sets according to the alignment between outputs and neural dynamics. It would be interesting to test the differences between BCI and non-BCI data on larger data sets, and different experiments with different dimensions of neural data (*Gao et al., 2017*; *Stringer et al., 2019a*).

## Discussion

We analyzed the relationship between neural dynamics and behavior, asking to which extent a network's output is represented in its dynamics. We identified two different limiting regimes: aligned dynamics, in which the dominant activity in a network is related to its output, and oblique dynamics, where the output is only a small modulation on top of the dominating dynamics. We demonstrated that these two regimes have different functional implications. We also examined how they arise through learning, and how they relate to experimental findings.

Linking neural activity to external variables is one of the core challenges of neuroscience (*Hubel and Wiesel, 1962*). In most cases, however, such links are far from perfect. The activity of single neurons can be related in a nonlinear, mixed manner, to task variables (*Rigotti et al., 2013*). Even when considering populations of neurons, a large fraction of neural activity is not easily accounted for by external variables (*Arieli et al., 1996*). Various explanations have been proposed for this disconnect. In the visual cortex, activity has been shown to be related to 'irrelevant' external variables, such as body movements (*Stringer et al., 2019b*). Follow-up work showed that, in primates, some of these effects can be explained by the induced changes on retinal images (*Talluri et al.,*

*2022*), but this study still explained only half of the neural variability. An alternative explanation hinges on the redundancy of the neural code, which allows 'null spaces' in which activity can visit without affecting behavior (*Rokni et al., 2007*; *Kaufman et al., 2014*; *Kao et al., 2021*). Through the oblique regime, our study offers a simple explanation for this phenomenon: in the presence of large output weights, resistance to noise or perturbations requires large, potentially task-unrelated neural dynamics. Conversely, generating task-related output in the presence of large, task-unrelated dynamics requires large readout weights.

We showed theoretically and in simulations that, when training RNNs, the magnitude of output weights is a central parameter that controls which regime is reached. This finding is vital for the use of RNNs as hypothesis generators (*Sussillo, 2014*; *Barak, 2017*; *Vyas et al., 2020*), where it is often implicitly assumed that training results in universal solutions (*Maheswaranathan et al., 2019*) (even though biases in the distribution of solutions have been discussed; *Sussillo et al., 2015*). Here, we show that a specific control knob allows one to move between qualitatively different solutions of the same task, thereby expanding the control over the hypothesis space (*Turner et al., 2021*; *Pagan et al., 2022*). Note in particular that the default initialization in standard learning frameworks has large output weights, which results in oblique dynamics (or unstable solutions if training without noise, see Methods, section Analysis of solutions under noiseless conditions).

The role of the magnitude of output weights is also discussed in machine learning settings, where different learning regimes have been found (*Jacot et al., 2018*; *Chizat et al., 2019*; *Mei et al., 2018*; *Jacot et al., 2022*). In particular, 'lazy' solutions were observed for large output weights in feedforward networks. We show in Methods, section Analysis of solutions under noiseless conditions, that these are unstable for recurrent networks and are replaced in a second phase of learning by oblique solutions. This second, slower, phase is reminiscent of implicit regularization in overparameterized networks (*Ratzon et al., 2024*; *Blanc et al., 2020*; *Li et al., 2021*; *Yang et al., 2023*). On a broader scale, which learning regime is relevant when modeling biological learning is an open question that has only just begun to be explored (*Flesch et al., 2022*; *Liu et al., 2023*).

The particular control knob we studied has an analog in the biological circuit – the synaptic weights. We can thus use experimental data to study whether the brain might rely on oblique or aligned dynamics. Existing experimental work has partially addressed this question. In particular, the work by *Russo et al., 2018*, has been a major inspiration for our study. Our results share some of the key findings from that paper – the importance of stability leading to 'untangled' dynamics (*Susman et al., 2021*) and a dissociation between hidden dynamics and output. In addition, we suggest a specific mechanism to reach oblique dynamics – training networks with large output weights. Furthermore, we characterize the aligned and oblique regimes along experimentally accessible axes.

We see three avenues for exploring our results experimentally. First, simultaneous measurements of neural dynamics and muscle activity could be used to quantify noise along the output direction. This would allow checking whether noise is compressed in this direction, and in particular, whether such compression occurs on a slow time scale after initial task acquisition. We suggest how to test this in *Figure 6C*. Second, we show how the dynamical regimes dissociate under perturbations along specific directions. Experiments along these lines have recently become possible (*Russell et al., 2022*; *Finkelstein et al., 2021*; *Chettih and Harvey, 2019*). Future work is left to combine our model with biological constraints that induce additional effects during perturbations, e.g., through non-normal synaptic connectivity (*O'Shea et al., 2022*; *Kim et al., 2023*; *Bondanelli and Ostojic, 2020*; *Logiaco et al., 2021*). Third, our work connects to the setting of BCI, where the experimenter chooses the output weights at the beginning of learning (*Sadtler et al., 2014*; *Golub et al., 2018*; *Willett et al., 2021*; *Rajeswaran et al., 2024*). Typically, the output weights are set to lie 'within the manifold' of the leading PCs so that we expect aligned dynamics (*Sadtler et al., 2014*). In experiments where the output weights were rotated out of the manifold (without changing the norm), learning took longer and led to a rotation of the manifold, i.e., at least a partial alignment (*Oby et al., 2019*). Our theory suggests directly comparing the degree of alignment between dynamics obtained from within- and out-of-manifold initializations. Furthermore, it would be interesting to systematically change the norm of the output weights (in particular for out-of-manifold initializations) to see whether larger output weights lead to more oblique solutions. If this is the case, we suggest testing whether such more oblique solutions meet our predictions, e.g., higher variability between individuals and noise suppression.

Overall, our results provide an explanation for the plethora of relationships between neural activity and external variables. It will be interesting to see whether future studies will find hallmarks of either regime for different experiments, tasks, or brain regions.

## Methods
### Details on RNN models and training

We consider rate-based RNNs with $N$ neurons. The states $\mathbf{x}(t) \in \mathbb{R}^N$ are governed by

$$\dot{\mathbf{x}} = -\mathbf{x} + W\phi(\mathbf{x}) + W_{\text{in}}\mathbf{s}(t) + \boldsymbol{\xi}(t), \tag{4}$$

where $W$ is a recurrent weight matrix and $\phi = \tanh$ a nonlinearity applied element-wise. The network receives a low-dimensional input $\mathbf{s}(t) \in \mathbb{R}^{N_{\text{in}}}$ via input weights $W_{\text{in}}$. It is also driven by white, isotropic noise with zero mean and covariance $\mathbb{E}[\xi_i(t)\xi_j(t')] = 2\sigma_{\text{noise}}^2\delta_{ij}\delta(t - t')$. The initial states $\mathbf{x}(0)$ are drawn from a centered normal distribution with variance $\sigma_{\text{init}}^2$ at each trial. This serves as additional noise. The output is a low-dimensional, linear projection of the states: $\mathbf{z}(t) = W_{\text{out}}\mathbf{x}(t)$ with $W_{\text{out}} = [\mathbf{w}_{\text{out},1}, \ldots, \mathbf{w}_{\text{out},N_{\text{out}}}]^T$.

The initial output weights are drawn from centered normal distributions with variance $\sigma_{\text{out}}^2/N$. 'Small' output weights refer to $\sigma_{\text{out}} = 1/\sqrt{N}$, and 'large' ones to $\sigma_{\text{out}} = 1$. We have $\|\mathbf{w}_{\text{out},i}\| = \sigma_{\text{out}}[1 + O(1/\sqrt{N})]$ at initialization. Note that large initial output weights are the current default in standard learning environments (*Paszke et al., 2017*; *Yang and Hu, 2020*). The recurrent weights were initialized from centered normal distributions with variance $g^2/N$. We chose $g = 1.5$ so that dynamics were chaotic before learning (*Sompolinsky et al., 1988*).

To simulate the noisy RNN dynamics numerically, we used the Euler-Maruyama method (*Kloeden and Platen, 1992*) with a time step of $\Delta t$. We used the Adam algorithm (*Kingma and Ba, 2014*) implemented in PyTorch (*Paszke et al., 2017*). Apart from the learning rate, we kept the parameters for Adam at the default (some filtering, no weight decay). We selected learning rates and the number of training steps such that learning was relatively smooth and converged sufficiently within the given

**Table 1.** Task, simulation, and network parameters for *Figures 3–6*.

| Parameter | Symbol | Cycling | Flip-flop | Mante | Romo | Complex sine |
|---|---|---|---|---|---|---|
| # inputs | $N_{\text{in}}$ | 2 | 3 | 4 | 1 | 1 |
| # outputs | $N_{\text{out}}$ | 2 | 3 | 1 | 1 | 1 |
| Trial duration | $T$ | 72 | 25 | 48 | 29 | 50 |
| Fixation duration | $t_{\text{fix}}$ | 0 | $\mathcal{U}(0,1)$ | 3 | $\mathcal{U}(1.3)$ | 0 |
| Stimulus duration | $t_{\text{stim}}$ | 1 | 1 | 20 | 1 | 50 |
| Stimulus delay | $t_{\text{sd}}$ | – | $\mathcal{U}(3,10)$ | – | $\mathcal{U}(2,12)$ | – |
| Decision delay | $t_{\text{delay}}$ | 1 | 2 | 5 | 4 | 0 |
| Decision duration | $t_{\text{dec}}$ | 71 | $t_{\text{sd}}$ | 20 | 8 | 50 |
| Simulation time step | $\Delta t$ | – 0.2 – | | | | |
| Target time step | $\Delta t_{\text{target}}$ | 1.0 | 1.0 | 1.0 | 1.0 | 0.2 |
| Activation noise | $\sqrt{2}\sigma_{\text{noise}}$ | 0.2 | 0.2 | 0.05 | 0.2 | 0.2 |
| Initial state noise | $\sigma_{\text{init}}$ | – 1.0 – | | | | |
| Network size | $N$ | – 512 – | | | | |
| # training epochs | | 1000 | 4000 | 4000 | 6000 | 6000 |
| Learning rate aligned | $\eta_0$ | 0.02 | 0.005 | 0.002 | 0.005 | 0.005 |
| Learning rate oblique | $\eta_0$ | 0.02 | 0.01 | 0.02 | 0.01 | 0.005 |
| Batch size | | – 32 – | | | | |

number of trials. Learning rates were set to $\eta = \eta_0/N$. Details for all simulation parameters can be found in **Table 1**.

For the comparisons over different tasks (**Figures 3–6**), we trained five networks for each task. All weights ($W_{\text{out}}, W, W_{\text{in}}$) were adapted. For the example networks trained on the cycling task (**Figures 2, 5, and 6**), we used networks with $N = 256$ neurons and only changed the recurrent weights $W$. We also trained for longer (5000 training steps) and with a higher learning rate ($\eta = 0.1/N$).

## Task details

The tasks the networks were trained on are taken from the neuroscience literature: a cycling task (**Russo et al., 2018**), a 3-bit flip-flop task, and a 'complex sine' task (with input-dependent frequencies) (**Sussillo and Barak, 2013**), a context-dependent decision-making task ('Mante') (**Mante et al., 2013**; **Schuessler et al., 2020a**), and a working memory task comparing the amplitudes of two pulse stimuli ('Romo') (**Romo et al., 1999**; **Schuessler et al., 2020a**). All tasks have similar structure (**Schuessler et al., 2020b**): A trial of length $T$ starts with a fixation period (length $t_{\text{fix}}$). This is followed by an input for $t_{\text{stim}}$. For the cycling and flip-flop task, the inputs are pulses of amplitude 1; else see below. After a delay $t_{\text{delay}}$, the output of the network is required to reach an input-dependent value during a time period $t_{\text{dec}}$. During this decision period, we set target points $t_i$ every $\Delta t_{\text{target}}$ time steps. The loss was defined as the mean squared error between network output and target at these time points. Below, we provide further details for each task.

### Cycling task

The network receives an initial pulse, whose direction ($[1, 0]^T$ or $[0, 1]^T$) determines the sense of direction of the target. The target is given by a rotation in 2D, $\hat{\mathbf{z}}(t) = [\sin(a2\pi ft), \cos(2\pi ft)]^T$, with frequency $f = 0.1$ and $a = \pm 1$ for the two directions (clockwise or anticlockwise).

### Flip-flop task

The network repeatedly receives input pulses along one of three directions, followed by decision periods. In each decision period, the output coordinate corresponding to the last input should reach ±1 depending on the sign of the input. All other coordinates should remain at ±1 as defined by the last time they were triggered. To make sure that this is well defined, we trigger all inputs at time steps $k\Delta t$ for $k \in [N_{\text{in}}]$ with random signs.

### Mante task

Input channels for this task are split into two groups of size $N_{\text{in}}/2$: half of the channels for the signal and the other half for the context, indicating which of the signal channels is relevant. All signal channels $s_i(t)$ deliver a constant mean $\hat{s}_i$ plus additional white noise: $s_i(t) = \hat{s}_i + a_{\text{noise}}\eta_i(t)$. The mean is drawn uniformly from $\{\pm 1, \pm\frac{1}{2}, \pm\frac{1}{4}, \pm\frac{1}{8}\}$, and the noise amplitude is $a_{\text{noise}} = 0.05$. For simulations, we draw a standard normal variable $n_{i,k} \sim \mathcal{N}(0, 1)$ at time step $k$, and set $\eta_{i,k} = n_{i,k}/\sqrt{\Delta t}$. Only a single contextual input is active at each trial, $s_{i+N_{\text{in}}/2} = \delta_{ij}$, with $j$ chosen uniformly from the number of context $N_{\text{in}}/2$. The target during the decision period is the sign of the relevant input, $\hat{z}(t) = \text{sign}(\hat{s}_j)$.

### Romo task

For the Romo task, the input consists of two input pulses separated by random delays $t_{\text{sd}}$. The amplitude of the inputs is drawn independently from $\mathcal{U}(0.5, 1.5)$ with the condition of being at least 0.2 apart (else both are redrawn). During the decision period, the network needs to yield $\hat{z}(t) = \pm 1$, depending on which of the two pulses was larger.

### Complex sine

The target is $\hat{z}(t) = \sin(2\pi ft)$, with frequency $f = (1 - a)f_{\text{min}} + af_{\text{max}}$, and boundaries $f_{\text{min}} = 0.04$, $f_{\text{max}} = 0.2$, and where $a \sim \mathcal{U}(0, 1)$. The input is a constant input of amplitude $s = a + 0.25$.

## Generalized correlation

For **Figure 3A**, we used a generalized correlation measure which allows for multiple output dimensions, multiple time points, and noisy data. Consider neural activity of $N$ neurons at $P$ time points

stacked into the matrix $X = [\mathbf{x}(1), \dots, \mathbf{x}(P)] \in \mathbb{R}^{N \times P}$. We assume the states to be centered in time, $\frac{1}{P} \sum_{t=1}^{P} x_i(t) = 0$ for $i \in [N]$. The corresponding $D$-dimensional output is summarized in the $D \times P$ matrix

$$Z = W_{\text{out}}^T X, \tag{5}$$

with weights $W_{\text{out}} \in \mathbb{R}^{N \times D}$. We define the generalized correlation as

$$\rho = \frac{\|W_{\text{out}}^T X\|}{\|W_{\text{out}}\| \, \|X\|}. \tag{6}$$

The norm is the Frobenius norm, $\|X\| = \sqrt{\sum_{ij} X_{ij}^2}$. In particular, we have

$$\|Z\| = \rho \, \|W_{\text{out}}\| \, \|X\|. \tag{7}$$

The case of 1D output and a single time step discussed in the main text, *Equation 3*, is recovered up to the sign, which we discard. Note that in that case, the vectors $\mathbf{w}_{\text{out}}$ and $\mathbf{x}(t)$ should be centered along coordinates to receive a valid correlation. Our numerical results did not change qualitatively when centering across coordinates only or both coordinates and time.

For trajectories with multiple conditions, we stack these instances in a matrix $X \in \mathbb{R}^{N \times N_c N_t}$, with $N_c$ the number of conditions, and $N_t$ the number of time points per trajectory. For noisy trajectories, we first average over multiple instances per condition and time point to obtain a similar matrix $\bar{X}$.

## Regression

In *Figure 3B and C* we computed the number of PCs necessary to either represent the dynamics or fit the output. We simulated the trained networks again on their corresponding tasks. We did not apply noise during these simulations, since keeping the same noise as during training would reduce the quality of the output for large output weights; trial averaging yielded similar results to the ones obtained without noise (not shown).

The simulations yielded neural states $X \in \mathbb{R}^{N \times P}$ and outputs $Z \in \mathbb{R}^{N_{\text{out}}} \times P$, where $P$ is the number of data points (batch size times number of time points $T$). We applied PCA to the states $X$. The cumulative explained variance ratio obtained from PCA is plotted in *Figure 3B*. We then projected $X$ onto the first $k$ PCs and fitted these projections to the output with ridge regression (cross-validated, using scikit-learn's RidgeCV *Pedregosa et al., 2011*).

## Dissimilarity measure

For measuring the dissimilarity between learners in *Figure 4*, we apply a measure following *Williams et al., 2021*. We define the distance between two sets with $P$ data points $X, Y \in \mathbb{R}^{N \times P}$ as

$$d(X, Y) = \underset{Q \in \mathcal{O}}{\arg\min} \arccos \frac{\text{Tr}(\hat{X}\hat{Y}^T Q)}{\|\hat{X}\| \, \|\hat{Y}\|}, \tag{8}$$

where the hat corresponds to centering along the rows, and $Q$ is an orthogonal matrix. The solution to this so-called orthogonal Procrustes' problem is found via the singular value decomposition $\hat{X}\hat{Y}^T = U\Sigma V^T$. The optimal transformation is $Q^* = VU^T$, and the numerator in *Equation 8* is then $\text{Tr}(\hat{X}\hat{Y}^T Q^*) = \text{Tr}(\Sigma)$.

Note that this is more restricted than canonical correlation analysis (CCA), which is also commonly used (*Gallego et al., 2018*; *Gallego et al., 2020*). In particular, CCA involves whitening the matrices $X$ and $Y$ before applying a rotation (*Williams et al., 2021*). This sets all singular values to 1. For originally low-D data, this mostly means amplifying the noise, unless the data was previously projected onto a small number of PCs. In the latter case, the procedure still removes the information about how much each PC contributes.

## Experimental data

We detail the analyses of neural data in section Different degrees of alignment in experimental settings. We made use of publicly available data sets: data from the cycling task of *Russo et al., 2018*, two data sets available through the NLB (*Pei et al., 2021*), and data from monkeys trained on a

center-out reaching task with a BCI (*Golub et al., 2018*; *Hennig et al., 2018*; *Degenhart et al., 2020*). For all data sets, we first obtain firing rates $X \in \mathbb{R}^{N \times T}$, where $N$ is the number of measured neurons, and $T$ the number of data points, (see *Figure 7B* for these numbers). We also collect the simultaneously measured behavior in the matrix $Z \in \mathbb{R}^{D \times T}$. In *Figure 7*, we only analyzed cursor or hand velocity for behavior, so that the output dimension is $D = 2$. See *Appendix 1—figure 8* for similar results for hand position and acceleration or the largest two PCs of the EMG data recorded for the cycling task.

For the cycling task, the firing rates were binned in 1 ms bins and convolved with a 25 ms Gaussian filter. The mean firing rate was 22 and 18 Hz for the two monkeys, respectively. For the NLB data, spikes came in 1 ms bins. We binned data to 45 ms bins and applied a Gaussian filter with 45 ms width. This increased the quality of the fit, as firing rates were much lower (mean of 5 Hz for both) than in the cycling data set. For the BCI experiments, firing rates came as spike counts in 45 ms bins. The mean firing rate was (45, 45, 55) Hz for the data of *Golub et al., 2018*; *Hennig et al., 2018*; *Degenhart et al., 2020*, respectively. In agreement with the original BCI experiments, we did not apply a filter to the neural data.

For fitting, we centered both firing rates $X$ and output $Z$ across time (but not coordinates). We also added a delay of 100 ms between firing rates and output for the cycling and NLB data sets, which increased the quality of the fits. We then fitted the output $Z$ to the firing rates $X$ with ridge regression, with regularization obtained from cross-validation. We treated the coefficients as output weights $W_{\text{out}}$. The trial average data of the cycling tasks was very well fitted for both monkeys, $R^2 = [0.97, 0.98]$. For the NLB tasks with single-trial data, the fits were not as good, $R^2 = [0.73, 0.69]$. For two of the BCI data sets (*Golub et al., 2018*; *Hennig et al., 2018*), the output weights were also given, and we checked that the fit recovers these. For the third BCI data set (*Degenhart et al., 2020*), we did not have access to the output weights, and only access to the cursor velocity after Kalman filtering. Here, fitting yielded $R^2 = 0.83$.

For the fitting dimension $D_{\text{fit},90}/D_{x,90}$ in *Figure 7B*, bottom, we used an adapted definition of $D_{\text{fit},90}$: Because $R^2 = 90\%$ is not reached for all data sets, we asked for the number of PCs necessary to obtain 90% of the $R^2$ value obtained for the full data set.

We also considered whether the correlation $\rho$ scales with the number of neurons $N$. In our model, oblique and aligned dynamics can be defined in terms of such a scaling: aligned dynamics have highly correlated output weights and low-dimensional dynamics, so that $\rho \sim N^0 = 1$, i.e., independent of the network size. For oblique dynamics, large output weights with norm $\mathbf{w}_{\text{out}} \sim 1$ lead to vanishing correlation, $\rho \sim 1/\sqrt{N}$. This is indeed similar to the relation between two random vectors, for which the correlation is precisely $1/\sqrt{N}$ (in the limit of large $N$). In *Figure 8*, we show the scaling of $R^2$ and $\rho$ with the number of subsampled neurons. For the cycling task and NLB data, the correlation scaled slightly weaker than $\rho \sim N^{-1/2}$. For the BCI data, the scaling was closer to $\rho \sim N^{-1/4}$ which is in between the aligned and oblique regimes of the model. These insights, however, are limited due to the trial averaging for the cycling task and the limited number of time points for the NLB tasks (not enough to reach $R^2 = 1$). Applying these measures to larger data sets could yield more definitive insights.

## Analysis of solutions under noiseless conditions

In the sections below, we explore in detail under which conditions aligned and oblique solutions arise, and which other solutions arise if these conditions are not met.

We first consider small output weights and show that these lead to aligned solutions. Then, for large output weights, we show that without noise, two different, unstable solutions arise. Finally, we consider how adding noise affects learning dynamics. For a linear model, we can solve the dynamics of learning analytically and show how a negative feedback loop arises, that suppresses noise along the output direction. However, the linear model does not yield an oblique solution, so we also consider a nonlinear model for which we show in detail why oblique solutions arise.

We start by analyzing a simplified version of the network dynamics (*Equation 4*): autonomous dynamics without noise,

$$\dot{\mathbf{x}} = -\mathbf{x} + W\phi(\mathbf{x}), \tag{9}$$

with fixed initial condition $\mathbf{x}(0)$. We assume a 1D output $z(t)$ and a target $\hat{z}(t_i)$ defined on a finite set of time points $t_i$.

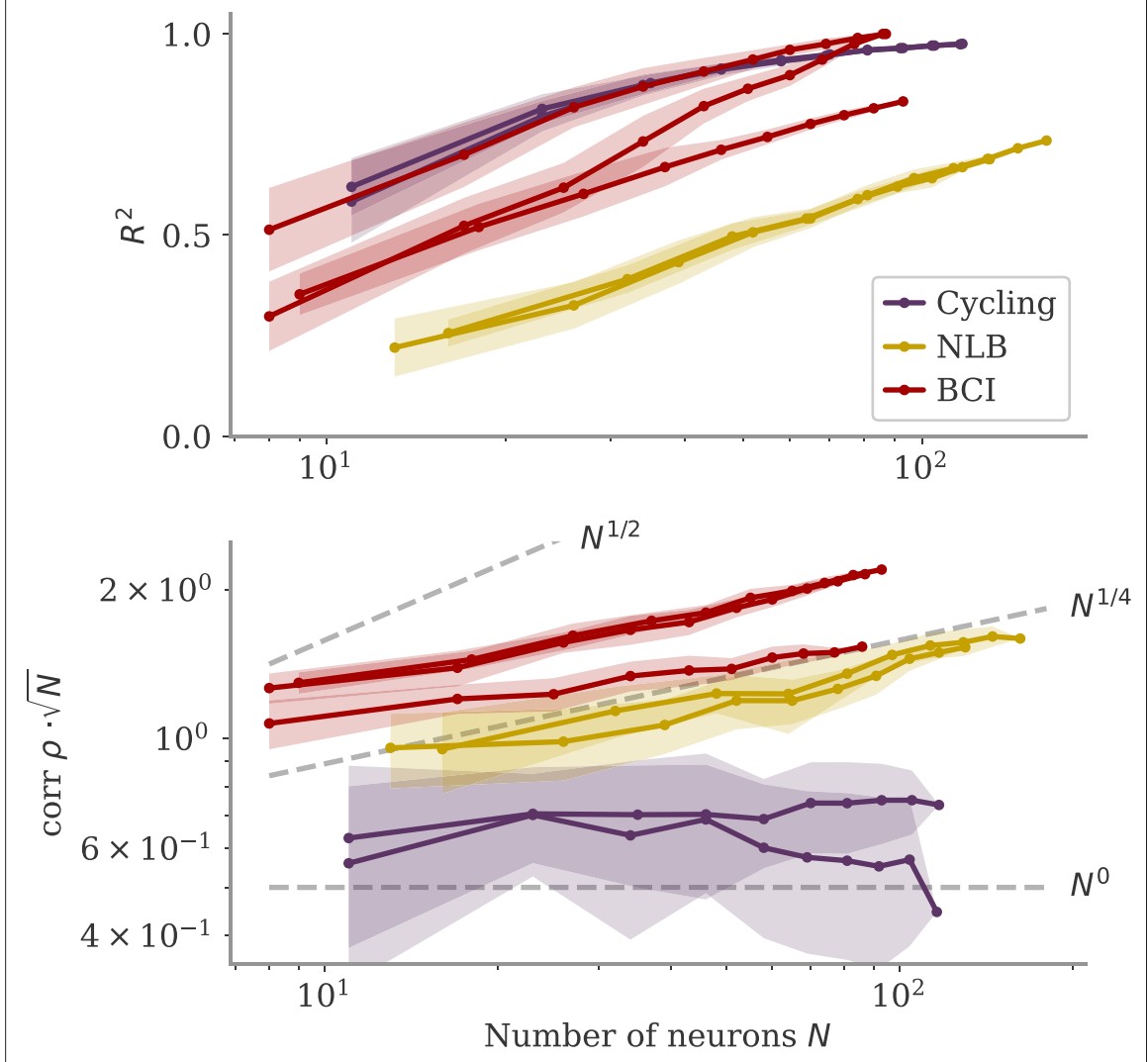

**Figure 8.** Correlation scaling with number of neurons. Scaling of the correlation $\rho$ with the number of neurons $N$ in experimental data. We fitted the output weights to subsets of $N$ neurons and computed the quality of fit (top) and the correlation between the resulting output weight and firing rates (bottom). To compare with random vectors, the correlation is scaled by $\sqrt{N}$. Dashed lines are $N^p/2$, for $p \in \{1/2, 1/4, 0\}$ for comparison. The aligned regime corresponds to $p = 1/2$, and the oblique one to $p = 0$.

We illustrate the theory with an example of a simple sine wave task (**Figure 9**). We demand the network to autonomously produce a sine wave with fixed frequency $f = 0.1$. At the beginning of the task, the network receives an input pulse that sets the starting point of the trajectory. We set the noise $\sigma_{\text{init}}$ on the initial state $\mathbf{x}(0)$ to zero. We define the target as 20 target points in the interval $t \in [1, 21]$ (two cycles; purple dots in **Figure 9**).

### Small weights lead to aligned solutions

For small output weights, $\|\mathbf{w}_{\text{out}}\| = 1/\sqrt{N}$, gradient-based learning in such a noise-less system has been analyzed by **Schuessler et al., 2020b**. Learning changes the dynamics qualitatively through low-rank weight changes $\Delta W$. These weight changes are spanned by existing directions such as the output weights. The resulting dynamics $\mathbf{x}(t)$ are thus aligned to the output weights. This means that the correlation between the two is large, independent of the network size, $\rho = O(1)$. The target of the

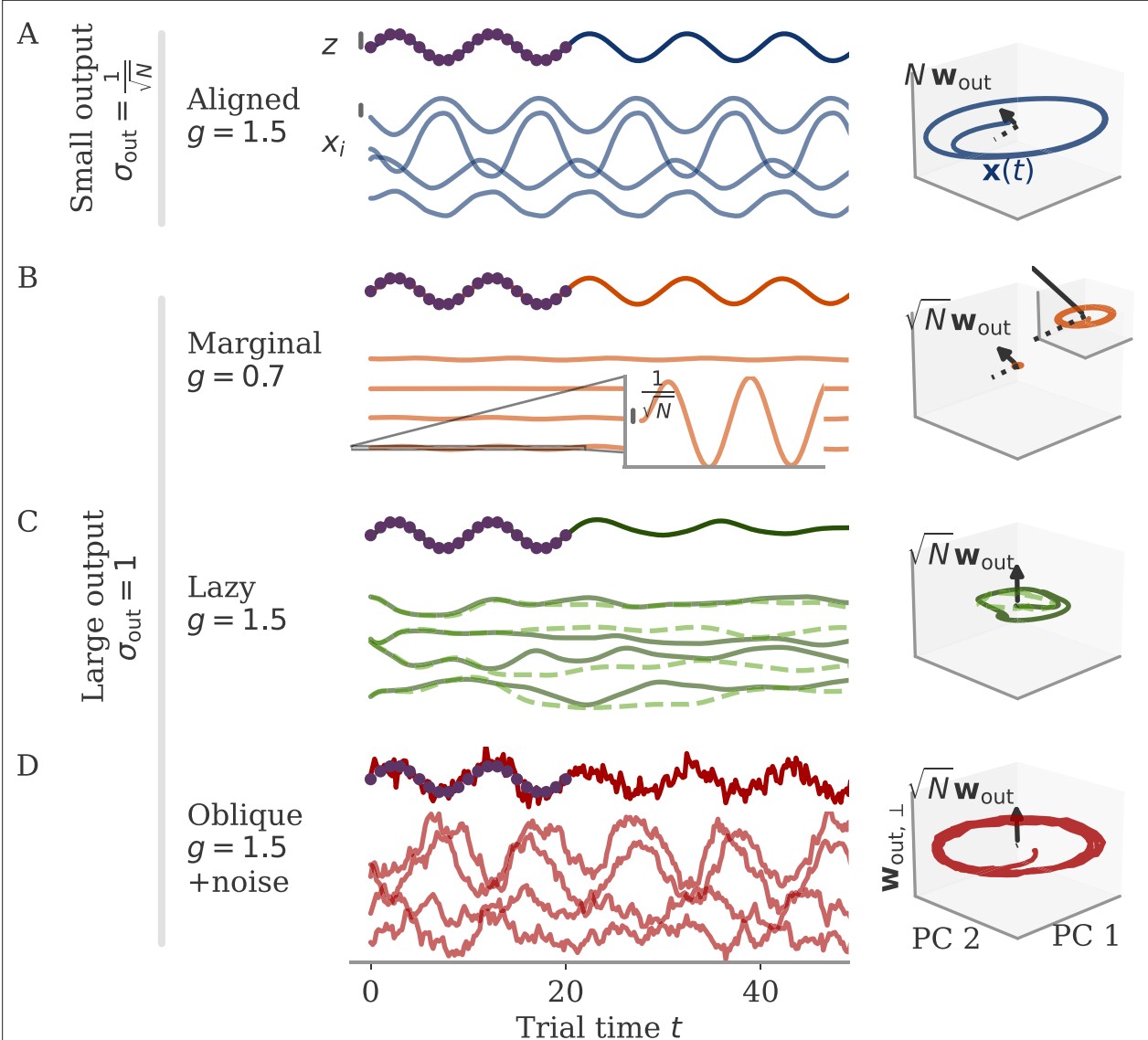

**Figure 9.** Different solutions for networks trained on a sine wave task. All networks have $N = 512$ neurons. Four regimes: (**A**) aligned for small output weights, (**B**) marginal for large output weights, small recurrent weights, (**C**) lazy for both large output and recurrent weights, (**D**) oblique for large output weights and noise added during training. Left: Output (dark), target (purple dots), and four states (light) of the network after training. Black bars indicate the scales for output and states (length = 1; same for all regimes). The output beyond the target interval $t \in [1, 21]$ can be considered as extrapolation. The network in the oblique regime, (**D**), receives white noise during training, and the evaluation is shown with the same noise. Without noise, this network still produces a sine wave (not shown). Right: Projection of states on the first two principal components (PCs) and the orthogonal component $\mathbf{w}_{\text{out},\perp}$ of the output vector. All axes have the same scale, which allows for comparison between the dynamics. Vectors show the (amplified) output weights, dotted lines the projection on the PCs (not visible for lazy and oblique). The insets for the marginal solution (B, left and right) show the dynamics magnified by $\sqrt{N}$.

task is also independent of $N$, so that after learning we have $z = O(1)$. Given the small output weights, we can thus infer the size of the states:

$$\underbrace{z}_{O(1)} = \underbrace{\rho}_{O(1)} \; \underbrace{\|\mathbf{w}_{\text{out}}\|}_{O(1/\sqrt{N})} \; \underbrace{\|\mathbf{x}\|}_{O(\sqrt{N})} \,, \tag{10}$$

so that $\|\mathbf{x}\| = O(\sqrt{N})$, or equivalently single neuron activations $x_i = O(1)$. This scaling corresponds to the aligned regime.

For the sine wave task, training with small output weights converges to an intuitive solution (**Figure 9A**). Neural activity evolves on a limit cycle, and the output is a sine wave that extrapolates

beyond the training interval. Plotting activity and output weights along the largest two PCs and the remaining direction $\mathbf{w}_{\mathrm{out},\perp}$ confirms substantial correlation, $\rho = O(1)$, as expected. The solution was robust to adding noise after training (not shown). Changes in the initial dynamics or the presence of noise during training did not lead to qualitatively different solutions (*Appendix 1—figure 10*). A further look at the eigenvalue spectrum of the trained recurrent weights revealed a pair of complex conjugate outliers corresponding to the limit cycle (*Mastrogiuseppe and Ostojic, 2018*; *Schuessler et al., 2020a*), and a bulk of remaining eigenvalues concentrated on a disk with radius $g$, *Appendix 1—figure 9*.

## Large weights, no noise: linearization of dynamics

We now consider learning with large output weights, $\|\mathbf{w}_{\mathrm{out}}\| = 1$, for noise-less dynamics, *Equation 9*. We start with the assumption that the activity changes for each neuron are small, $\Delta x_i(t) = O(1/\sqrt{N})$, or $\|\Delta \mathbf{x}\| = O(1)$. Here, $\Delta \mathbf{x}(t) = \mathbf{x}(t) - \mathbf{x}_0(t)$, where $\mathbf{x}_0(t)$ is the activity before learning. To perform a task, learning needs to induce output changes $\Delta z = O(1)$ to reach the target $\hat{z}(t)$. Note that a possible order-one initial output $z_0$ must also be compensated. Together with the output weight scale, we arrive at

$$\underbrace{\Delta z}_{O(1)} = \underbrace{\rho_{\Delta}}_{O(1)} \underbrace{\|\mathbf{w}_{\mathrm{out}}\|}_{O(1)} \underbrace{\|\Delta \mathbf{x}\|}_{O(1)}, \tag{11}$$

where $\rho_{\Delta} = \mathrm{corr}(\mathbf{w}_{\mathrm{out}}, \Delta \mathbf{x})$. This shows that our assumption of small state changes $\Delta \mathbf{x}$ is consistent – it allows for a solution – and that such small changes need to be strongly correlated to the output weights. Note that we make the distinction between the changes $\Delta \mathbf{x}$ and the final activity $\mathbf{x} = \mathbf{x}_0 + \Delta \mathbf{x}$, because the latter may be dominated by $\mathbf{x}_0$. In the main text, we only consider the correlation between $\mathbf{x}$ and $\mathbf{w}_{\mathrm{out}}$, because (as we show below) solutions with small $\Delta \mathbf{x}$ are not robust, and the final $\mathbf{x}$ will be dominated by $\Delta \mathbf{x}$.

For now, however, we ignore robustness and continue with the assumption of small $\Delta \mathbf{x}$. Given this assumption, we linearize the dynamics around the initial trajectory $\mathbf{x}_0(t)$:

$$\frac{\mathrm{d}\Delta \mathbf{x}}{\mathrm{d}t} = \underbrace{\Delta W \phi(\mathbf{x}_0)}_{\mathbf{a}} + \underbrace{[-I + (W_0 + \Delta W)R'(\mathbf{x}_0)]\Delta \mathbf{x}}_{\mathbf{b}} + O(\Delta \mathbf{x}^2), \tag{12}$$

with diagonal matrix $R'(\mathbf{x})_{ij} = \delta_{ij}\phi'(x_i)$ and the weights changes $\Delta W = W - W_0$ that induce $\Delta \mathbf{x}$. Note that we haven't yet constrained the weight changes $\Delta W$ so we cannot discard the terms of the kind $\Delta W \Delta \mathbf{x}$. The next steps depend on the initial trajectories $\mathbf{x}_0(t)$.

## Initially decaying dynamics lead to a marginal regime

We first consider networks with decaying dynamics before learning. This is obtained by drawing the initial recurrent weights independently from $W_{ij} \sim \mathcal{N}(0, \frac{g^2}{N})$, with $g < 1$ (*Sompolinsky et al., 1988*). With such dynamics, $\mathbf{x}_0(t)$ vanishes exponentially in time. In *Equation 12*, we disregard the term $\mathbf{a}$ and have $R' = I$, so that

$$\frac{\mathrm{d}\Delta \mathbf{x}}{\mathrm{d}t} = (-I + W_0 + \Delta W)\Delta \mathbf{x} + O(\Delta \mathbf{x}^2). \tag{13}$$

To have self-sustained dynamics, the matrix $W_0 + \Delta W$ must have a leading eigenvalue $\lambda_+$ with real part above the stability line: $\Re\lambda_+ = 1 + \epsilon$.

The distance $\epsilon > 0$ must be small, else the states would become large. To understand how $\epsilon$ needs to scale with $N$, we turn to a simple model studied before (*Mastrogiuseppe and Ostojic, 2018*; *Schuessler et al., 2020a*): an autonomously generated fixed point and rank-one connectivity $W = \frac{1}{N}\lambda_+ \mathbf{u}\mathbf{u}^T$. The vector $\mathbf{u}$ has entries $u_i \sim \mathcal{N}(0, 1)$. A fixed point of the dynamics (*Equation 9*) fulfills $\mathbf{x} = \frac{1}{N}\lambda_+ \mathbf{u}\mathbf{u}^T \phi(\mathbf{x})$. Projecting on $\mathbf{u}$ and applying partial integration in the limit $N \to \infty$, we obtain

$$\lambda_+ = \frac{1}{\langle \phi' \rangle}, \tag{14}$$

where $\langle\phi'\rangle = \int \mathcal{D}u \, \phi'(\sigma_x u)$, with standard normal measure $\mathcal{D}u = du \frac{1}{\sqrt{2\pi}} e^{-u^2/2}$. Here, $\sigma_x$ is the scale of the states, $\sigma_x = \|\mathbf{x}\|/\sqrt{N}$ or $x_i = O(\sigma_x)$. The fixed point is situated along the vector $\mathbf{u}$. To have the smallest possible fixed point generate some output, we set the output weights to $\mathbf{w}_{\text{out}} = \frac{1}{\sqrt{N}}\mathbf{u}$. Then, we have correlation $\rho = 1$ and $z = \mathbf{w}_{\text{out}}^T\mathbf{x} = \rho \|\mathbf{w}_{\text{out}}\| \|\mathbf{x}\| = 1 \cdot 1 \cdot \sqrt{N}\sigma_x$. In other words, a small fixed point with $\sigma_x \sim \frac{1}{\sqrt{N}}$. We expand $\phi'$ in *Equation 14* around zero. For even $\phi$, we have $\phi''(0) = 0$ and

$$\lambda_+ = \frac{1}{1 + \frac{1}{2}\phi'''(0)\sigma_x^2} = 1 - \frac{1}{2}\phi'''(0)\sigma_x^2 \,. \tag{15}$$

Sigmoidal functions have $\phi''' < 0$, e.g., $\phi'''(0) = -2$ for $\phi = \tanh$. Hence $\lambda = 1 + \epsilon$, with $\epsilon = \sigma_x^2 \sim \frac{1}{N}$. Remarkably, the perturbation leading to states with $\sigma_x = O(1/\sqrt{N})$ only needs to have a distance $\epsilon = O(1/N)$ away from the stability line.

The insights from this simplified setting extend to the example of the sine wave task (*Figure 9B*). The model with large output weights, $g = 0.7$, and no noise yields a limit cycle. The output extrapolates in time, but the states are very small, scaling as $x_i = O(1/\sqrt{N})$ (analysis over different $N$ not shown). Such a solution is only marginally stable – adding a white noise with $\sigma_{\text{noise}} = 0.2$ after training destroyed the rotation (not shown). The eigenvalues were again split into two outliers and a bulk (*Appendix 1—figure 9*). However, the two outliers now had a real part $1 + \epsilon$, i.e., they were very close to the stability line of the fixed point at zero. To better illustrate the marginal solution, we also set the initial state $\mathbf{x}(t = 0)$ to small values, $x_i(t = 0) \sim \mathcal{N}(0, 1/N)$. For $x_i(t = 0) \sim \mathcal{N}(0, 1)$, there would be an initial decay much larger than the limit cycle.

## Initially chaotic dynamics lead to a lazy regime

In contrast to the situation before, initially chaotic dynamics (for $g > 1$) imply order-one initial states, $(x_0(t))_i = O(1)$ for all trial times $t$. The driving term $\mathbf{a}$ in *Equation 12* can thus not be ignored and we expect it to be on the same scale as $\Delta\mathbf{x}$:

$$1 \sim \|\Delta\mathbf{x}\| \sim \|\Delta W\phi(\mathbf{x}_0)\| \,. \tag{16}$$

The smallest possible weight changes $\Delta W$ will be those for which $\phi(\mathbf{x}_0)$ yields a maximal response, but other vectors do not yield a strong response. This is captured by the operator norm, $\|\Delta W\|_2 = \max\{\|\Delta W\mathbf{x}\| : \mathbf{x} \in \mathbb{R}^N \text{ with } \|\mathbf{x}\| = 1\}$. We can then write $\|\Delta W\phi(\mathbf{x}_0)\| \sim \|\Delta W\|_2 \|\mathbf{x}_0\|$, and hence $\|\Delta W\|_2 \sim \frac{1}{\sqrt{N}}$. The operator norm also bounds the eigenvalues $\Delta W$ and hence the effect of the matrix on the dynamics of the system. For large $N$, this implies that the changes $\Delta W$ are too small to change the dynamics qualitatively, and the latter remain chaotic. Note that because the network dynamics are chaotic, the term $\mathbf{b}$ in *Equation 12* diverges, so that our discussion is only valid for short times. Numerically, we find small weight changes and chaotic solutions even for large target times $t_i$ (not shown).

For the sine wave task, the network with initially chaotic dynamics indeed converges to such a solution (*Figure 9C*). The output does not extrapolate beyond the training interval $t \in [1, 21]$, and dynamics remain qualitatively similar to those before training. During the training interval, the dynamics also remain close to the initial trajectories (dashed line). Testing the response to small perturbations in $\mathbf{x}(0)$ indicated that the dynamics remain chaotic (not shown). No limit cycle was formed, and the spectrum of eigenvalues did not show outliers (*Appendix 1—figure 9*).

We called this regime 'lazy', following similar settings in feedforward networks (*Jacot et al., 2018*; *Chizat et al., 2019*). Note that there, the output $z(t)$ is linearized around the weights at initialization (as opposed to the dynamics, *Equation 9*). This can be done in our case as well:

$$z(t) = z_0(t) + \sum_{ij} \mathbf{w}_{\text{out}}^T \frac{d\mathbf{x}(t)}{dW_{ij}}\bigg|_{W_0} \Delta W_{ij} + O(\Delta W^2) \,. \tag{17}$$

Demanding $\hat{z}(t) = z(t)$ yields a linear equation for each time point $t$. As we have $N^2$ parameters, this system is typically underconstrained. Gradient descent for this linear system leads to the minimal norm solution, which can also be found directly using the Moore-Penrose pseudo-inverse. Numerically, we

found that the weights $\Delta W_{\mathrm{lin}}$ obtained by this linearization are very close to those found by gradient descent (GD) on the nonlinear system, $\Delta W_{\mathrm{GD}}$, with Frobenius norm $\|\Delta W_{\mathrm{GD}} - \Delta W_{\mathrm{lin}}\| \sim \frac{1}{N}$ (section Linear approximation for lazy learning).

## Marginal and lazy solutions disappear with noise during training

The deduction above hinges on the assumption that $\Delta \mathbf{x}$ is small. This assumption does not hold if dynamics are noisy, *Equation 4*. For marginal dynamics, the noise would push solutions to different attractors or different positions along the limit cycle. For lazy dynamics, the chaotic dynamics would amplify any perturbations along the trajectory (if chaos persists under noise; *Schuecker et al., 2018*).

We will explore how learning is affected by noise in the sections below. Here, we only show that adding noise for our example task abolishes the marginal or lazy solutions and leads to oblique ones (*Figure 9D*). We added white noise with amplitude $\sigma_{\mathrm{noise}} = 1/\sqrt{2}$ to the dynamics during learning. After training, the output was a noisy sine wave. States were order one, and the 3D projection showed dynamics along a limit cycle that was almost orthogonal to the output vector. The noise in the 2D subspace of the first two PCs was small, $O(\frac{1}{\sqrt{N}})$, and thus did not disrupt the dynamics (e.g. very little phase shift). The eigenvalue spectrum had two outliers whose real part was increased in comparison to those in the aligned regime (*Appendix 1—figure 9*). Note that for the chosen values $g = 1.5$ and $\sigma_{\mathrm{noise}} = 1/\sqrt{2}$, the network actually was not chaotic at initialization (*Schuecker et al., 2018*). However, the choice of $g$ does not influence the solution in the oblique regime qualitatively, so both marginal and lazy solutions cease to exist g.

## Learning with noise for linear RNNs

In the next two sections, we aim to understand how adding noise affects dynamics and training. We start with a simple setting of a linear RNN which allows us to track the learning dynamics analytically. Despite its simplicity, this setting already captures a range of observations: different time scales for learning the bias and variance part, and the rise of a negative feedback loop for noise suppression. Oblique dynamics, however, do not arise, showing that these need autonomously generated, nonlinear dynamics, covered in section Oblique solutions arise for noisy, nonlinear systems.

We consider a linear network driven by a constant input and additional white noise. The dynamics read

$$\dot{\mathbf{x}} = (-I + W)\mathbf{x} + \mathbf{w}_{\mathrm{in}} + \boldsymbol{\xi}, \tag{18}$$

with noise $\boldsymbol{\xi}$ as in *Equation 4*. We focus on a simplified task, which is to produce a constant nonzero output $\hat{z}$ once the average dynamics converged, i.e., for large trial times. The output is $z = \mathbf{w}_{\mathrm{out}}^T \mathbf{x} = \bar{z} + \delta z$, where the bar denotes average over the noise, and the delta fluctuations around the average. The average is given by $\bar{z} = \mathbf{w}_{\mathrm{out}}^T (I - W)^{-1} \mathbf{w}_{\mathrm{in}}$. We train the network by changing only the recurrent weights $W$ via gradient descent. For small output weights, the fluctuations are too small to affect training: $\delta z = O(1/\sqrt{N})$. Apart from a small correction, learning dynamics are then the same as for small output weights and no noise, a setting analyzed in *Schuessler et al., 2020b*. Here, we only consider the case of large output weights.

The loss separates into two parts, $L = L_{\mathrm{bias}} + L_{\mathrm{var}}$ with $L_{\mathrm{bias}} = (\bar{z} - \hat{z})^2$ and $L_{\mathrm{var}} = \overline{\delta z^2} = \mathrm{var}(\delta z)$. Learning aims to minimize the sum. We first consider learning based on each part alone and then join both to describe the full learning dynamics.

Learning based on the bias part alone converges to a lazy solution (see section Details linear model: Bias only: lazy learning). For no initial weights, $W_0 = 0$, we have to leading order

$$\Delta W(\tau) = \frac{b_1(\tau)}{\sqrt{N}} \hat{\mathbf{w}}_{\mathrm{out}} \hat{\mathbf{w}}_{\mathrm{in},\perp}^T, \tag{19}$$

with

$$b_1(\tau) = (1 - e^{-2N\eta\tau})(\hat{z} - \sqrt{N}\rho_{io}). \tag{20}$$

Thus, $\Delta W$ is rank one with norm $\|\Delta W\|_2 \sim \frac{1}{\sqrt{N}}$. Furthermore, for a learning rate $\eta$, it converges in $O(\frac{1}{\eta N})$ time steps. We will see that this is very fast compared to learning the variance part.

## Learning to reduce noise alone slowly produces a negative feedback loop

Next, we consider learning based on the variance part $L_{var}$ alone, i.e., to reduce fluctuations in the output while ignoring the mean. The network dynamics are linear, so that $\delta\mathbf{x}$ is an Ornstein-Uhlenbeck process. Its stationary variance $\Sigma$ is the solution to the Lyapunov equation

$$0 = A\Sigma + \Sigma A^T + 2\sigma^2_{noise}I, \tag{21}$$

where $A = -I + W$. The variance part of the loss is then

$$L_{var} = \mathbf{w}^T_{out}\Sigma\mathbf{w}_{out}. \tag{22}$$

One can state the gradient of this loss in terms of a second Lyapunov equation (*Yan et al., 2016*):

$$G_{var} = \frac{dL_{var}}{dW} = 2\Omega\Sigma, \tag{23}$$

where $\Omega$ is the solution to

$$0 = A\Omega + \Omega A^T + \mathbf{w}_{out}\mathbf{w}^T_{out}. \tag{24}$$

Generally, solving both Lyapunov equations analytically is not possible, and even results for random matrices are still sparse (e.g. for symmetric Wigner matrices $W$ *Preciado and Rahimian, 2016*). To gain intuition, we thus restrict ourselves to the case of no initial connectivity, which leads to the connectivity spanned by input and output weights only. We start with the simplest case of a rank-one matrix only spanned by the output weights, $W = \lambda_-\mathbf{w}_{out}\mathbf{w}^T_{out}$, and extend to rank two in the following section. The Lyapunov equations then become 1D, and we obtain

$$\Sigma = \sigma^2_{noise}\left(I + \frac{\lambda_-}{1 - \lambda_-}\mathbf{w}_{out}\mathbf{w}^T_{out}\right), \tag{25}$$

$$\Omega = \frac{1}{2(1 - \lambda_-)}\mathbf{w}_{out}\mathbf{w}^T_{out}. \tag{26}$$

The gradient $G_{var} = 2\Omega\Sigma$ is therefore in the same subspace as $W$, and we can evaluate the 1D dynamics

$$\frac{d\lambda_-(\tau)}{d\tau} = \frac{-\eta\sigma^2_{noise}}{[1 - \lambda_-(\tau)]^2}, \tag{27}$$

where $\tau$ is the number of update steps. We assume $\tau$ to be continuous, i.e., we assume a sufficiently small learning rate and approximate the discrete dynamics of gradient descent with gradient flow. With initial condition $\lambda_-(0) = 0$, the solution is

$$\lambda_-(\tau) = 1 - \left(3\eta\sigma^2_{noise}\tau + 1\right)^{\frac{1}{3}}, \tag{28}$$

which is negative for $\tau > 0$. The loss then decays as

$$L_{var} = \sigma^2_{noise}\left(3\eta\sigma^2_{noise}\tau + 1\right)^{-\frac{1}{3}}, \tag{29}$$

namely at order $O(\tau^{-1/3})$ in learning time. We thus obtained that, during the variance phase of learning, connectivity develops a negative feedback look aligned with the output weights, which serves to suppress output noise. For very long learning times, $\tau \sim N^3/\eta$, learning can in principle reduce the output fluctuations to $\frac{1}{\sqrt{N}}$. Note, however, that this implies a huge negative feedback loop, $\lambda_- = O(N)$, which potentially leads to instability in a system with delays or discretized dynamics (*Kadmon et al., 2020*).

## Optimizing mean and fluctuations occurs on different time scales

We now consider learning both the mean and the variance part. For zero initial recurrent weights, $W_0 = 0$, the input and output vectors make up the only relevant directions in space. We thus express

the recurrent weights as a rank-two matrix, $W = \hat{U}M\hat{U}^T$, with orthonormal basis $\hat{U} = [\hat{\mathbf{w}}_{\text{out}}, \hat{\mathbf{w}}_{\text{in},\perp}]$. The hats indicate normalized vectors. For large networks and large output weights, the first vector is already normalized, $\hat{\mathbf{w}}_{\text{out}} = \mathbf{w}_{\text{out}}$. The second vector is the input weights after Gram-Schmidt. Assuming that $\mathbf{w}_{\text{out}}$ and $\mathbf{w}_{\text{in}}$ are drawn independently, we have a small, random correlation $\rho_{io} = O(1/\sqrt{N})$, and can write $\hat{\mathbf{w}}_{\text{in},\perp} = \hat{\mathbf{w}}_{\text{in}} - \mathbf{w}_{\text{out}}\rho_{io} + O(\frac{1}{N})$.

We computed the learning dynamics in terms of the coefficient matrix $M$, using the same tools introduced above, and the insight that learning the mean is much faster than learning to reduce the variance. The details are relegated to section Details linear model: Bias and variance combined, here, we summarize the results. We obtained

$$M(\tau) = \begin{bmatrix} \lambda_-(\tau) & \dfrac{b(\tau)}{\sqrt{N}} \\ 0 & 0 \end{bmatrix}. \tag{30}$$

Before discussing the temporal evolution of the two components, we analyze the structure of the matrix. The eigenvalue $M_{11} = \lambda_-$ is a negative feedback loop along the eigenvector $\mathbf{w}_{\text{out}}$, and $M_{21} = b/\sqrt{N}$ is a small feedforward component $b$ which maps the input to the output. Along the input direction $\hat{\mathbf{w}}_{\text{in}}$, learning does not change the dynamics: The second eigenvalue, corresponding to this direction, is zero.

The dynamics unfold on two time scales. First, there is a very fast learning of the bias via the feedforward coefficient

$$b(\tau) = (1 - e^{-2N\eta\tau})(\hat{z} - \sqrt{N}\rho_{io}). \tag{31}$$

During this phase, the eigenvalue $M_{11} = \lambda_-$ remains at zero, so that overall weight changes remain small, $\|W\| \sim 1/\sqrt{N}$. The average fixed point also does not change much, $\|\Delta\bar{\mathbf{x}}(\tau)\| = b_1(\tau) = O(1)$, in comparison to the fixed point before learning, $\|\bar{\mathbf{x}}_0\| = \|\mathbf{w}_{\text{in}}\| = \sqrt{N}$. The loss evolves as

$$L_{\text{bias}} = e^{-4N\eta\tau}(\hat{z} - \sqrt{N}\rho_{io})^2. \tag{32}$$

The variance part of the loss does not change during this phase.

In a second, slower learning phase, the eigenvalue $\lambda_-$ evolves like above, *Equation 28*, the case where only the variance part is learned. The second coefficient compensates for the resulting change in the output. This compensation happens at a much faster time scale ($N^3$ times faster than $\lambda_-$), so we consider its steady state:

$$b(\tau) = \hat{z}[1 - \lambda_-(\tau)] - \sqrt{N}\rho_{io}. \tag{33}$$

Because of this compensation, the bias part of the loss always remains at zero, and the full loss $L(\tau) = L_{\text{var}}(\tau)$ evolves as before, *Equation 29*. Meanwhile, the average fixed point does not change anymore; we have $\|\Delta\bar{\mathbf{x}}\| = \hat{z} - \rho_{io}\sqrt{N}$.

We compare our theoretical predictions against numerical simulations in *Figure 10*. For the task, we let the linear network dynamics converge from $\mathbf{x}(0) = \mathbf{0}$ until $t = 15$, and demand the output $z(t)$ to be at the target $\hat{z}$ during the interval $t \in [15, 20]$. Because the first learning phase converges $N$ times faster than the second one (with $N = 256$), using a single learning rate $\eta$ is problematic. One can either observe the first phase only (for small $\eta$) or risk unstable learning during the first phase (for large $\eta$). We thus split learning into two parts with adapted learning rates. For the initial phase, we set a learning rate to $\eta = \eta_0/N$, with $\eta_0 = 0.002$ (*Figure 10*, left column). For the second phase, we set $\eta = \eta_0$ (*Figure 10*, right column). Theory and simulation agree well for both phases. Small deviations can be observed for the second phase and long learning times: nonzero coefficients $M_{21}$ and $M_{22}$ and a corresponding increase in the norm $\|\Delta\bar{\mathbf{x}}\|$. Testing with larger network sizes showed that these errors decreased as $O(1/N)$, which is consistent with our theory above (not shown).

Our results show that learning in this linear system is a hybrid between oblique and aligned during the second phase. We have a large term that compresses the output noise, but a very small, 'lazy' correction in the feedforward component that corrects the output, and the states are only marginally changed. Although this system is a somewhat degenerate limiting case, we can still derive important insights. The two most striking features – the different time scales of the two learning processes, and

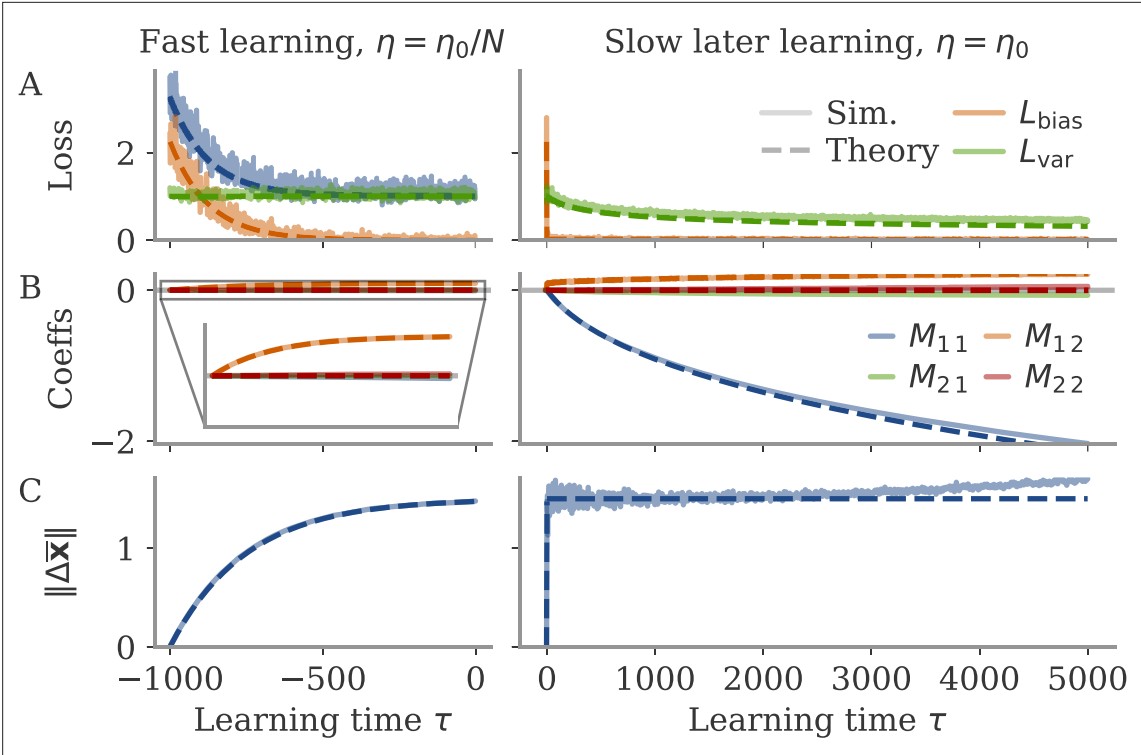

**Figure 10.** Noise-induced learning for a linear network with input-driven fixed point. Learning separates into fast learning of the bias part of the loss (left), and slow learning reducing the variance part (right). Learning rates are $\eta = \eta_0/N$ and $\eta = \eta_0$, respectively, with $\eta_0 = 0.002$ and network size $N = 256$. Learning epochs in the first phase are counted from –1000, so that the second phase starts at 0. In the right column, the initial learning phase with learning time steps multiplied by $1/N$ is shown for comparison. In all plots, simulations (full lines) are compared with theory (dashed lines). (**A**) Loss $L = L_{bias} + L_{var}$. The two components are obtained by averaging over a batch with 32 examples at each learning step. The full loss is not plotted in the slow phase, because it is indistinguishable from $L_{var}$. (**B**) Coefficients of the 2-by-2 coupling matrix $M$. $M_{11} = \lambda_-$ is the feedback loop along the output weights, $M_{12} = b/\sqrt{N}$ a feedforward coupling from input to output. The theory predicts $M_{21} \sim M_{22} \sim O(1/N)$. (**C**) Norm of state changes during training. The theory predicts that it remains constant during the second phase and small compared to $\|\bar{\mathbf{x}}_0\| = \sqrt{N}$. Other parameters: target $\hat{z} = 1$, $\sigma_{noise} = 1$, overlap between input and output vectors $\rho_{io} = -0.5/\sqrt{N}$.

the slow emergence of negative feedback, $\lambda_-(\tau) \sim -\tau^{1/3}$, along the output – are found robustly also for nonlinear networks and other tasks.

## Oblique solutions arise for noisy, nonlinear systems

We now examine the origin of oblique solutions. The linear system did not yield oblique solutions, so we turn to a nonlinear model. We consider a 1D flip-flop task, where the network has to yield a constant, nonzero output $\hat{z}$ depending on the sign of the last input pulse. We further simplify the analysis by only considering the steady state of a network, not how the input pulse mediates the transition. At the output, we thus only consider the average $\bar{z}$ and the fluctuations $\delta z$. As for the linear network, the loss splits into a bias part $L_{bias} = (\bar{z} - \hat{z})^2$ and variance part $L_{var} = \overline{\delta z^2}$. As before, we assume that learning only acts on a low-dimensional parameter matrix $M$.

Because following the learning dynamics in nonlinear networks is difficult, we take a different approach. We develop a mean field theory to show how noise affects the dynamics of a nonlinear network with autonomous fixed points. This allows us to compute the loss components $L_{bias}$ and $L_{var}$ in terms of $M$. We then show that the minimum of the loss function corresponds to oblique solutions. This leads to a clear interpretation of the mechanisms pushing for oblique solutions. Finally, we show that the theory quantitatively predicts the outcome of learning with gradient descent.

### Rank-two connectivity model with fixed point

We first introduce the connectivity model and compute the latent dynamics using mean field theory. We constrain the recurrent connectivity to a rank-two model of the form $W = \frac{1}{N}UMU^T$, where $M$ is a

2×2 coefficient matrix to be learned. The randomly drawn projection matrix $U \in \mathbb{R}^{N \times 2}$ has entries $U_{ia}$ drawn independently from a standard normal distribution. This implies orthogonality to leading order, $\frac{1}{N} U^T U = I_2 + O(1/\sqrt{N})$. We will discard the correction term, as it does not change our results apart from a constant bias. We further assume that the input and output vectors are spanned by $U$, although not necessarily identified with the components of $U$ as in the previous section. Note also that for clarity we do not normalize $U$ as $\hat{U}$ before. The assumption of Gaussian connectivity greatly simplifies the math, while its restrictions are irrelevant to the task considered here (*Schuessler et al., 2020a*; *Beiran et al., 2021*; *Dubreuil et al., 2021*).

To understand the dynamics *Equation 4* analytically, we make use of the low-rank connectivity. Following previous work (*Rivkind and Barak, 2017*; *Kadmon et al., 2020*), we split the dynamics into two parts: a parallel part $\mathbf{x}_\parallel$ in the subspace spanned by $U$, and an orthogonal part $\mathbf{x}_\perp$. This yields

$$\dot{\mathbf{x}}_\parallel = -\mathbf{x}_\parallel + \frac{1}{N} U M U^T \phi(\mathbf{x}_\parallel + \mathbf{x}_\perp) + \frac{1}{N} U U^T \boldsymbol{\xi} , \tag{34}$$

$$\dot{\mathbf{x}}_\perp = -\mathbf{x}_\perp + (I - \frac{1}{N} U U^T)\boldsymbol{\xi} . \tag{35}$$

Notice that the parallel part is partially driven by the orthogonal one, but not vice versa. The parallel part can be expressed in terms of the latent variable

$$\boldsymbol{\kappa} = \frac{1}{N} U^T \mathbf{x} = \frac{1}{N} U^T \mathbf{x}_\parallel . \tag{36}$$

The scaling here ensures that $\boldsymbol{\kappa}$ is order one if $\mathbf{x}_\parallel$ has order-one states. Because the readout is assumed to be spanned by $U$, the output is fully determined by the parallel part. We can write

$$z = \mathbf{w}_{\text{out}}^T \mathbf{x}_\parallel = \sqrt{N} \mathbf{v}_{\text{out}}^T \boldsymbol{\kappa} , \tag{37}$$

with projected output weights $\mathbf{v}_{\text{out}} = \frac{1}{\sqrt{N}} U^T \mathbf{w}_{\text{out}}$. Note that we assume large output weights, $\|\mathbf{w}_{\text{out}}\| = 1$, so that $\mathbf{v}_{\text{out}}$ is also normalized.

We split the latent state into its average over the noise $\boldsymbol{\xi}$ and fluctuations, $\boldsymbol{\kappa} = \bar{\boldsymbol{\kappa}} + \delta\boldsymbol{\kappa}$. Similarly, the output splits into $z = \bar{z} + \delta z$. The loss then has two components, $L = L_{\text{bias}} + L_{\text{var}}$, with

$$L_{\text{bias}} = (\bar{z} - \hat{z})^2 = \left( \sqrt{N} \mathbf{v}_{\text{out}}^T \bar{\boldsymbol{\kappa}} - \hat{z} \right)^2 , \tag{38}$$

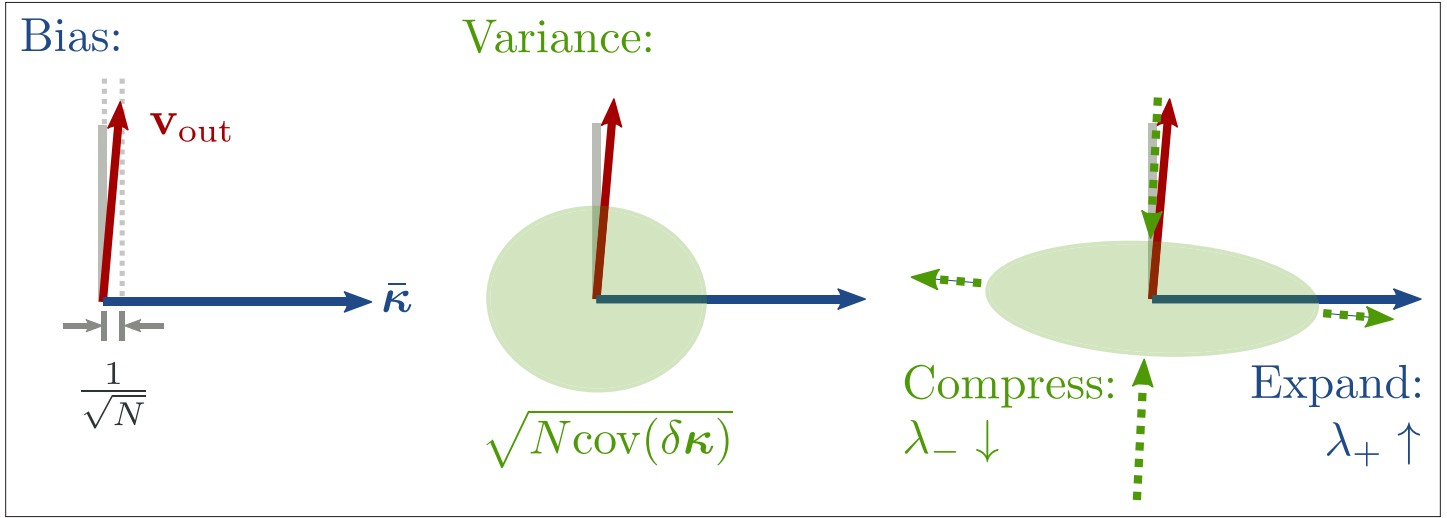

**Figure 11.** Cartoon illustrating the split into bias and variance components of the loss, and noise suppression along the output direction. The two-dimensional subspace spanned by $U$ illustrates the main directions under consideration: the principal components (PCs) of the average trajectories (here only a fixed point $\bar{\mathbf{x}} = U\bar{\boldsymbol{\kappa}}$), and the direction of output weights $\mathbf{w}_{\text{out}} = \frac{1}{\sqrt{N}} U \mathbf{v}_{\text{out}}$. Left: During learning, a fast process keeps the average output close to the target so that $L_{\text{bias}} = 0$. Center: The variance component, $L_{\text{var}}$, is determined by the projection of the fluctuations $\delta\boldsymbol{\kappa}$ onto the output vector. Note that the noise in the low-D subspace is very small, $\delta\boldsymbol{\kappa} = O(1\sqrt{N})$, but the output is still affected due to the large output weights. Right: During training, the noise becomes non-isotropic. Along the average direction $\bar{\boldsymbol{\kappa}}$, the fluctuations are increased as a byproduct of the positive feedback $\lambda_+$. Meanwhile, a slow learning process suppresses the output variance via a negative feedback $\lambda_-$.

$$L_{\text{var}} = \overline{\delta z^2} = N \mathbf{v}_{\text{out}}^T \overline{\delta\boldsymbol{\kappa}\delta\boldsymbol{\kappa}^T} \mathbf{v}_{\text{out}} . \tag{39}$$

We want to understand situations with small loss. For the bias term, the average $\bar{\kappa}$ must be either small or oblique to the readout weights. For the variance term, the covariance of the fluctuations, $\text{cov}(\delta\boldsymbol{\kappa}) = \overline{\delta\boldsymbol{\kappa}\delta\boldsymbol{\kappa}^T}$, must be compressed along the output direction: Even though the covariance is $O(1/N)$, it still projects to the output at $O(1)$, as reflected by the factor $N$ in **Equation 39**. **Figure 11** illustrates the situation in a cartoon from this high-level perspective.

To understand the underlying mechanisms in detail, we next explore how the relevant variables $\bar{\kappa}$ and $\delta\boldsymbol{\kappa}$ are determined by the coupling matrix $M$. We do so by applying mean field theory, following previous works (**Mastrogiuseppe and Ostojic, 2018**; **Schuessler et al., 2020a**; **Kadmon et al., 2020**; **Schuecker et al., 2018**). Detailed derivations can be found in the section Details nonlinear autonomous system with noise. Here, we present the high-level results. The average dynamics converge to a fixed point determined by the equation for the latent variable,

$$\bar{\boldsymbol{\kappa}} = \langle \phi' \rangle M \bar{\boldsymbol{\kappa}} + O(\tfrac{1}{\sqrt{N}}) . \tag{40}$$

The $O(1/\sqrt{N})$ term is a constant offset for any given network that we ignore without loss of generality. The average slope is

$$\langle \phi' \rangle = \langle \phi'(\overline{\sigma_x} u) \rangle = \int \mathcal{D}u \, \phi'\left(\overline{\sigma_x} u\right) , \tag{41}$$

with variance

$$\overline{\sigma_x^2} = \|\bar{\boldsymbol{\kappa}}\|^2 + \overline{\sigma_\perp^2} . \tag{42}$$

(We have $\overline{\sigma_x} = \sqrt{\overline{\sigma_x^2}}$ because the fluctuations are small.) The orthogonal variance is simply the variance of the noise, $\overline{\sigma_\perp^2} = \sigma_{\text{noise}}^2$. The fixed point (**Equation 40**) implies that for a nonzero fixed point, the matrix $M$ must have an eigenvalue

$$\lambda_+ = \frac{1}{\langle \phi' \rangle} . \tag{43}$$

This is very similar to the noiseless situation discussed briefly above, **Equation 14**. However, here the additional variance $\overline{\sigma_\perp^2}$ decreases the average slope due to saturation of the nonlinearity. This in turn increases the minimal eigenvalue for a nonzero fixed point, which can be found by setting $\bar{\boldsymbol{\kappa}} = \mathbf{0}$, and hence $\overline{\sigma_x} = \overline{\sigma_\perp}$:

$$\lambda_{+,\text{min}} = \frac{1}{\langle \phi'(\overline{\sigma_\perp} u) \rangle} . \tag{44}$$

In other words, the noise decreases the effective gain $\langle \phi' \rangle$, and thus the connectivity eigenvalue $\lambda_+$ needs to compensate. From the point of view of the spectrum, we are thus pushed away from the margin $1 + \epsilon$. These considerations, however, do not exclude the possibility that dynamics converge to an average fixed point that is small and correlated, which is at odds with oblique dynamics. To understand why learning leads to oblique dynamics, we need to move beyond the average $\bar{\kappa}$ and take into account the fluctuations $\delta\boldsymbol{\kappa}$.

## Fluctuations of the latent variable

The fluctuations $\delta\boldsymbol{\kappa}$ around a fixed point $\bar{\kappa}$ are driven by the noise, both directly and indirectly via the dynamics. The direct contribution is a white noise term of order $1/\sqrt{N}$ because $\boldsymbol{\xi}$ is isotropic and independent of $U$. A detailed analysis (section Details nonlinear autonomous system with noise) shows that the indirect contribution is given by a colored noise term which originates from the finite size fluctuations in the variance of the orthogonal part. This second term is also $O(1/\sqrt{N})$, which implies that the fluctuations are small, $\delta\boldsymbol{\kappa} = O(1/\sqrt{N})$. We can thus linearize their dynamics around the mean $\bar{\kappa}$, which yields

$$\frac{\mathrm{d}\,\delta\boldsymbol{\kappa}(t)}{\mathrm{d}t} = A\delta\boldsymbol{\kappa}(t) + \frac{1}{\sqrt{N}}\boldsymbol{\zeta}(t),\tag{45}$$

where the order-one term $\boldsymbol{\zeta}$ contains both the white and the colored noise term. The Jacobian $A$ depends on $M$ and $\bar{\boldsymbol{\kappa}}$:

$$A = -I + \langle\phi'\rangle M + \frac{\langle\phi'''\rangle}{\langle\phi'\rangle}\bar{\boldsymbol{\kappa}}\bar{\boldsymbol{\kappa}}^T.\tag{46}$$

The averages are again evaluated at the joint variance $\sigma_x^2 = \|\bar{\boldsymbol{\kappa}}\|^2 + \sigma_\perp^2$. Apart from the increased variance, the stability analysis yields the same results as in the noise-free case (**Schuessler et al., 2020a**): The Jacobian has the eigenvalues $\gamma_+ = \frac{\langle\phi'''\rangle}{\langle\phi'\rangle}\|\bar{\boldsymbol{\kappa}}\|^2$ and $\gamma_- = \frac{\lambda_-}{\lambda_+} - 1$. The average over the third derivative $\langle\phi'''\rangle$ is negative, so that $\gamma_+ < 0$. We assume that the second eigenvalue is smaller than the first, $\lambda_- < \lambda_+$, so that $\gamma_- < 0$. The fixed point under consideration is hence stable.

Next, we compute the covariance of the fluctuations at steady state, see section Details nonlinear autonomous system with noise. The outcome is

$$\overline{\delta\boldsymbol{\kappa}\delta\boldsymbol{\kappa}^T} = \frac{1}{N}\left[\sigma_{\mathrm{noise}}^2\Sigma_A + \frac{\sigma_{\mathrm{noise}}^4}{\|\bar{\boldsymbol{\kappa}}\|^2}\frac{-\gamma_+}{2(2-\gamma_+)}\mathbf{v}_+\mathbf{v}_+^T\right],\tag{47}$$

where $\mathbf{v}_+ = \bar{\boldsymbol{\kappa}}/\|\bar{\boldsymbol{\kappa}}\|$ is the normalized eigenvector of $M$ corresponding to eigenvalue $\lambda_+$. The 2×2 matrix $\Sigma_A$ is the covariance introduced by the white noise part alone and obeys the Lyapunov equation

$$0 = A\Sigma_A + \Sigma_A A^T + 2I_2.\tag{48}$$

The second term in **Equation 47** stems from the colored noise component of $\boldsymbol{\zeta}$.

The loss (**Equation 39**) is obtained by projecting the covariance on the output weights. Importantly, the factor $1/N$ in the covariance is compensated by the factor $N$ in the loss. Hence, even if the covariance shrinks with increasing network size, the output is still affected at order one. We next explore the implications of minimizing this loss.

## Minimizing the loss by balancing saturation and negative feedback loop

To gain an understanding of how the output fluctuations responsible for $L_{\mathrm{bias}}$ can be reduced, we first consider the case of a symmetric coefficient matrix $M$. Simulations of networks trained with gradient descent below show that this approximation is reasonable. For symmetric $M$, the orthogonal eigenvectors $\mathbf{v}_\pm$ with eigenvalues $\lambda_\pm$ of the recurrent weights $M$ are also eigenvectors of $A$, in that case corresponding to the eigenvalues $\gamma_\pm$. This allows to diagonalize the Lyapunov **Equation 48** and yields the solution

$$\Sigma_A = \frac{\mathbf{v}_+\mathbf{v}_+^T}{-\gamma_+} + \frac{\mathbf{v}_-\mathbf{v}_-^T}{-\gamma_-}.\tag{49}$$

For the loss (**Equation 39**), we further need the relation between the eigenvectors $\mathbf{v}_\pm$ and the output weights. Because the fixed point $\bar{\boldsymbol{\kappa}}$ is parallel to the eigenvector, we have $\mathbf{v}_{\mathrm{out}}^T\mathbf{v}_+ = \rho$. For the other eigenvector, orthogonality yields $\mathbf{v}_{\mathrm{out}}^T\mathbf{v}_- = \sqrt{1-\rho^2}$. Inserting this into **Equations 39 and 47** yields an expression in terms of the Jacobian eigenvalues $\gamma_\pm$, the correlation $\rho$, and the norm of the fixed point $\|\bar{\boldsymbol{\kappa}}\|$:

$$L_{\mathrm{var}} = \rho^2\left[\frac{\sigma_{\mathrm{noise}}^2}{-\gamma_+} + \frac{\sigma_{\mathrm{noise}}^4}{\|\bar{\boldsymbol{\kappa}}\|^2}\frac{\gamma_+}{2(\gamma_+-2)}\right] + (1-\rho^2)\frac{\sigma_{\mathrm{noise}}^2}{-\gamma_-}.\tag{50}$$

For more explicit insight, we choose the nonlinearity $\phi(x) = \mathrm{erf}(\alpha x)$ with $\alpha = \sqrt{\pi}/2$. Similar to tanh, this function is bounded between ±1 and has slope $\phi'(0) = 1$ at the origin. For this function, we can explicitly compute the relevant Gaussian integrals. The minimal eigenvalue of $M$ to produce a fixed point, **Equation 44**, is then given by $\lambda_{+,\mathrm{min}} = 1 + 2\alpha^2\sigma_{\mathrm{noise}}^2$. For $\lambda_+ > \lambda_{+,\mathrm{min}}$, the resulting fixed point has norm

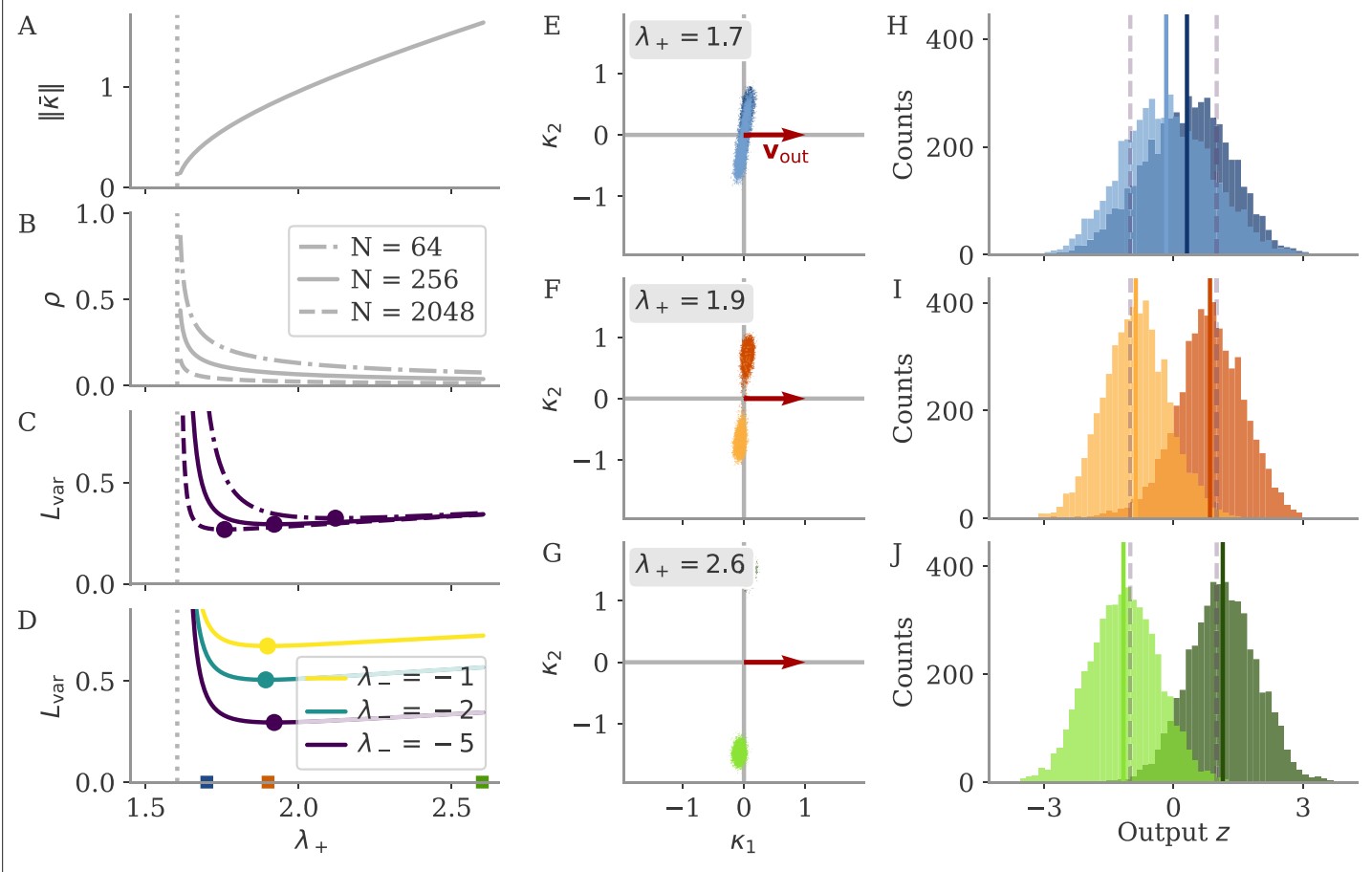

**Figure 12.** Mechanisms behind oblique solutions predicted by mean field theory. (**A–D**) Mean field theory predictions as a function of positive feedback strength $\lambda_+$. The dotted lines indicate $\lambda_{+,\min}$, the minimal eigenvalue necessary to generate fixed points. (**A**) Norm of fixed point $\|\bar{\kappa}\|$. (**B**) Correlation $\rho$ so that $L_{\text{bias}} = 0$. (**C, D**) Loss due to fluctuations for different $\lambda_-$ or networks sizes $N$. Dots indicate minima. (**E–G**) Latent states $\kappa$ of simulated networks for randomly drawn projections $U$. The symmetric matrix $M$ is fixed by setting $\lambda_+$ as noted, $\lambda_- = -5$, and demanding $L_{\text{bias}} = 0$ (for the mean field prediction). Dots are samples from the simulation interval $t \in [20, 100]$. (**H–J**) Histogram for the corresponding output $z$. Mean is indicated by full lines, the dashed lines indicate the target $\hat{z}$. Other parameters: $N = 256$, $\sigma_{\text{noise}} = 1$, $\hat{z} = 1$.

$$\|\bar{\kappa}\|^2 = \frac{\lambda_+^2 - \lambda_{+,\min}^2}{2\alpha^2}, \tag{51}$$

as shown in *Figure 12A*. The larger eigenvalue of the Jacobian is $\gamma_+ = -(\lambda_+^2 - \lambda_{+,\min}^2)/\lambda_+^2$. We next obtain the correlation between fixed point and output weights by assuming that the bias part of the loss (*Equation 38*) is kept at zero. This is reasonable because it requires only a small adaptation to the weights that leaves the variance part mostly untouched. The resulting correlation is

$$\rho^2 = \frac{1}{N} \frac{\hat{z}^2}{\|\bar{\kappa}\|^2}, \tag{52}$$

so that increasing $\lambda_+$ also decreases the correlation (*Figure 12B*). Finally, we obtain an expression for the variance part of the loss only in terms of the eigenvalues of $M$:

$$L_{\text{var}} = \rho^2 \left( \frac{\sigma_{\text{noise}}^2 \lambda_+^2}{\lambda_+^2 - \lambda_{+,\min}^2} + \frac{\sigma_{\text{noise}}^4 \alpha^2}{3\lambda_+^2 - \lambda_{+,\min}^2} \right) + (1 - \rho^2) \frac{\sigma_{\text{noise}}^2}{1 - \frac{\lambda_-}{\lambda_+}}. \tag{53}$$

We show $L_{\text{var}}$ over $\lambda_+$ for different negative feedback loop sizes $\lambda_-$ (*Figure 12C*) and different network sizes $N$ (*Figure 12D*). The first term diverges at the phase transition where the fixed point

appears, $\lambda_+ \searrow \lambda_{+,\min}$. Learning will thus push the weights away from the phase transition toward larger $\lambda_+$. With such increasing $\lambda_+$, the fixed point norm increases, and the fixed point rotates away from the output, decreasing the correlation. This in term emphasizes the last term, scaled by $1 - \rho^2$. The last term is reduced with increasingly negative $\lambda_-$, corresponding to the negative feedback loop that suppresses noise.

Learning can in principle strengthen this feedback further and further, $\lambda_- \to -\infty$ (apart from possible stability issues; *Kadmon et al., 2020*). However, as for the linear network, section Learning with noise for linear RNNs, this process takes time. We thus assume $\lambda_-$ to be fixed and search for a minimum across $\lambda_+$. For $\lambda_- < 0$, the last term in *Equation 53* *increases* with increasing $\lambda_+$: the effective feedback loop in the full, nonlinear system is weakened by saturation. The loss $L_{\mathrm{var}}$ thus has a minimum at some moderate $\lambda_+$.

To illustrate the mechanisms described above, we simulated networks at different $\lambda_+$ and with $\lambda_- = -5$. For each $\lambda_+$, we compute $\rho$ according to *Equation 52*. Setting $\mathbf{v}_{\mathrm{out}} = [1, 0]^T$, we then set the resulting symmetric $M = V\Lambda V^T$. For each network sample, we then draw independent random projections $U \in \mathbb{R}^{N \times 2}$. We started simulations at either one of the two nonzero fixed points. For $\lambda_+$ just above $\lambda_{+,\min}$, the noise pushes activity from one basin of attraction to the next (*Figure 12C*). The resulting output becomes centered around zero and independent of the initial condition for long simulation times (*Figure 12F*). For the optimal $\lambda_+$, the trajectories remain close to either one fixed point (*Figure 12F*). The output forms two overlapping distributions, each closely matching the target on average (*Figure 12I*). For larger $\lambda_+$, the fixed points become increasingly larger (*Figure 12G*). While this decreases the probability of leaving the basin of attraction even further, the variance along the output weights becomes larger (slightly wider histograms in *Figure 12J*). Note that the mean starts to deviate from the prediction. This is not covered by our theory and is potentially due to the linearization of the fluctuations.

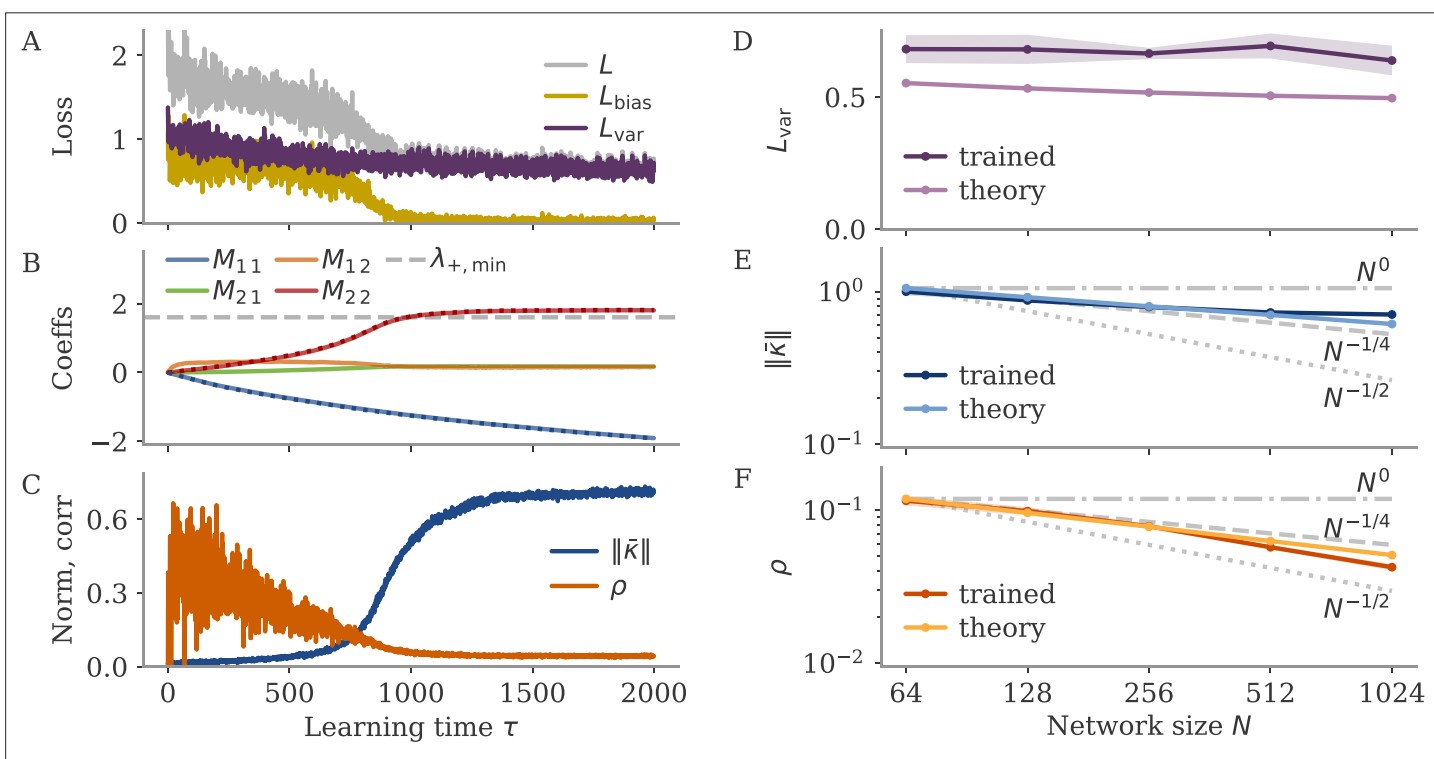

**Figure 13.** Mean field theory predicts learning with gradient descent. (**A–C**) Learning dynamics with gradient descent for example network with $N = 1024$ neurons and with noise variance $\sigma_{\mathrm{noise}}^2 = 1$. (**A**) Loss with separate bias and variance components. (**B**) Matrix coefficients $M_{ij}$. The dotted lines almost identical to $M_{22}$ and $M_{11}$ indicate the eigenvalues $\lambda_+$ and $\lambda_-$, respectively. The dashed line indicates $\lambda_{+,\min}$. (**C**) Fixed point norm and correlation. (**D–F**) Final loss, fixed point norm, and correlation for networks of different sizes $N$. Shown are mean (dots and lines) and standard deviation (shades) for five sample networks, and the prediction by the mean field theory. Gray lines indicate scaling as $aN^k$, with $k \in \{0, -1/4, -1/2\}$. Note the log-log axes for (**E**, **F**).

All-in-all, this section revealed a potential path to oblique solutions, initiated by the large fluctuations close to a phase transition, and the interplay between the negative feedback loop and saturation. In the following section, we show that learning via gradient descent actually follows this path and that the parameters predicted by the minimum of $L_{\text{var}}$ quantitatively predict solutions from learning.

## Oblique solutions from learning are predicted by the mean field theory

We trained neural networks on the fixed points task described above by applying gradient descent to the 2×2 matrix $M$, initialized at $M_0 = 0$. The gradients $G_M$ with respect to $M$ are equivalent to those with respect to $W = \frac{1}{N} U M U^T$ restricted to the subspace spanned by $U$, i.e., $G_M = \frac{1}{N} U U^T G_W \frac{1}{N} U U^T$. Such a restriction is exact in the case of linear RNNs without random initial connectivity, and yields qualitative insights even for nonlinear networks with random initial connectivity (**Schuessler et al., 2020b**).

The output weights were set to $\mathbf{w}_{\text{out}} = \frac{1}{\sqrt{N}} \mathbf{u}_1$, where $\mathbf{u}_1$ is the first of the two projection vectors $U = [\mathbf{u}_1, \mathbf{u}_2]$. We thus have large output weights with norm $\|\mathbf{w}_{\text{out}}\| = 1$. Trajectories were initialized at $\mathbf{x}(0) = \mathbf{0}$. At the beginning of a trial with target output $\hat{z} = \pm 1$, the network receives one pulse $s(t) = \pm\delta(t)$ along the input weights $\mathbf{w}_{\text{in}} = \mathbf{u}_2$. The input direction is hence the second available direction for the rank-two connectivity. This is a sensible choice as networks without the restriction to rank-two weights would span the recurrent weights from existing directions, and a rank-two connectivity would hence also be spanned by $\mathbf{w}_{\text{out}}$ and $\mathbf{w}_{\text{in}}$ (**Schuessler et al., 2020b**).

The loss over learning time for one network is shown in **Figure 13A**. Learning consisted of two phases: a first phase in which the network did not possess a fixed point and hence did not match the target on average, $L_{\text{bias}} > 0$. At some point, $L_{\text{bias}}$ rapidly decreases, and $L_{\text{var}}$ dominates the overall loss. During the second phase, $L_{\text{var}}$ slowly decreases, with $L_{\text{bias}}$ hovering around zero.

The coefficients of the matrix $M$ indicate the underlying learning dynamics (**Figure 13B**). The coefficient along the output weights, $M_{11}$, is almost identical with $\lambda_-$. It continually grows in the negative direction, unaffected by the different phases. Its time course is very similar to the time course observed for the linear system (**Figure 10B**). In contrast, the coefficient along the input weights, $M_{22}$, mirrors the two phases. It grows increasingly fast in the first phase and saturates during the second phase. Its value is very close to the larger eigenvalue, $\lambda_+$. The transition between the two phases of learning happens at the phase transition of the dynamical system when the fixed point emerges for $\lambda_+ = \lambda_{+,\text{min}}$. The off-diagonal entries show that $M$ is asymmetric during the first phase and becomes symmetric later on. The coefficient $M_{12}$ corresponds to the feedforward mode mapping the state decaying from $\mathbf{x}(0) = \mathbf{u}_2$ after the pulse to the output weights $\mathbf{w}_{\text{out}} = \frac{1}{\sqrt{N}} \mathbf{u}_1$.

Tracing the fixed point norm $\|\bar{\boldsymbol{\kappa}}\|$ and the correlation $\rho$ over learning time shows what we expected (**Figure 13C**): The norm grows rapidly at the phase transition, which is accompanied by a decrease in the correlation. The example yields a fixed point with norm $\|\bar{\boldsymbol{\kappa}}\| \approx 0.7$ and correlation $\rho \approx 0.05$ for a network with $N = 1024$. The mere numbers already suggest that we call this an oblique solution, but the theory description above is based on how these numbers scale with network size $N$. We trained networks with different $N$ but otherwise the same conditions. They reached the same loss (**Figure 13A**, $L_{\text{bias}} \approx 0$ not shown). The fixed point norm decreases weakly with $N$, with $\|\bar{\boldsymbol{\kappa}}\| = O(N^k)$, for some $k \geq -1/4$ (**Figure 13E**). The correlation decreases faster, yet not quite with $1/\sqrt{N}$ (**Figure 13F**).

We compared the outcome of learning to our mean field theory. Given that $M$ is approximately symmetric at the end of learning, we directly applied our results from the previous sections, again assuming $L_{\text{bias}} = 0$. We fixated $\lambda_-$ to match the value at the end of training and computed the $\lambda_+$ that minimized $L_{\text{var}}$, **Equation 53**. The results for the norm and correlation match those values obtained with gradient descent very closely.

Our high-level description of oblique and aligned dynamics did not involve scaling with network size (section Aligned and oblique population dynamics). However, the underlying assumption was that the activity of single neurons $x_i$ is not vanishing for large networks, i.e., $x_i = O(1)$. This would imply $\|\bar{\boldsymbol{\kappa}}\| = O(1)$ and $\rho = O(1/\sqrt{N})$. This is indeed what we observed for the more complex tasks when training networks of different sizes (not shown). We note that our results for the simple fixed point task deviate weakly from this (**Figure 13E and F**). This hints at other factors pushing solutions to $\|\bar{\boldsymbol{\kappa}}\| = O(1)$. One such factor may be that the loss function $L_{\text{var}}(\lambda_+)$ is very flat for $\lambda_+$ larger than the optimum (**Figure 12C and D**). If learning pushes trajectories beyond the optimum at some point, e.g., due to large updates or if the optimum shifts over learning, then the learning signal to reduce $\lambda_+$ afterward may be too small to yield visible effects in finite learning time.

In summary, the last two sections capture the main mechanisms that drive solutions to the oblique regime in nonlinear, noise-driven networks. Although marginally small solutions seem possible for large output weights, such solutions are close to a phase transition, so the resulting system is very susceptible to noise. To accommodate robust solutions, the trajectories (here the fixed points) must increase in magnitude, while rotating away from the output weights. This further allows the implementation of a negative feedback loop that suppresses noise along the output direction. Its efficacy can be reduced by too much saturation, which in turn keeps solutions from growing ever larger. The resulting sweet spot is a network with oblique dynamics.

## Mechanisms behind decoupling of neural dynamics and output

Here, we discuss in more detail the underlying mechanisms for the qualitative decoupling in oblique networks. We make a high-level argument that splits into two parts: first the possibility of decoupling in oblique, but not aligned, networks, and second a putative mechanism driving the decoupling.

For the first part, we observe that the output in oblique networks can be obtained from the leading components of the dynamics (along the PCs), but importantly also from the non-leading ones. To see this, we unpack the output in *Equation 1* in a slightly different way than before, *Equation 3*. Namely, we split the activity vector $\mathbf{x}$ into its component along the leading PCs, $\mathbf{x}_{\text{lead}}$, and the remaining, trailing component, $\mathbf{x}_{\text{trail}}$. By definition of the leading PCs as the directions of largest variance, the leading component is expected to be large, and the trailing one small. Inserting this decomposition $\mathbf{x} = \mathbf{x}_{\text{lead}} + \mathbf{x}_{\text{trail}}$ into *Equation 1* leads to

$$z = \mathbf{w}_{\text{out}}^T(\mathbf{x}_{\text{lead}} + \mathbf{x}_{\text{trail}}) = \|\mathbf{w}_{\text{out}}\| \left( \rho_{\text{lead}} \|\mathbf{x}_{\text{lead}}\| + \rho_{\text{trail}} \|\mathbf{x}_{\text{trail}}\| \right) , \tag{54}$$

with separately defined correlations $\rho_{\text{lead}} = \text{corr}(\mathbf{w}_{\text{out}}, \mathbf{x}_{\text{lead}})$ and $\rho_{\text{trail}} = \text{corr}(\mathbf{w}_{\text{out}}, \mathbf{x}_{\text{trail}})$.

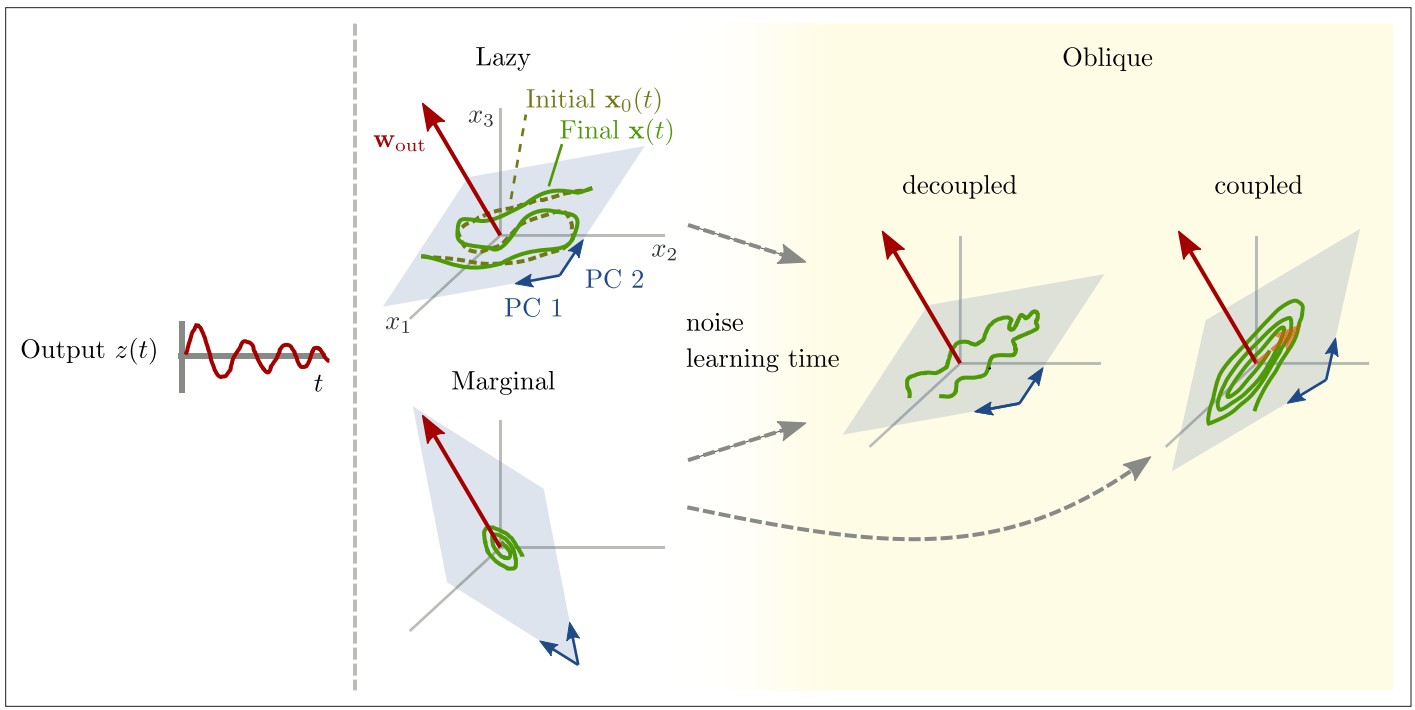

**Figure 14.** Path to oblique solutions for networks with large output weights. Left: All networks produce the same output (*Figure 1*). Center: Unstable solutions that arise early in learning. For lazy solutions, initial chaotic activity is slightly adapted, without changing the dynamics qualitatively. For marginal solutions, vanishingly small initial activity is replaced with very small dynamics sufficient to generate the output. Right: With more learning time and noise added during the learning process, stable, oblique solutions arise. The neural dynamics along the largest principal components (PCs) can be either decoupled from the output (center right) or coupled (right). For decoupled dynamics, the components along the largest PCs (blue subspace) differ qualitatively from those generating the output (same as *Figure 1B*, bottom). The dynamics along the largest PCs inherit task-unrelated components from the initial dynamics or randomness during learning. Another possibility are oblique, but coupled dynamics (right). Such solutions don't inherit task-unrelated components of the dynamics at initialization. They are qualitatively similar to aligned solutions, and the output is generated by a small projection of the output weights onto the largest PCs (dashed orange arrow).

For aligned networks, we recover the results from before: $\mathbf{w}_{\text{out}}$ is small so that the output can only be generated by the leading part with large correlation $\rho_{\text{lead}}$. The trailing part is unconstrained but is also not contributing to either the output or the leading dynamics, and hence not of interest.

For oblique networks, $\mathbf{w}_{\text{out}}$ is large so that the output can be generated by either of the two terms in *Equation 54*. The correlation $\rho_{\text{lead}}$ has to be small because else the output would be too large. The other correlation, $\rho_{\text{trail}}$, can be large, because non-dominant component $\mathbf{x}_{\text{trail}}$ is small. Both terms are potentially of the same magnitude, which means both can potentially contribute to the output. If the dominant part alone generates the output, then neural dynamics and output are coupled and the solution is similar to an aligned one (*Figure 14*, right). If, however, the non-dominant part alone generates the output, and the correlation $\rho_{\text{lead}}$ is so small that the dominant part does not contribute to the output, then the dominant part is not constrained by the task (*Figure 14*, center right). In that case, the dominant dynamics and the output can decouple qualitatively, and we may see the large variability between learners observed above.

The existence of decoupled solutions for oblique dynamics leads to the second question: Why and when do such solutions arise? Understanding this requires more detailed insights into the learning process. Roughly speaking, learning in the oblique regime has two opposite goals: first, to generate the desired output as fast as possible, and hence to induce changes to activity that are as small as possible (section Analysis of solutions under noiseless conditions); and second, to generate solutions that are robust and stable, and hence to induce changes in activity that are large enough to not be disrupted by noise (section Oblique solutions arise for noisy, nonlinear systems). During the process of learning, small, unstable solutions appear first (*Figure 14*, center). These may be highly variable, depending strongly on random initialization or other randomness experienced during learning. Such solutions then slowly solidify into stable solutions, that may inherit the variability of the early solutions (*Figure 14*, right).

The process of how learning transforms small, unstable solutions to larger, robust ones is analyzed in section Learning with noise for linear RNNs and Oblique solutions arise for noisy, nonlinear systems. The details of how this process introduces variability between learners, however, are not discussed there and left for future work.

# Additional information

## Competing interests

Srdjan Ostojic: Reviewing editor, eLife. The other authors declare that no competing interests exist.

## Funding

| Funder | Grant reference number | Author |
|--------|------------------------|--------|
| Deutsche Forschungsgemeinschaft | EXC 2002/1 "Science of Intelligence" - project no. 390523135 | Friedrich Schuessler |
| Israel Science Foundation | 1442/2 | Omri Barak |
| Human Frontier Science Program | RGP0017/2021 | Omri Barak |
| Agence Nationale de la Recherche | ANR-17-EURE-0017 | Srdjan Ostojic |

The funders had no role in study design, data collection and interpretation, or the decision to submit the work for publication.

## Author contributions

Friedrich Schuessler, Conceptualization, Software, Formal analysis, Investigation, Visualization, Methodology, Writing – original draft, Writing – review and editing; Francesca Mastrogiuseppe, Conceptualization, Writing – original draft, Writing – review and editing; Srdjan Ostojic, Conceptualization, Writing – original draft, Project administration, Writing – review and editing; Omri Barak, Conceptualization, Supervision, Funding acquisition, Writing – original draft, Project administration, Writing – review and editing

## Author ORCIDs

Friedrich Schuessler  https://orcid.org/0000-0002-6716-7492
Francesca Mastrogiuseppe  https://orcid.org/0000-0002-7682-5178
Srdjan Ostojic  https://orcid.org/0000-0002-7473-1223
Omri Barak  https://orcid.org/0000-0002-7894-6344

Reviewer #1 (Public review): https://doi.org/10.7554/eLife.93060.3.sa1
Reviewer #2 (Public review): https://doi.org/10.7554/eLife.93060.3.sa2
Author response https://doi.org/10.7554/eLife.93060.3.sa3

## Additional files

### Supplementary files
• MDAR checklist

### Data availability

The code to reproduce the results can be accessed via GitHub (copy archived at *Schuessler, 2024*).
No new data has been generated.

The following previously published dataset was used:

| Author(s) | Year | Dataset title | Dataset URL | Database and Identifier |
|---|---|---|---|---|
| Russo A | 2020 | Two-target cycling task, primate motor cortex | https://data.mendeley.com/datasets/tfcwp8bp5j/1 | Mendeley Data, 10.17632/tfcwp8bp5j.1 |

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

# Appendix 1

## Supplementary information

### A.1. Linear approximation for lazy learning

We numerically investigated the linearization, *Equation 17*, of the output trajectory $z(t)$ in weight changes $\Delta W$, *Equation 17*. For this, we used the sine wave task and the same configuration as for the lazy network in *Figure 9C*. In *Appendix 1—figure 1A*, we plot the output of the full network after inserting the linear solution $\Delta W_{\text{lin}}$. The fit to the training points got more accurate with increasing network size. However, it did not reach zero error for the networks shown. In general, the error increased with increasing trial time. This indicates that the linear approximation was not valid. This is in line with the nature of the dynamical system: Trajectories become more nonlinear in $W$ with increasing time. The loss on the training set (*Appendix 1—figure 1C*) summarizes the observations seen before: The linear solution became more accurate with increasing network size.

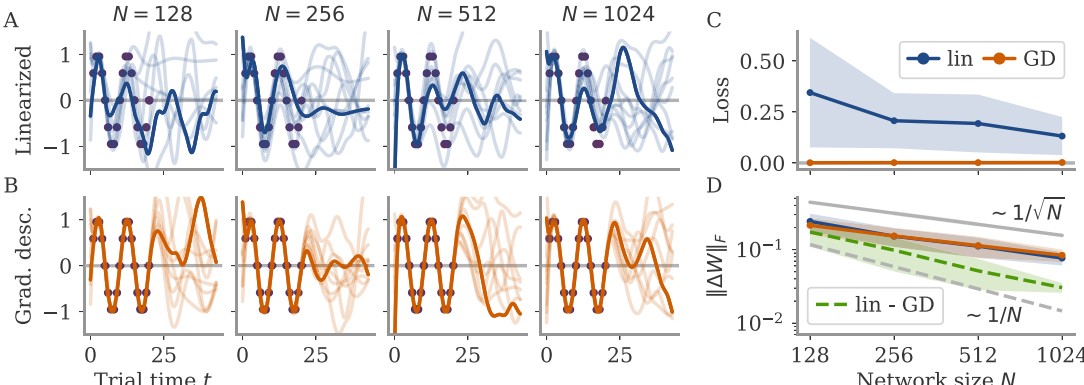

**Appendix 1—figure 1.** Solution to linearized network dynamics in the lazy regime. (**A**) Network output for weight changes $\Delta W_{\text{lin}}$ obtained from linearized dynamics for different network sizes. Each plot shows 10 different networks (one example in bold). Target points in purple. (**B**) Output for networks trained with gradient descent (GD) from the same initial conditions as those above. (**C**) Loss on the training set for the linear (lin) and GD solutions. (**D**) Frobenius norms of weight changes $\Delta W$ of the linear and GD solutions, as well as of the difference between the two. $\Delta W_{\text{lin}} - \Delta W_{\text{GD}}$ (dashed green). Gray and black dashed lines for comparison of scales.

How close was the linear solution to the one found by GD? To test this, we optimized the network with GD starting the same initial condition $W_0$. The output of this network is shown in orange in *Appendix 1—figure 1B*. As seen before (*Figure 9C*) training points were met, but the dynamics after the last training points remained chaotic. We then quantified the norms of the two solutions and the difference between both. As shown in *Appendix 1—figure 1D*, both solutions had similar norm, which decays as $1/\sqrt{N}$. The difference decayed faster, as $1/N$. This indicates that the linearized system can give a good insight into the solution found by training with GD, despite the inaccuracies of the linear approximation. In particular, the main features (scaling, unstable solutions, no extrapolation) are the same for the linear and the gradient descent solutions.

### A.2 Details of linear model: bias only: lazy learning

We consider learning for the linear, input-driven model, *Equation 4*.

We start with the bias part, which is equivalent to the noise-free model (*Schuessler et al., 2020b*). Disregarding initial transients, the average output is $\bar{z} = \mathbf{w}_{\text{out}}^T B \mathbf{w}_{\text{in}}$, with $B = (\mathbb{1} - W)^{-1}$. The gradient of the loss $L_{\text{bias}} = (\bar{z} - \hat{z})^2$ is then

$$G_{\text{bias}} = \frac{dL_{\text{bias}}}{dW} = 2B^T \mathbf{w}_{\text{out}}(\bar{z} - \hat{z})\mathbf{w}_{\text{in}}^T B^T. \tag{55}$$

We make the ansatz of small weight changes and hence linearize the output in weights (like in feedforward networks, *Liu et al., 2020*):

$$\bar{z} = \bar{z}_0 + G_{z,0} \cdot \Delta W + O(\Delta W^2), \tag{56}$$

with $G_{z,0}$ the gradient of the output at initialization,

$$G_{z,0} = \frac{\mathrm{d}z}{\mathrm{d}W}\bigg|_0 = B_0^T \mathbf{w}_{\text{out}} \mathbf{w}_{\text{in}}^T B_0^T, \tag{57}$$

and $B_0 = (\mathbb{1} - W_0)^{-1}$. The dot-product signifies sum over all entries, $A \cdot B = \sum_{ij} A_{ij} B_{ij}$ for two matrices $A, B$. Because the output is linear, the weight changes are spanned by the gradient, $\Delta W = b(\tau) G_{z,0} / \|G_{z,0}\|$. Insertion into gradient descent dynamics yields

$$\dot{b}(\tau) = -2\eta m \left[ \bar{z}_0 - \hat{z} - b(\tau)m \right], \tag{58}$$

with

$$m = \|G_{z,0}\| = \|B_0^T \mathbf{w}_{\text{out}}\| \|B_0^T \mathbf{w}_{\text{in}}\| = \frac{\|\mathbf{w}_{\text{out}}\| \|\mathbf{w}_{\text{in}}\|}{1 - g^2} = \frac{\sqrt{N}}{1 - g^2}. \tag{59}$$

We used that $\|B_0 \mathbf{v}\|^2 = \|B_0^T \mathbf{v}\|^2 = \frac{1}{1-g^2} \|\mathbf{v}\|^2$ for any vector $\mathbf{v}$ independent of $W_0$ (which is the case for the input and output vectors) (*Schuessler et al., 2020b*). For initial condition $b(0) = 0$, the solution to *Equation 58* is

$$b(\tau) = \frac{\hat{z} - z_0}{m} (1 - e^{-2m^2 \eta \tau}). \tag{60}$$

The corresponding output changes are $\Delta \bar{z} = \bar{z} - \bar{z}_0 = b(\tau)m$. Because $m^2 \sim N$, the output converges to the target in $O(\eta N)$ learning steps. The resulting weights are

$$\Delta W = \frac{(\hat{z} - \bar{z}_0)(1 - g^2)^2}{\sqrt{N}} B_0^T \hat{\mathbf{w}}_{\text{out}} \hat{\mathbf{w}}_{\text{in}}^T B_0^T, \tag{61}$$

where the hats indicate normalized vectors. Consistent with linearization, the norm (both Frobenius and operator) of the changes is small: $\|\Delta W\| = \|\Delta W\|_2 = (\hat{z} - \bar{z}_0)(1 - g^2)/\sqrt{N} \sim 1/\sqrt{N}$.

To connect with the following paragraph, we consider the case $g = 0$, so that $\Delta W = W$. This is for reference, and *Figure 10*; see details below. We express the results in the orthonormalized bases introduced below:

$$\Delta W(\tau) = \frac{b_1(\tau)}{\sqrt{N}} \hat{\mathbf{w}}_{\text{out}} \hat{\mathbf{w}}_{\text{in},\perp}^T + O\left(\frac{1}{N}\right), \tag{62}$$

with

$$b_1(\tau) = (1 - e^{-2N\eta\tau})(\hat{z} - \sqrt{N}\rho_{io}). \tag{63}$$

The corresponding activity changes are $\Delta \bar{\mathbf{x}} = b_1(\tau) \hat{\mathbf{w}}_{\text{out}}$, with norm $\|\Delta \bar{\mathbf{x}}\| = b_1(\tau)$. The output changes are $\Delta z = b_1(\tau)$, so that the loss is $L_{\text{bias}} = [b_1(\tau) + \sqrt{N}\rho_{io} - \hat{z}]^2 = e^{-4N\eta\tau}(\hat{z} - \sqrt{N}\rho_{io})^2$.

## A.3 Details of linear model: bias and variance combined

We now combine results from the previous section, concerning the bias, with results from the Methods, concerning the variance, to derive the full learning dynamics of the linear model. For the case of no initial connectivity, $W_0 = 0$, the gradients for both the bias and the variance component, *Equations 23 and 55*, are spanned by the output and input vector. By orthonormalizing, we define $\hat{U} = [\hat{\mathbf{w}}_{\text{out}}, \hat{\mathbf{w}}_{\text{in},\perp}]$ with $\hat{U}^T \hat{U} = \mathbb{1}_2$. The weights are then constrained to $W = \hat{U} M \hat{U}^T$. For ease of writing, we define the vectors $\mathbf{v}_{\text{out}} = \hat{U}^T \hat{\mathbf{w}}_{\text{out}} = [1, 0]^T$ and $\mathbf{v}_{\text{in}} = \hat{U}^T \hat{\mathbf{w}}_{\text{in}} = [\rho_{io}, \sqrt{1 - \rho_{io}^2}]^T$, where $\rho_{io} = \hat{\mathbf{w}}_{\text{out}}^T \hat{\mathbf{w}}_{\text{in}} \sim \mathcal{N}(0, 1/N)$ is the random, $O(\frac{1}{\sqrt{N}})$ correlation between input and output vector. As before, the norms of input and output vectors are $\|\mathbf{w}_{\text{in}}\| = \sqrt{N}$ and $\|\mathbf{w}_{\text{out}}\| = \sigma_{\text{out}}$.

The projection of the deterministic gradient is then

$$G_{\text{det},\hat{U}} = \hat{U}^T G_{\text{det}} \hat{U} = 2(\bar{z} - \hat{z})\sigma_{\text{out}} B_{\hat{U}}^T \mathbf{v}_{\text{out}} \mathbf{v}_{\text{in}}^T B_{\hat{U}}^T, \tag{64}$$

with deterministic output

$$\bar{z} = \sigma_{\text{out}} \mathbf{v}_{\text{out}}^T B_{\hat{U}} \mathbf{v}_{\text{in}} , \tag{65}$$

and projection $B_{\hat{U}} = \hat{U}^T B \hat{U} = (1 - M)^{-1}$.

For the stochastic part, we need to solve the two Lyapunov *Equations 21 and 24*. They reduce to low-D versions after projection. For the first one, we have

$$\Sigma = \sigma_{\text{noise}}^2 \left[ \mathbb{1} 1_N + \hat{U}(-\mathbb{1}_2 + \Sigma_{\hat{U}})\hat{U}^T \right] , \tag{66}$$

and the projection solves the 2D Lyapunov equation

$$A_{\hat{U}} \Sigma_{\hat{U}} + \Sigma_{\hat{U}} A_{\hat{U}}^T + 2\mathbb{1}_2 = 0 , \tag{67}$$

with $A_{\hat{U}} = -\mathbb{1}_2 + M$. Similarly, the low-D version of the second Lyapunov *Equation 24* is obtained by defining $\Omega = \sigma_{\text{out}}^2 \hat{U} \Omega_{\hat{U}} \hat{U}^T$, with

$$A_{\hat{U}} \Omega_{\hat{U}} + \Omega_{\hat{U}} A_{\hat{U}}^T + \mathbf{v}_{\text{out}} \mathbf{v}_{\text{out}}^T = 0 . \tag{68}$$

The full learning dynamics for gradient flow are then given by

$$\dot{M} = -\eta \left( G_{\text{det}, \hat{U}} + G_{\text{sto}, \hat{U}} \right) , \tag{69}$$

with $M(0) = 0$ and

$$G_{\text{sto}, \hat{U}} = \hat{U}^T G_{\text{sto}} \hat{U} = 2\sigma_{\text{noise}}^2 \sigma_{\text{out}}^2 \Omega_{\hat{U}} \Sigma_{\hat{U}} . \tag{70}$$

For a small output scale, the deterministic part is order one, while the stochastic one is order $1/N$ (through $\Omega$). Assuming $\eta = O(1)$, learning thus reaches an arbitrary small order-one loss after $O(1)$ updates, with noise adding a $1/N$ stochastic loss, and noise-driven learning equally adding order $1/N$ deviations to the entries of $M$. Hence, learning remains unaffected by the noise unless it is continued for $O(\eta N^2)$ steps.

For a large output scale, the deterministic part is fast but only leads to $1/\sqrt{N}$ entries. In such a situation, the stochastic part can be expanded in orders of $\epsilon = 1/\sqrt{N}$. To leading order, it yields dynamics along the first entry of $M$.

In detail: For a large readout, $\sigma_{\text{out}} = 1$, we expand $M$ to first order in $\epsilon = \frac{1}{\sqrt{N}}$:

$$M = \begin{bmatrix} a_0 & b_0 \\ c_0 & d_0 \end{bmatrix} + \epsilon \begin{bmatrix} a_1 & b_1 \\ c_1 & d_1 \end{bmatrix} . \tag{71}$$

The leading order of the output is

$$\bar{z} = \frac{1}{\epsilon} \frac{b_0}{1 - a_0 + a_0 d_0 - b_0 c_0 - d_0} + \mathcal{O}(1) , \tag{72}$$

which implies that $b_0 = 0$ throughout learning. Under this assumption, we have

$$\bar{z} = \frac{b_1 + (1 - d_0)\sqrt{N}\rho_{io}}{(1 - a_0)(1 - d_0)} + \mathcal{O}(\epsilon) . \tag{73}$$

Furthermore, the deterministic gradient is

$$G_{\text{det}, \hat{U}} = \frac{2}{\epsilon}(\bar{z} - \hat{z}) \begin{bmatrix} 0 & -\dfrac{\hat{z}}{(1 - a_0)(1 - d_0)} \\ 0 & 0 \end{bmatrix} + \mathcal{O}(1) . \tag{74}$$

This component of learning converges on the order of $\eta/\epsilon^2$ learning steps due to the prefactor $1/\epsilon$ and the fact that $b_1$ enters $\bar{z}$ at order one. The higher order terms therefore only lead to $O(\epsilon^2)$ corrections.

The stochastic part of the gradient simplifies by assuming $b_0 = c_0 = d_0 = 0$. The latter two are justified self-consistently because neither the deterministic nor the stochastic part contribute to $c$ and $d$ at order one if we start with $M = 0$. With this assumption, we have

$$G_{\text{sto},\hat{U}} = \sigma_{\text{noise}}^2 \frac{1}{(1 - a_0)^2} \left( \begin{bmatrix} 1 & 0 \\ 0 & 0 \end{bmatrix} + \epsilon \begin{bmatrix} \dfrac{2a_1}{1 - a_0} & \dfrac{b_1(1 - a_0) + c_1(2 - a_0)}{2 - a_0} \\ \dfrac{c_1}{2 - a_0} & 0 \end{bmatrix} + \mathcal{O}(\epsilon^2) \right) . \tag{75}$$

Consistent with our assumption, we only have order-one growth for the entry $a$. The resulting dynamics are

$$a_0(\tau) = 1 - (3\eta\sigma_{\text{noise}}^2\tau + 1)^{\frac{1}{3}} , \tag{76}$$

where $\tau$ is the number of learning steps. Further, because the deterministic part is so much faster, we can assume that the second entry, $b = b_1$, is always constrained by the target. Namely, from *Equation 73*, we have

$$b_1 = \hat{z}(1 - a_0) - \sqrt{N}\rho_{io} . \tag{77}$$

Lastly, the first-order corrections for the other entries all vanish due to vanishing initial conditions:

$$a_1 = c_1 = d_1 = 0 . \tag{78}$$

The loss due to the stochastic part is then

$$L_{\text{var}} = \sigma_{\text{noise}}^2 (3\eta\sigma_{\text{noise}}^2\tau + 1)^{-\frac{1}{3}} , \tag{79}$$

while the deterministic part is kept at $\mathcal{O}(1/N)$.

During learning, the average states change only very little. To see this, we express the connectivity as $W = \hat{w}_{\text{out}}\mathbf{w}_r^T$ with

$$\mathbf{w}_r = \lambda_-\hat{w}_{\text{out}}^T + \frac{b_1}{\sqrt{N}}\hat{w}_{\text{in},\perp} , \tag{80}$$

and orthonormalized input weights

$$\hat{w}_{\text{in},\perp} = \frac{\hat{w}_{\text{in}} - \hat{w}_{\text{out}}\rho_{io}}{\sqrt{1 - \rho_{io}^2}} = \hat{w}_{\text{in}} - \hat{w}_{\text{out}}\rho_{io} + O(1/N) . \tag{81}$$

The output is then

$$\begin{aligned} \bar{\mathbf{x}} &= (1 - W)^{-1}\mathbf{w}_{\text{in}} \\ &= \left( \mathbb{1} + \frac{\hat{w}_{\text{out}}\mathbf{w}_r^T}{1 - a_0} \right)\mathbf{w}_{\text{in}} \\ &= \mathbf{w}_{\text{in}} + \hat{w}_{\text{out}}\frac{a_0\rho_{io}\sqrt{N} + b_1}{1 - a_0} . \end{aligned} \tag{82}$$

The initial states are $\bar{\mathbf{x}}_0 = \mathbf{w}_{\text{in}}$, so the changes are

$$\Delta\bar{\mathbf{x}} = \frac{a_0\rho_{io}\sqrt{N} + b_1}{1 - a_0}\hat{w}_{\text{out}} = (\hat{z} - \rho_{io}\sqrt{N})\hat{w}_{\text{out}} , \tag{83}$$

where the scalar part is also the norm. Thus, $\|\Delta\bar{\mathbf{x}}\| \sim 1$, which is small, because the entries are $O(1/\sqrt{N})$. Paradoxically, the norm does not change with learning time. Essentially, once the lazy learning takes place at the beginning of learning, the norm does not change anymore (to leading order).

## A.4 Details of nonlinear autonomous system with noise

This section contains detailed derivations for the average and fluctuations of the latent projection $\kappa$. The theoretical treatment of the network dynamics is very similar to that for a network with only a

negative feedback loop for 'balance' by *Kadmon et al., 2020*. What is added here is a colored noise term that enters via the variance $\sigma_\perp^2$ of the orthogonal dynamics. This is dependent on a nonzero fixed point (not present in *Kadmon et al., 2020*).

The dynamics of the latent variable are

$$\dot{\boldsymbol{\kappa}} = -\boldsymbol{\kappa} + M\frac{1}{N}U^T\phi(U\boldsymbol{\kappa} + \mathbf{x}_\perp) + \frac{1}{N}U^T\boldsymbol{\xi}\,. \tag{84}$$

That is, the low-D variable is driven autonomously by itself, but externally by the high-D orthogonal part $\mathbf{x}_\perp$. To resolve this into low-D dynamics only, we apply two averages: the trial-conditioned average, which is over the noise $\boldsymbol{\xi}$, and denoted by the bar, $\bar{\cdot} = \mathbb{E}_\xi[\cdot]$; and the average over the statistics of $U$, which arises naturally from the projection on $U$ (a population average). We start with the latter, writing

$$\frac{1}{N}U^T\phi(\mathbf{x}) = \frac{1}{N}\mathbb{E}_U[U^T\phi(\mathbf{x})] + \boldsymbol{\epsilon}\,. \tag{85}$$

One can show that the error is small, $\mathbb{E}_U[\epsilon_a] = 0$ and $\mathbb{E}_U[\epsilon_a\epsilon_b] \sim 1/N$. For the average, we apply partial integration as in the noise-free case, *Equation 14*, and arrive at

$$\frac{1}{N}\mathbb{E}_U[U^T\phi(\mathbf{x})] = \langle\phi'(\sigma_x u)\rangle\boldsymbol{\kappa}\,, \tag{86}$$

where the average on the right-hand side is over the standard normal variable $u$. In contrast to the noise-free case, the variance is now comprised of two terms,

$$\sigma_x(t)^2 = \|\boldsymbol{\kappa}(t)\|^2 + \sigma_\perp(t)^2\,, \tag{87}$$

and $\sigma_\perp^2(t) = \frac{1}{N}\mathbf{x}_\perp(t)^T\mathbf{x}_\perp(t)$ is the population average of $\mathbf{x}_\perp(t)$. We included the time index here to emphasize that both variables still contain fluctuations around the trial-conditioned average.

We now split the term (*Equation 85*) into average and fluctuations around this. We self-consistently assume that all fluctuations, denoted with a $\delta$, are small, which allows us to linearize around the respective averages. We have

$$\frac{1}{N}U^T\phi(\mathbf{x}) = \overline{\langle\phi'(\sigma_x u)\rangle}\,\bar{\boldsymbol{\kappa}} + \bar{\boldsymbol{\epsilon}} + \overline{\langle\phi'(\sigma_x u)\rangle}\,\delta\boldsymbol{\kappa} + \delta\langle\phi'(\sigma_x u)\rangle\,\bar{\boldsymbol{\kappa}}\,. \tag{88}$$

We only keep terms up to order $O(1/\sqrt{N})$. The fluctuations of the error are $\delta\epsilon \sim 1/N$ and hence discarded. By linearization, we further have $\overline{\langle\phi'(\sigma_x u)\rangle} = \langle\phi'(\overline{\sigma_x}u)\rangle$, with

$$\overline{\sigma_x^2} = \|\bar{\boldsymbol{\kappa}}\|^2 + \overline{\sigma_\perp^2}\,. \tag{89}$$

The orthogonal part is an Ornstein-Uhlenbeck process with steady-state covariance

$$\overline{x_{\perp i}(t)x_{\perp j}(t+s)} = \delta_{ij}\overline{\sigma_\perp^2}\,e^{-|s|}\,, \tag{90}$$

where the variance is inherited from the white noise, $\overline{\sigma_\perp^2} = \sigma_{\text{noise}}^2$.

We take the average over the latent dynamics *Equation 84*. Apart from an initial transient, which we discard, this leaves us with a fixed point

$$\bar{\boldsymbol{\kappa}} = \langle\phi'(\overline{\sigma_x}u)\rangle M\bar{\boldsymbol{\kappa}} + \bar{\boldsymbol{\epsilon}}\,. \tag{91}$$

We now discuss the fluctuations in *Equation 88*. There are two sources of fluctuations, $\delta\boldsymbol{\kappa}$ and $\delta\sigma_\perp^2$, so that we have

$$\delta\langle\phi'\rangle = \partial_{\sigma_x}\langle\phi'\rangle\left(\partial_{\boldsymbol{\kappa}}^T\sigma_x\,\delta\boldsymbol{\kappa} + \partial_{\sigma_\perp^2}\sigma_x\,\delta\sigma_\perp^2\right)\,. \tag{92}$$

All derivatives are evaluated at the averages (we discard the bar in the following lines to keep notation at bay). We further compute

$$\partial_{\sigma_x}\langle\phi'(\sigma_x u)\rangle = \langle\phi''(\sigma_x u)u\rangle = \sigma_x\langle\phi'''(\sigma_x u)\rangle\,, \tag{93}$$

$$\partial_{\boldsymbol{\kappa}} \sigma_x = \frac{1}{\sigma_x} \boldsymbol{\kappa} \,, \tag{94}$$

$$\partial_{\sigma_{\perp}^2} \sigma_x = \frac{1}{2\sigma_x} \,. \tag{95}$$

Inserting these terms into the dynamics (**Equation 84**), we arrive at

$$\frac{\mathrm{d}\,\delta\boldsymbol{\kappa}(t)}{\mathrm{d}t} = \left(-\mathbb{1} + \langle\phi'\rangle M + \langle\phi'''\rangle M\bar{\boldsymbol{\kappa}}\bar{\boldsymbol{\kappa}}^T\right)\delta\boldsymbol{\kappa}(t) + \frac{\langle\phi'''\rangle}{2} M\bar{\boldsymbol{\kappa}}\,\delta\sigma_{\perp}^2(t) + \frac{1}{N}U^T\boldsymbol{\xi}\,. \tag{96}$$

The white noise term has variance

$$\frac{1}{N^2}\overline{U^T\boldsymbol{\xi}\boldsymbol{\xi}^T U} = \frac{2\sigma_{\text{noise}}^2}{N}\mathbb{1}_2\,. \tag{97}$$

For the fluctuations in the population average of the variance, we use **Equation 90** to compute the covariance function between different time points,

$$\begin{aligned}\overline{\delta\sigma_{\perp}^2(t)\delta\sigma_{\perp}^2(t+s)} &= \frac{1}{N^2}\sum_{ij}\overline{x_{\perp i}(t)^2 x_{\perp j}(t+s)^2} - (\overline{\sigma_{\perp}^2})^2 \\ &= (\overline{\sigma_{\perp}^2})^2\left(\frac{N(N-1)}{N^2} - 1\right) + \frac{1}{N}\sum_i\overline{x_{\perp i}(t)^2 x_{\perp}(t+s)^2} \\ &= \frac{\sigma_{\text{noise}}^4}{N}\left[-1 + \langle u^2\,(e^{-|s|}u + \sqrt{1 - e^{-2|s|}}u')^2\rangle\right] \\ &= \frac{2\sigma_{\text{noise}}^4}{N}e^{-2|s|}\,.\end{aligned} \tag{98}$$

For the third line, the average is taken over the independent standard normal variables $u, u'$. Putting all the pieces together, we have

$$\frac{\mathrm{d}\,\delta\boldsymbol{\kappa}(t)}{\mathrm{d}t} = A\delta\boldsymbol{\kappa}(t) + \frac{1}{\sqrt{N}}\boldsymbol{\zeta}(t)\,, \tag{99}$$

with Jacobian

$$A = -\mathbb{1} + \langle\phi'\rangle M + \frac{\langle\phi'''\rangle}{\langle\phi'\rangle}\bar{\boldsymbol{\kappa}}\bar{\boldsymbol{\kappa}}^T\,, \tag{100}$$

and noise term $\boldsymbol{\zeta}(t)$ with zero average and covariance

$$\overline{\boldsymbol{\zeta}(t)\boldsymbol{\zeta}(t+s)^T} = \left(2\sigma_{\text{noise}}^2\delta(s)\mathbb{1}_2 + \beta e^{-2|s|}\bar{\boldsymbol{\kappa}}\bar{\boldsymbol{\kappa}}^T\right)\,, \tag{101}$$

with

$$\beta = \frac{\sigma_{\text{noise}}^4}{2}\left(\frac{\langle\phi'''\rangle}{\langle\phi'\rangle}\right)^2\,. \tag{102}$$

We inserted the eigenvalue equation, $M\bar{\boldsymbol{\kappa}} = \frac{1}{\langle\phi'\rangle}\bar{\boldsymbol{\kappa}}$. We note that the colored noise term did not appear in previous studies such as **Kadmon et al., 2020**, because there was no positive feedback and corresponding fixed point $\bar{\boldsymbol{\kappa}}$.

We now compute the variance of the fluctuations $\delta\boldsymbol{\kappa}$. The computation is simplified by the fact the $\bar{\boldsymbol{\kappa}}$ is also eigenvector of $A$, with eigenvalue

$$\gamma_+ = \frac{\langle\phi'''\rangle}{\langle\phi'\rangle}\|\bar{\boldsymbol{\kappa}}\|^2\,. \tag{103}$$

The eigenvector is also the direction in which the colored noise acts. Because of this, we can write

$$\overline{\delta\boldsymbol{\kappa}(t)\delta\boldsymbol{\kappa}(t)^T} = \frac{1}{N}\int_0^t dt' \int_0^t dt'' e^{A(t-t')}\overline{\boldsymbol{\zeta}(t')\boldsymbol{\zeta}(t'')^T}e^{A^T(t-t')}$$

$$= \frac{1}{N}\left[\sigma_{\text{noise}}^2 \Sigma_A(t) + \beta\bar{\boldsymbol{\kappa}}\bar{\boldsymbol{\kappa}}^T \underbrace{\int_0^t dt' \int_0^t dt'' e^{\gamma_+(2t-t'-t'')-2|t'-t''|}}_{I(t)}\right].$$
(104)

The first summand stems from the white noise term,

$$\Sigma_A(t) = 2\int_0^t dt' e^{A(t-t')}e^{A^T(t-t')}.$$
(105)

The integral $I(t)$ can be computed to yield

$$I(t) = \frac{e^{2\gamma_+ t}}{\gamma_+(2-\gamma_+)}\left[1 - e^{-2\gamma_+ t} - \frac{2\gamma_+}{2+\gamma_+}\left(1 - e^{-(2+\gamma_+)t}\right)\right].$$
(106)

For the steady-state variance, we take $t \to \infty$. The first term yields $\lim_{t\to\infty}\Sigma_A(t) = \Sigma_A$, which can be expressed as the solution of the Lyapunov equation

$$0 = A\Sigma_A + \Sigma_A A^T + 2\mathbb{1}_2.$$
(107)

The integral converges to

$$\lim_{t\to\infty} I(t) = \frac{1}{-\gamma_+(2-\gamma_+)}.$$
(108)

Joining all the bits and pieces, we have the steady-state covariance

$$\overline{\delta\boldsymbol{\kappa}\delta\boldsymbol{\kappa}^T} = \frac{1}{N}\left[\sigma_{\text{noise}}^2 \Sigma_A + \frac{\sigma_{\text{noise}}^4}{\|\bar{\boldsymbol{\kappa}}\|^2}\frac{-\gamma_+}{2(2-\gamma_+)}\mathbf{v}_+\mathbf{v}_+^T\right],$$
(109)

with normalized eigenvector

$$\mathbf{v}_+ = \frac{\bar{\boldsymbol{\kappa}}}{\|\bar{\boldsymbol{\kappa}}\|}.$$
(110)

For the loss, we only need to consider the variance along the output vector. Namely, we have the output fluctuations

$$\delta z = \sqrt{N}\|\mathbf{w}_{\text{out}}\|\mathbf{v}_{\text{out}}^T\delta\boldsymbol{\kappa},$$
(111)

with projected output weights

$$\mathbf{v}_{\text{out}} = \frac{1}{\sqrt{N}}U^T\mathbf{w}_{\text{out}}/\|\mathbf{w}_{\text{out}}\|.$$
(112)

Thus,

$$L_{\text{var}} = \overline{\delta z^2} = N\mathbf{v}_{\text{out}}^T\overline{\delta\boldsymbol{\kappa}\delta\boldsymbol{\kappa}^T}\mathbf{v}_{\text{out}}.$$
(113)

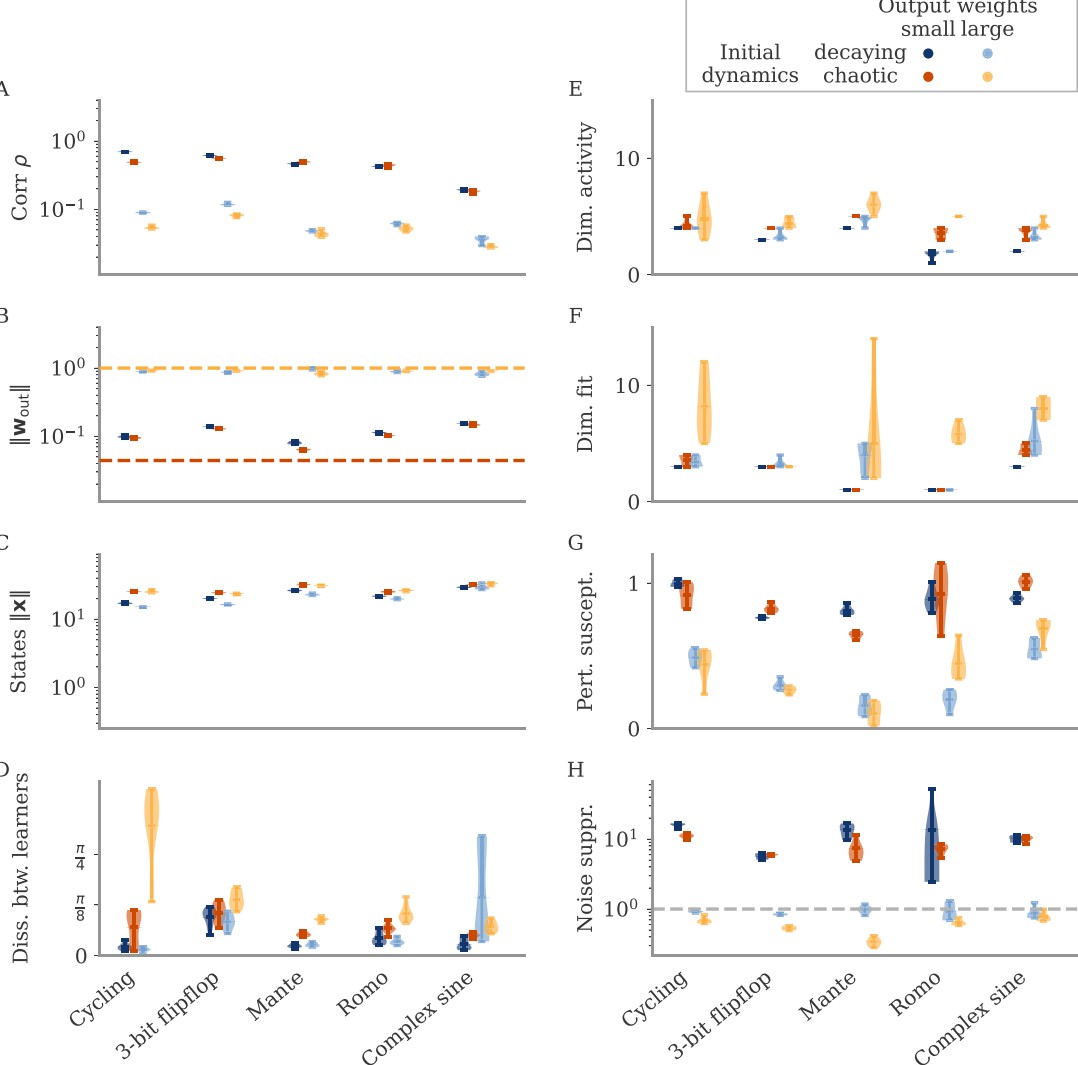

**Appendix 1—figure 2.** Summary of all measures for initially decaying or chaotic networks. (**A**) Correlation, (**B**) norm of output weights, (**C**) norm of states, (**D**) dissimilarity between learners, (**E**) dimension of activity $D_{x,90}$, (**F**) fit dimension $D_{\mathrm{fit},90}$, (**G**) susceptibility to perturbations, (**H**) noise suppression.

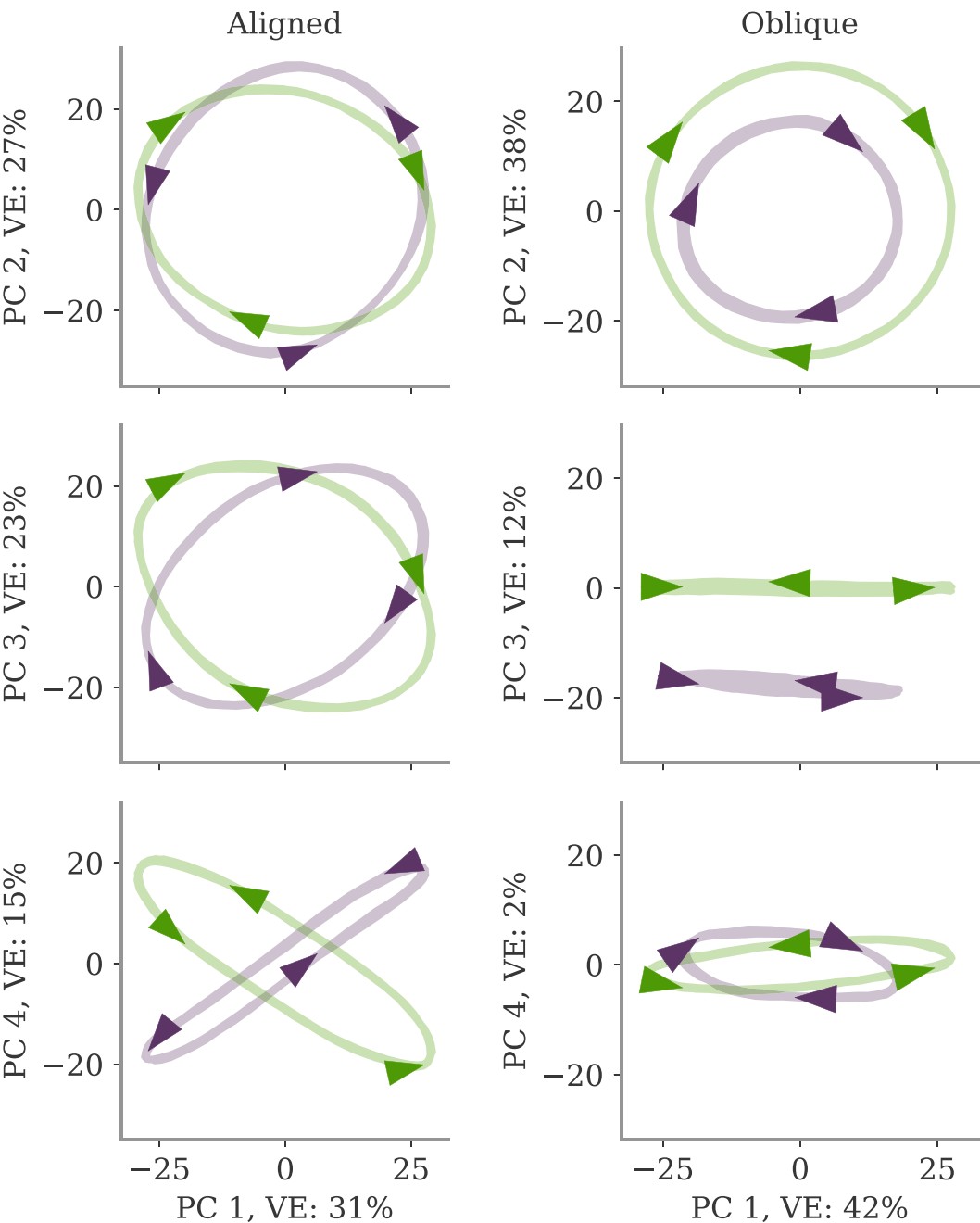

**Appendix 1—figure 3.** Projection of neural dynamics onto first four principal components (PCs) for the cycling task, *Figure 2*. The *x*-axis for all plots is PC 1 of the respective dynamics (left: aligned, right: oblique). The *y*-axes are PCs 2–4. Axis labels indicate the relative variance explained by each PC. Arrows indicate direction. Note that there is co-rotation for the aligned network for PCs 1 and 3, as well as the counter-rotation for the oblique network for PCs 1 and 4.

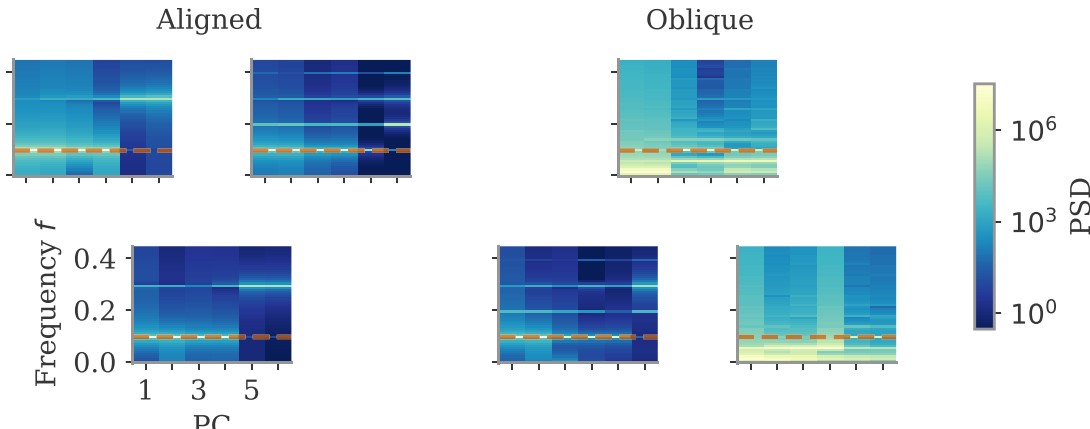

**Appendix 1—figure 4.** Power spectral densities for the six networks shown in *Figure 4*. The dashed orange line indicates the output frequency. Note the high power for non-target frequencies in the first principal components (PCs) in some of the large output solutions.

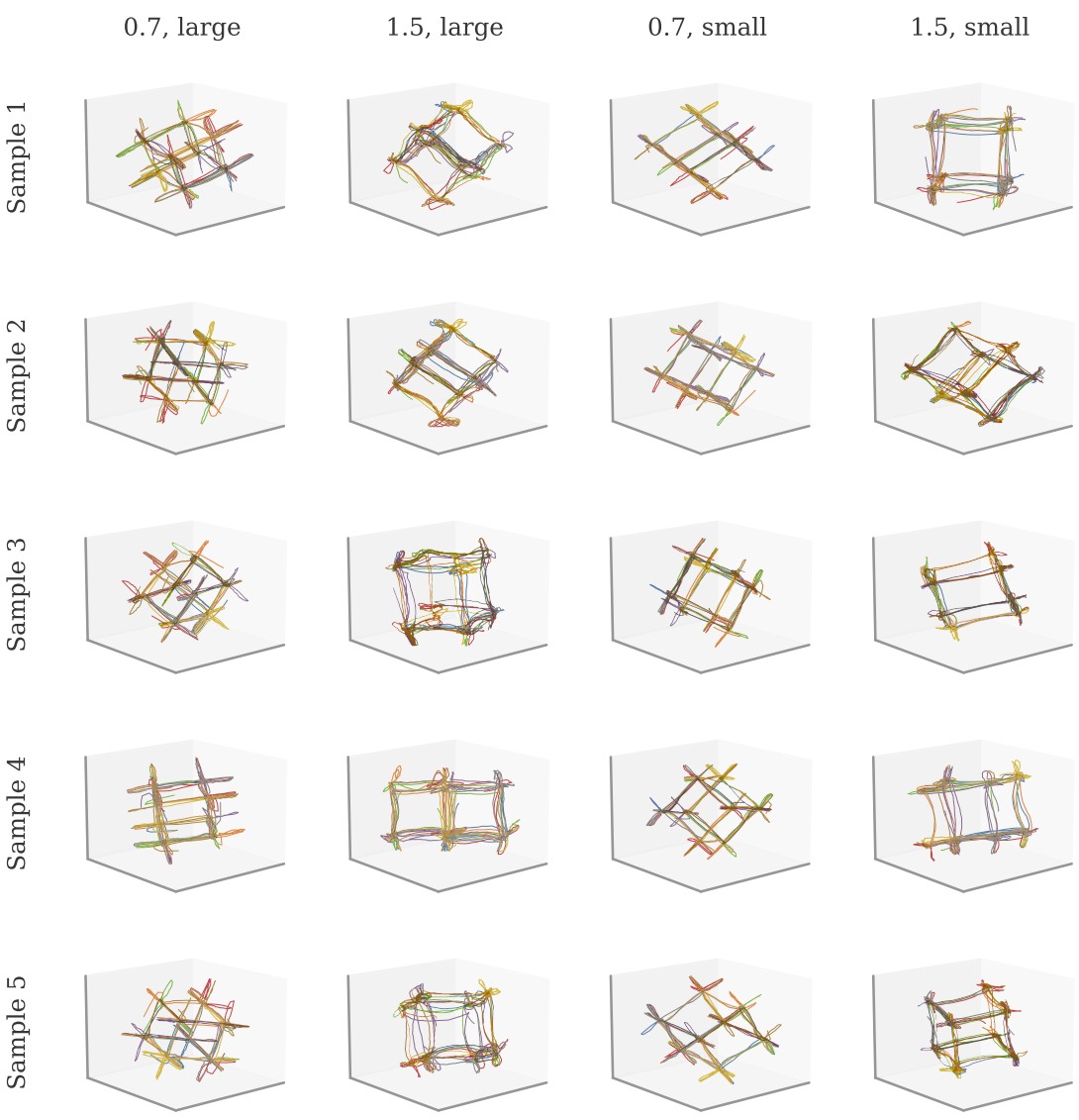

**Appendix 1—figure 5.** Example of task variability for the flip-flop task. The titles in each row indicate the spectral radius $g$ of the initial recurrent connectivity ($g > 1$ for initially chaotic, else decaying, activity), and the norm of initial output weights.

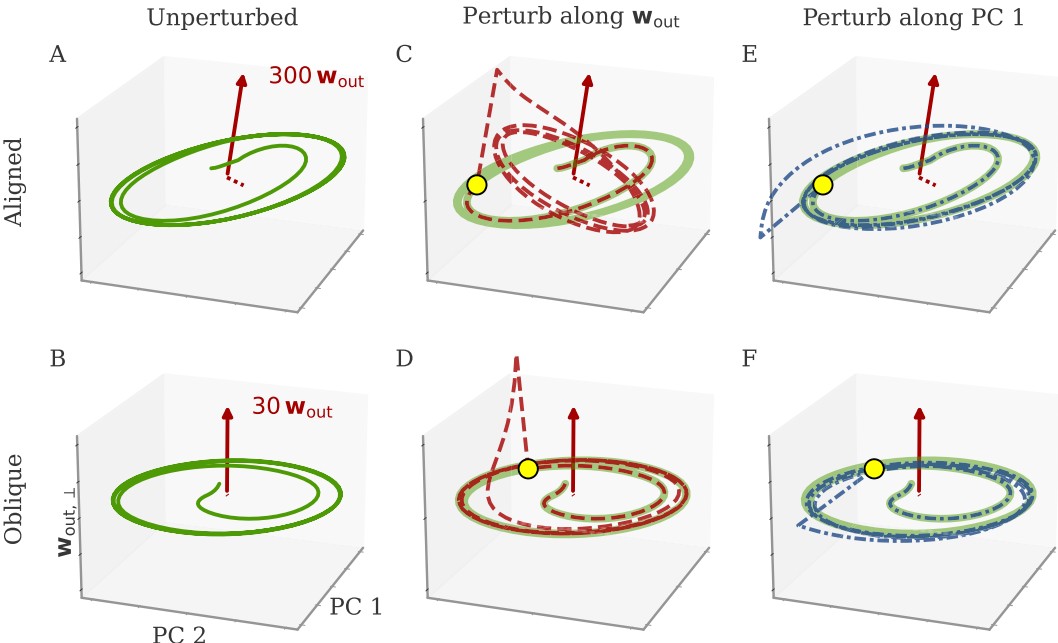

**Appendix 1—figure 6.** Neural activity in response to the perturbations applied in *Figure 5*. Activity is plotted in the space spanned by the leading two principal components (PCs) and the output weights $w_{\text{out},1}$. We first show the unperturbed trajectories in each network (**A, B**), then the perturbed ones for perturbations along the first output direction (**C, D**) and along the first PC (**E, F**). The unperturbed trajectories are also plotted for comparison. Yellow dots indicate the point where the perturbation is applied. All perturbations but the one along the output for aligned lead to trajectories on the same attractor, but potentially with a phase shift. Note that in general, perturbations can also lead to the activity converging on a different attractor. Here, we see a specific example of this happening for the cycling task in the aligned regime.

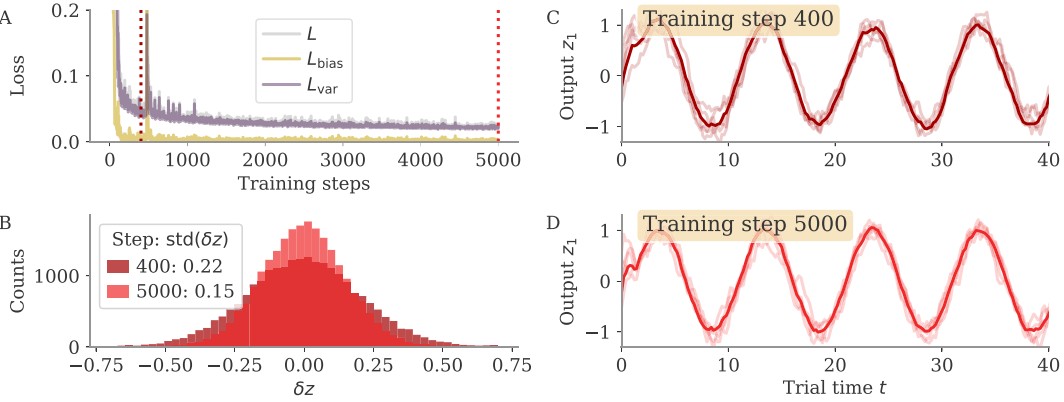

**Appendix 1—figure 7.** Noise compression over training time for the cycling task. Example networks trained on the cycling tasks with $\sigma_{\text{noise}} = 0.2$ and $N = 256$. The network at the end of training is analyzed in *Figure 6B*. (**A**) Full loss (gray) and decomposition in bias (golden) and variance (purple) parts over learning time. The bias part decays rapidly (the *y*-axis is clipped, initial loss $L_0 = 1.4$), whereas the variance part needs many more training steps to decrease. Dotted lines indicate the two examples in (**B–D**). (**B**) Output fluctuations $\delta z_i(t) = z_i(t) - \bar{z}_i(t)$ around the trial-conditioned average $\bar{z}$. Mean is over 16 samples for each of the two trial conditions (clockwise and anticlockwise rotation). Because both output dimensions $i \in \{1, 2\}$ are equivalent in scale, we collected both for the histogram. (**C, D**) Example output trajectories early (**C**) and late (**D**) in learning. Shown are the mean (dark) and five samples (light).

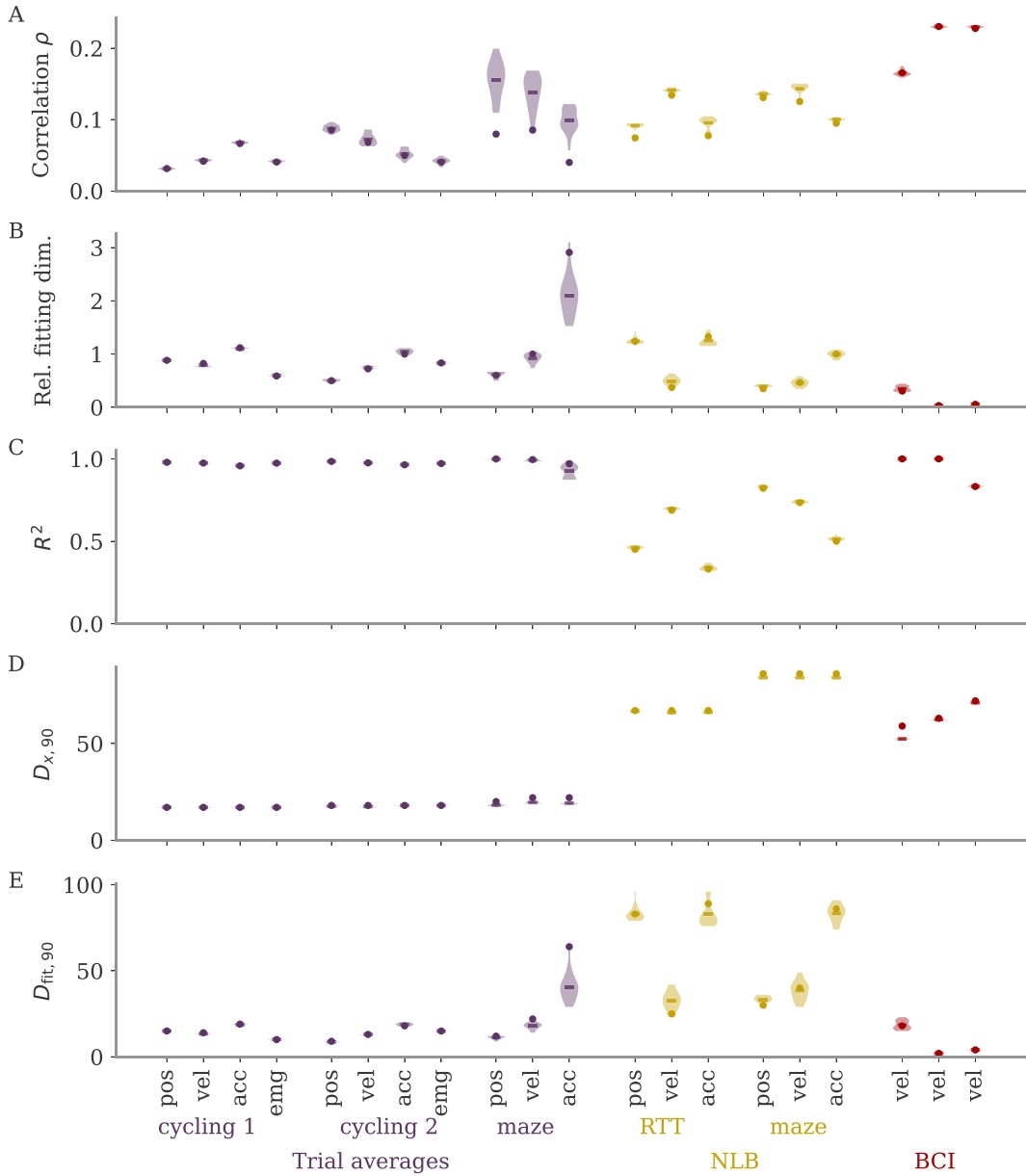

**Appendix 1—figure 8.** Fitting neural activity to different output modalities (hand position, velocity, acceleration, EMG). Output modality is indicated by the x-ticks, the corresponding data sets by the color and the labels below. (**A, B**) Correlation and relative fitting dimension. Similar to *Figure 7B*, where these are shown for velocity alone. (**C–E**) Additional details. (**C**) Coefficient of determination $R^2$ of the linear regression. (**D**) Number of principal components (PCs) necessary to reach 90% of the variance of the neural activity $X$. (**E**) Number of PCs necessary to reach 90% of the $R^2$ value of the full neural activity. For each output modality, the delay between activity and output is optimized. Position decodes earlier (300–200 ms) than velocity or acceleration (100–50 ms); no delay for EMG. The data $X$ is the same with each data set apart from a potential shift by the respective delay, so that dimension $D_{x,90}$ in (**D**) is almost the same. Note that we also computed trial averages for the Neural Latents Benchmark (NLB) maze task to test for the effect of trial averaging. These, however, have a small sample number (189 data points, but 162 neurons), which leads to considerable discrepancy between the dots (full data) and the subsamples with only 25% of the data points, for example in the correlation (**A**).

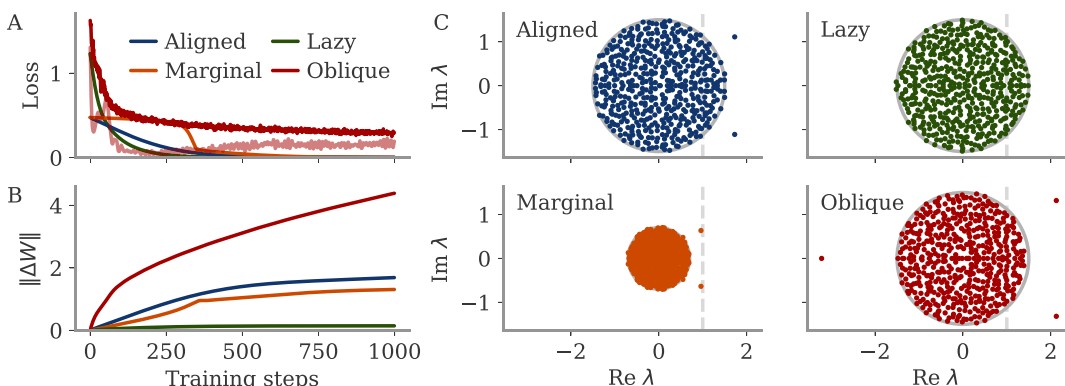

**Appendix 1—figure 9.** Learning curves and eigenvalue spectra for sine wave task. (**A**) Loss over training steps for four networks. The light red line is the bias term of the loss for the oblique network. (**B**) Norm of weight changes over learning time. (**C**) Eigenvalue spectra of connectivity matrix $W_f$ after training. The dashed line indicates the stability line for the fixed point at the origin.

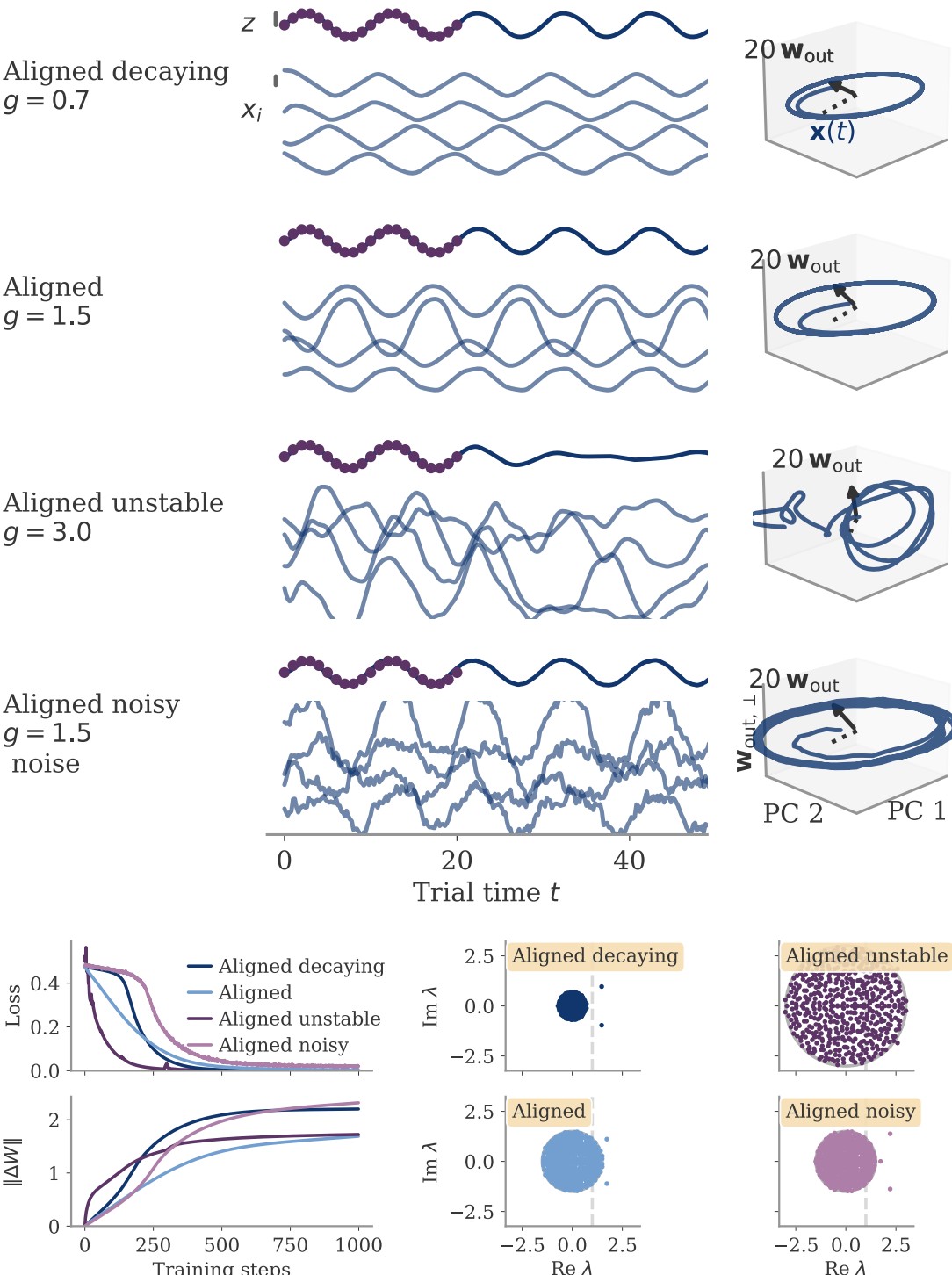

**Appendix 1—figure 10.** The aligned regime is robust to the choice of other hyperparameters.

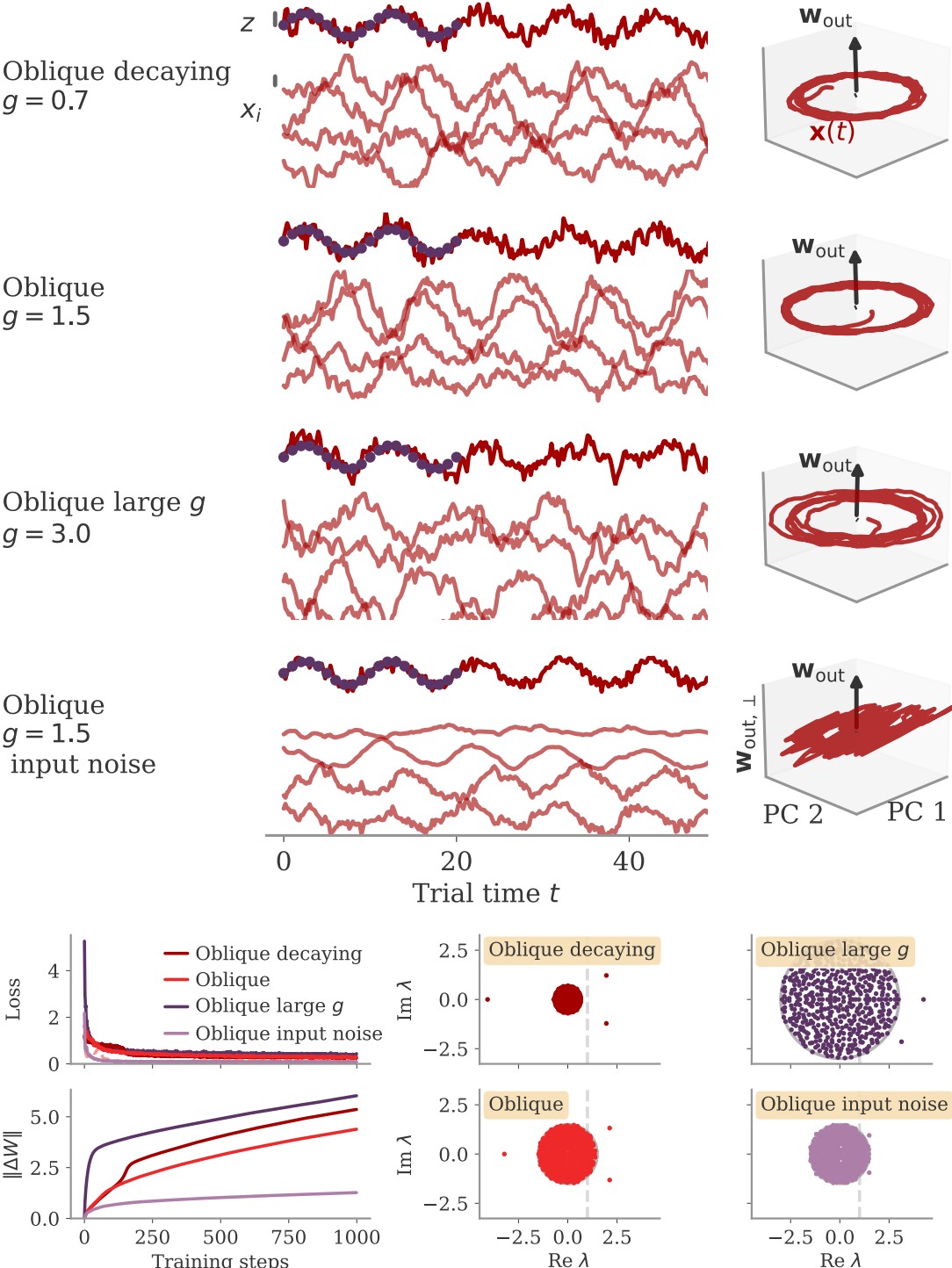

**Appendix 1—figure 11.** The oblique regime is robust to the choice of other hyperparameters.

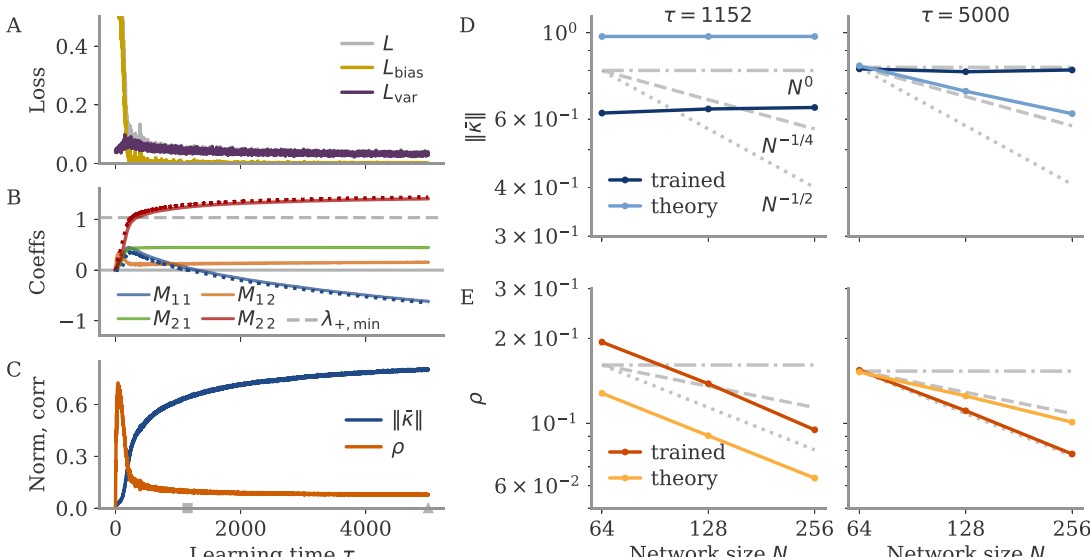

**Appendix 1—figure 12.** Training history leads to order-one fixed point norm. We trained recurrent neural networks (RNNs) on the example fixed point task. Similar to *Figure 13*, but with smaller noise $\sigma_{\text{noise}} = 0.2$, and with learning rate increased by 2 and number of epochs by 2.5. (A–C) Learning dynamics with gradient descent for one network with $N = 256$ neurons. The first 400 epochs are dominated by, $L_{\text{bias}}$ and $M_{11} \approx \lambda_-$ becomes *positive*. The negative feedback loop $\lambda_- < 0$, only forms later in learning. The matrix $M$ does not become symmetric during learning. (D, E) Fixed point norm and correlation for different $N$ evaluated when $\lambda_- = 0$ (left) and at the end of learning (right). The time points are indicated by a square and triangle in (C), respectively. At $\lambda_- = 0$, simulation and theory agree for the scaling: $\|\bar{k}\| = O(1)$ and $\rho = O(1/\sqrt{N})$. At the end of training, the theory predicts a decreasing fixed point norm, but the simulated networks inherit the order-one norm from the training history.

