## [Editor Report · eLife Assessment]

This work provides an **important** and novel framework for interpreting the interactions between recurrent dynamics across stages of neural processing. The authors report that two different kinds of dynamics exist in recurrent networks differing in the extent to which they align with the output weights. The authors also present **convincing** evidence that both types of dynamics exist in the brain.

---

## [Referee Report · Reviewer #1 (Public review)]

Summary:

In this work, authors utilize recurrent neural networks (RNNs) to explore the question of when and how neural dynamics and the network's output are related from a geometrical point of view. The authors found that RNNs operate between two extremes: an 'aligned' regime in which the weights and the largest PCs are strongly correlated and an 'oblique' regime where the output weights and the largest PCs are poorly correlated. Large output weights led to oblique dynamics, and small output weights to aligned dynamics. This feature impacts whether networks are robust to perturbation along output directions. Results were linked to experimental data by showing that these different regimes can be identified in neural recordings from several experiments.

Strengths:

Diverse set of relevant tasks

Similarity measure well chosen

Explored various hyperparameter settings

Weaknesses:

One of the major connections to found BCI data with neural variance aligned to the outputs. Maybe I was confused about something, but doesn't this have to be the case based on the design of the experiment? The outputs of the BCI are chosen to align with the largest principal components of the data.

Proposed experiments maybe have already been done (New neural activity patterns emerge with long-term learning, Oby et al. 2019). My understanding of these results is that activity moved to be aligned as the manifold changed, but more analyses could be done to more fully understand the relationship between those experiments and this work.

Analysis of networks was thorough, but connections to neural data were weak. I am thoroughly convinced of the reported effect of large or small output weights in networks. I also think this framing could aid in future studies of interactions between brain regions.

This is an interesting framing to consider the relationship between upstream activity and downstream outputs. As more labs record from several brain regions simultaneously, this work will provide an important theoretical framework for thinking about the relative geometries of neural representations between brain regions.

It will be interesting to compare the relationship between geometries of representations and neural dynamics across connected different brain areas that are closer to the periphery vs. more central.

Exciting to think about the versatility of the oblique regime for shared representations and network dynamics across different computations.

Versatility of oblique regime could lead to differences between subjects in neural data.

---

## [Referee Report · Reviewer #2 (Public review)]

Summary:

This paper tackles the problem of understanding when the dynamics of neural population activity do and do not align with some target output, such as an arm movement. The authors develop a theoretical framework based on RNNs showing that an alignment of neural dynamics to an output can be simply controlled by the magnitude of the read-out weight vector while the RNN is being trained: small magnitude vectors result in aligned dynamics, where low-dimensional neural activity recapitulates the target; large magnitude vectors result in "oblique" dynamics, where encoding is spread across many dimensions. The paper further explores how the aligned and oblique regimes differ, in particular that the oblique regime allows degenerate solutions for the same target output.

Strengths:

- A really interesting new idea that different dynamics of neural circuits can arise simply from the initial magnitude of the output weight vector: once written out (Eq 3) it becomes obvious, which I take as the mark of a genuinely insightful idea

- The offered framework potentially unifies a collection of separate experimental results and ideas, largely from studies of motor cortex in primate: the idea that much of the ongoing dynamics do not encode movement parameters; the existence of the "null space" of preparatory activity; and that ongoing dynamics of motor cortex can rotate in the same direction even when the arm movement is rotating in opposite directions.

- The main text is well written, with a wide-ranging set of key results synthesised and illustrated well and concisely

- Shows the occurrence of the aligned and oblique regimes generalises across a range of simulated behavioural tasks

- A deep analytical investigation of when the regimes occur and how they evolve over training

- Shows where the oblique regime may be advantageous: allows multiple solutions to the same problem; and differs in sensitivity to perturbation and noise

- An insightful corollary result that noise in training is needed to obtain the oblique regime

- Tests whether the aligned and oblique regimes can be seen in neural recordings from primate cortex in a range of motor control tasks

- The revised text offers greater clarity and precision about when the aligned and oblique regimes occur and in the interpretation of the analyses of neural data

Weaknesses:

- The depth of analytical treatment in the Methods is impressive; however, the paper and the Methods analyses are largely independent, with the numerous results in the latter not being mentioned in the Results or Discussion. It in effect operates as two papers.

---

## [Author Response]

The following is the authors’ response to the original reviews.

**Public Reviews:**

**Reviewer #1 (Public Review):**
Summary:In this work, the authors utilize recurrent neural networks (RNNs) to explore the question of when and how neural dynamics and the network's output are related from a geometrical point of view. The authors found that RNNs operate between two extremes: an 'aligned' regime in which the weights and the largest PCs are strongly correlated and an 'oblique' regime where the output weights and the largest PCs are poorly correlated. Large output weights led to oblique dynamics, and small output weights to aligned dynamics. This feature impacts whether networks are robust to perturbation along output directions. Results were linked to experimental data by showing that these different regimes can be identified in neural recordings from several experiments.Strengths:A diverse set of relevant tasks.A well-chosen similarity measure.Exploration of various hyperparameter settings.Weaknesses:One of the major connections found BCI data with neural variance aligned to the outputs.Maybe I was confused about something, but doesn't this have to be the case based on the design of the experiment? The outputs of the BCI are chosen to align with the largest principal components of the data.

The reviewer is correct. We indeed expected the BCI experiments to yield aligned dynamics. Our goal was to use this as a comparison for other, non-BCI recordings in which the correlation is smaller, i.e. dynamics closer to the oblique regime. We adjusted our wording accordingly and added a small discussion at the end of the experimental results, Section 2.6.

Proposed experiments may have already been done (new neural activity patterns emerge with long-term learning, Oby et al. 2019). My understanding of these results is that activity moved to be aligned as the manifold changed, but more analyses could be done to more fully understand the relationship between those experiments and this work.

The on- vs. off-manifold experiments are indeed very close to our work. On-manifold initializations, as stated above, are expected to yield aligned solutions. Off-manifold initializations allow, in principle, for both aligned and oblique solutions and are thus closer to our RNN simulations. If, during learning, the top PCs (dominant activity) rotate such that they align with the pre-defined output weights, then the system has reached an aligned solution. If the top PCs hardly change, and yet the behavior is still good, this is an oblique solution. There is some indication of an intermediate result (Figure 4C in Oby et al.), but the existing analysis there did not fully characterize these properties. Furthermore, our work suggests that systematically manipulating the norm of readout weights in off-manifold experiments can yield new insights. We thus view these as relevant results but suggest both further analysis and experiments. We rewrote the corresponding section in the discussion to include these points.

Analysis of networks was thorough, but connections to neural data were weak. I am thoroughly convinced of the reported effect of large or small output weights in networks. I also think this framing could aid in future studies of interactions between brain regions.This is an interesting framing to consider the relationship between upstream activity and downstream outputs. As more labs record from several brain regions simultaneously, this work will provide an important theoretical framework for thinking about the relative geometries of neural representations between brain regions.It will be interesting to compare the relationship between geometries of representations and neural dynamics across connected different brain areas that are closer to the periphery vs. more central.It is exciting to think about the versatility of the oblique regime for shared representations and network dynamics across different computations.The versatility of the oblique regime could lead to differences between subjects in neural data.

Thank you for the suggestions. Indeed, this is precisely why relative measures of the regime are valuable, even in the absence of absolute thresholds for regimes. We included your suggestions in the discussion.

**Reviewer #2 (Public Review):**
Summary:This paper tackles the problem of understanding when the dynamics of neural population activity do and do not align with some target output, such as an arm movement. The authors develop a theoretical framework based on RNNs showing that an alignment of neural dynamics to output can be simply controlled by the magnitude of the read-out weight vector while the RNN is being trained. Small magnitude vectors result in aligned dynamics, where low-dimensional neural activity recapitulates the target; large magnitude vectors result in "oblique" dynamics, where encoding is spread across many dimensions. The paper further explores how the aligned and oblique regimes differ, in particular, that the oblique regime allows degenerate solutions for the same target output.Strengths:- A really interesting new idea that different dynamics of neural circuits can arise simply from the initial magnitude of the output weight vector: once written out (Eq 3) it becomes obvious, which I take as the mark of a genuinely insightful idea.- The offered framework potentially unifies a collection of separate experimental results and ideas, largely from studies of the motor cortex in primates: the idea that much of the ongoing dynamics do not encode movement parameters; the existence of the "null space" of preparatory activity; and that ongoing dynamics of the motor cortex can rotate in the same direction even when the arm movement is rotating in opposite directions.- The main text is well written, with a wide-ranging set of key results synthesised and illustrated well and concisely.- The study shows that the occurrence of the aligned and oblique regimes generalises across a range of simulated behavioural tasks.- A deep analytical investigation of when the regimes occur and how they evolve over training.- The study shows where the oblique regime may be advantageous: allows multiple solutions to the same problem; and differs in sensitivity to perturbation and noise.- An insightful corollary result that noise in training is needed to obtain the oblique regime.- Tests whether the aligned and oblique regimes can be seen in neural recordings from primate cortex in a range of motor control tasks.Weaknesses:- The magnitude of the output weights is initially discussed as being fixed, and as far as I can tell all analytical results (sections 4.6-4.9) also assume this. But in all trained models that make up the bulk of the results (Figures 3-6) all three weight vectors/matrices (input, recurrent, and output) are trained by gradient descent. It would be good to see an explanation or results offered in the main text as to why the training always ends up in the same mapping (small->aligned; large->oblique) when it could, for example, optimise the output weights instead, which is the usual target (e.g. Sussillo & Abbott 2009 Neuron).

We understand the reviewer’s surprise. We chose a typical setting (training all weights of an RNN with Adam) to show that we don’t have to fine-tune the setting (e.g. by fixing the output weights) to see the two regimes. However, other scenarios in which the output weights do change are possible, depending on the algorithm and details in the way the network is parameterized. Understanding why some settings lead to our scenario (no change in scale) and others don’t is not a simple question. A short explanation here, nonetheless:

- Small changes to the internal weights are sufficient to solve the tasks.

- Different versions of gradient descent and different ways of parametrizing the network lead to different results in which parts of the weights get trained. This goes in particular for how weight scales are introduced, e.g. [Jacot et al. 2018 Neurips], [Geiger et al. 2020 Journal of Statistical Mechanics], or [Yang, Hu 2020, arXiv, Feature learning in infinite-width networks]. One insight from these works is that plain gradient descent (GD) with small output weights leads to learning only at the output (and often divergence or unsuccessful learning). For this reason, plain GD (or stochastic GD) is not suitable for small output weights (the aligned regime). Other variants of GD, such as Adam or RMSprop, don’t have this problem because they shift the emphasis of learning to the hidden layers (here the recurrent weights). This is due to the normalization of the gradients.

- FORCE learning [Sussillo & Abbott 2009] is somewhat special in that the output weights are simultaneously also used as feedback weights. That is, not only the output weights but also an additional low-rank feedback loop through these output weights is trained. As a side note: By construction, such a learning algorithm thus links the output directly to the internal dynamics, so that one would only expect aligned solutions – and the output weights remain correspondingly small in these algorithms [Mastrogiuseppe, Ostojic, 2019, Neural Comp].

- In our setting, the output is not fed back to the network, so training the output alone would usually not suffice. Indeed, optimizing just the output weights is similar to what happens in the lazy training regime. These solutions, however, are not robust to noise, and we show that adding noise during the training does away with these solutions.

To address this issue in the manuscript, we added the following sentence to section 2.2: “While explaining this observation is beyond the scope of this work, we note that (1) changing the internal weights suffices to solve the task, and that (2) the extent to which the output weights change during learning depends on the algorithm and specific parametrization [21, 27, 85].”

- It is unclear what it means for neural activity to be "aligned" for target outputs that are not continuous time-series, such as the 1D or 2D oscillations used to illustrate most points here.Two of the modeled tasks have binary outputs; one has a 3-element binary vector.

For any dynamics and output, we compare the alignment between the vector of output weights and the main PCs (the leading component of the dynamics). In the extreme of binary internal dynamics, i.e., two points {x_1, x_2}, there would only be one leading PC (the line connecting the two points, i.e. the choice decoder).

- It is unclear what criteria are used to assign the analysed neural data to the oblique or aligned regimes of dynamics.

Such an assignment is indeed difficult to achieve. The RNN models we showed were at the extremes of the two regimes, and these regimes are well characterized in the case of large networks (as described in the methods section). For the neural data, we find different levels of alignment for different experiments. These differences may not be strong enough to assign different regimes. Instead, our measures (correlation and relative fitting dimension) allow us to order the datasets. Here, the BCI data is more aligned than non-BCI data – perhaps unsurprisingly, given the experimental design of the prior and the previous findings for the rotation task [Russo et al, 2018]. We changed the manuscript accordingly, now focusing on the relative measure of alignment, even in the absence of absolute thresholds. We are curious whether future studies with more data, different tasks, or other brain regions might reveal stronger differentiation towards either extreme.

**Recommendations for the authors:**

**Reviewer #1 (Recommendations For The Authors):**
There's so much interesting content in the supplement - it seemed like a whole other paper! It is interesting to read about the dynamics over the course of learning. Maybe you want to put this somewhere else so that more people read it?

We are glad the reviewer appreciated this content. We think developing these analysis methods is essential for a more complete understanding of the oblique regime and how it arises, and that it should therefore be part of the current paper.

Nice schematic in Figure 1.There were some statements in the text highlighting co-rotation in the top 2 PCs for oblique networks. Figure 4a looks like aligned networks might also co-rotate in a particular subspace that is not highlighted. I could be wrong, but the authors should look into this and correct it if so. If both aligned and oblique networks have co-rotation within the top 5 or so PCs, some text should be updated to reflect this.

This is indeed the case, thanks for pointing this out! For one example, there is co-rotation for the aligned network already in the subspace spanned by PCs 1 and 3, see the figure below. We added a sentence indicating that co-rotation can take place at low-variance PCs for the aligned regime and pointed to this figure, which we added to the appendix (Fig. 17).

While these observations are an important addition, we don’t think they qualitatively alter our results, particularly the stronger dissociation between output and internal dynamics for oblique than aligned dynamics.

Figure 4 color labels were 'dark' and 'light'. I wasn't sure if this was a typo or if it was designed for colorblind readers? Either way, it wasn't too confusing, but adding more description might be useful.

Fixed to red and yellow.

Typo "Aligned networks have a ratio much large than one"Typo "just started to be explored" Typo "hence allowing to test"

Fixed all typos.

**Reviewer #2 (Recommendations For The Authors):**
- Explain/discuss in the main text why the initial output weights reliably result in the required internal RNN dynamics (small->aligned; large->oblique) after training. The magnitude of the output weights is initially discussed as being fixed, and as far as I can tell all analytical results (sections 4.6-4.9) also assume this. But in all trained models that make up the bulk of the results (Figures 3-6) all three weight vectors/matrices (input, recurrent, and output) are trained by gradient descent. It would be good to see an explanation or results offered in the main text as to why the training always ends up in the same mapping (small->aligned; large->oblique) when it could, for example, just optimise the output weights instead.

See the answer to a similar comment by Reviewer #1 above.

- Page 6: explain the 5 tasks.

We added a link to methods where the tasks are described.

- Page 6/Fig 3 & Methods: explain assumptions used to compute a reconstruction R^2 between RNN PCs and a binary or vector target output.

We added a new methods section, 4.4, where we explain the fitting process in Fig. 3. For all tasks, the target output was a time series with P specified target values in N_out dimensions. We thus always applied regression and did not differentiate between binary and non-binary tasks.

- Page 8: methods and predictions are muddled up: paragraph ending "along different directions" should be followed by paragraph starting "Our intuition...". The intervening paragraph ("We apply perturbations...") should start after the first sentence of the paragraph "To test this,...".

Right, these sentences were muddled up indeed. We put them in the correct order.

- Page 10: what are the implications of the differences in noise alignment between the aligned and oblique regimes?

The noise suppression in the oblique regime is a slow learning process that gradually renders the solution more stable. With a large readout, learning separates into two phases. An early phase, in which a “lazy” solution is learned quickly. This solution is not robust to noise. In a second, slower phase, learning gradually leads to a more robust solution: the oblique solution. The main text emphasizes the result of this process (noise suppression). In the methods, we closely follow this process. This process is possibly related to other slow learning process fine-tuning solutions, e.g., [Blanc et al. 2020, Li et al. 2021, Yang et al. 2023]. Furthermore, it would be interesting to see whether such fine-tuning happens in animals [Ratzon et al. 2024]. We added corresponding sentences to the discussion.

- Neural data analysis:(i) Page 11 & Fig 7: the assignment of "aligned" or "oblique" to each neural dataset is based on the ratio of D_fit/D_x. But in all cases this ratio is less than 1, indicating fewer dimensions are needed for reconstruction than for explaining variance. Given the example in Figure 2 suggests this is an aligned regime, why assign any of them as "oblique"?

We weakened the wording in the corresponding section, and now only state that BCI data leans more towards aligned, non-BCI data more towards oblique. This is consistent with the intuition that BCI is by construction aligned (decoder along largest PCs) and non-BCI data already showed signs of oblique dynamics (co-rotating leading PCs in the cycling task, Russo et al. 2018).

We agree that Fig 2 (and Fig 3) could suggest distinguishing the regimes at a threshold D_fit/D_x = 1, although we hadn’t considered such a formal criterion.

(ii) Figure 23 and main text page 11: discuss which outputs for NLB and BCI datasets were used in Figure 7 & and main text; the NLB results vary widely by output type - discuss in the main text; D_fit for NLB-maze-accuracy is missing from panel D; as the criterion is D_fit/D_x, plot this too.

We now discuss which outputs were used in Fig. 7 in its caption: the velocity of the task-relevant entity (hand/finger/cursor). This was done to have one quantity across studies. We added a sentence to the main text, p. 11, which points to Fig 22 (which used to be Fig 23) and states that results are qualitatively similar for other decoded outputs, despite some fluctuations in numerical values and decodability.

Regarding Fig 22: D_fit for NLB-maze-accuracy was beyond the manually set y-limit (for visibility of the other data points). We also extended the figure to include D_fit/D_x. We also discovered a small bug in the analysis code which required us to rerun the analysis and reproduce the plots. This also changed some of the numbers in the main text.

- Discussion:"They do not explain why it [the "irrelevant activity"] is necessary", implies that the following sentence(s) will explain this, but do not. Instead, they go on to say:"Here, we showed that merely ensuring stability of neural dynamics can lead to the oblique regime": this does not explain why it is necessary, merely that it exists; and it is unclear what results "stability of neural dynamics" is referring to.

We agree this was not a very clear formulation. We replaced these last three sentences with the following:

“Our study systematically explains this phenomenon: generating task-related output in the presence of large, task-unrelated dynamics requires large readout weights. Conversely, in the presence of large output weights, resistance to noise or perturbations requires large, potentially task-unrelated neural dynamics (the oblique regime).”

- The need for all 27 figures was unclear, especially as some seemed not to be referenced or were referenced out of order. Please check and clarify.

Fig 16 (Details for network dynamics in cycling tasks) and Fig 21 (loss over learning time for the different tasks) were not referenced, and are now removed.

We also reordered the figures in the appendix so that they would appear in the order they are referenced. Note that we added another figure (now Fig. 17) following a question from Reviewer #1.